# MXene-driven nanoscale field-effect junction for advanced 4-terminal perovskite/silicon tandem solar panels

Antonio Agresti [1,10] ✉, Sara Pescetelli[1,10] ✉, George Viskadouros[2], Anna Pazniak[3], Enrico Leonardi[4], Alessia Di Vito[1], Peyman Amiri[1], Matthias Auf Der Maur [1], Francesca Menchini [5], Silvano Del Gobbo[5], Francesco Di Giacomo[1], Giuseppe Bengasi[6], Carmelo Connelli[6], Luca Sorbello[4], Marina Foti[6], Francesco Bonaccorso [7], Emmanuel Kymakis [2,8] & Aldo Di Carlo [1,9] ✉

The commercialization of perovskite/silicon tandem solar cells hinges on achieving high efficiency and stability while maintaining scalability. This study demonstrates an original approach for inducing the formation of a field effect junction within the perovskite active layer for efficient semi-transparent top modules to be integrated in four-terminal perovskite/silicon tandem panels. A synergy of MXene-based doping and surface gradient passivation enabled semi-transparent perovskite modules with efficiencies surpassing 16% on 60 cm² active area. These were integrated into a four terminal tandem panel (0.2 m²) with a power conversion efficiency of 19.45%, further enhanced by bifacial silicon heterojunction cells to reach a power generation density exceeding 23 mWcm⁻² under 30% ground albedo conditions. The tandem panel, installed in Crete, retained over 95% of its initial delivered power after three months, showcasing robust real-world stability. This work provides a significant step toward industrial adoption, presenting a scalable, high-efficiency solution for next-generation photovoltaics with minimal modifications to silicon production lines.

The need to shift from fossil fuels to renewable energy has inspired the scientific community to innovate efficient and affordable photovoltaic (PV) technologies with low Levelized Cost of Electricity. A promising strategy involves integrating in tandem configuration emerging technologies such as perovskite solar cells (PSCs) (recently showing impressive power conversion efficiency (PCE) above 27%) with established solutions like silicon (Si) PV, now close to its theoretical PCE limit of 29.56%[1,2]. Small-area perovskite/Si cells have achieved efficiencies above 35%[3], yet replicating this success in larger devices has been challenging. Thus, developing and analyzing large-area perovskite devices that align with standard silicon wafer dimensions is crucial for addressing commercialization challenges[4]. Indeed, numerous advancements in large-area tandem devices have been announced but scientific report and publications

[1]CHOSE, Centre for Hybrid and Organic Solar Cells, University of Rome Tor Vergata, Roma, Italy. [2]Department of Electrical and Computer Engineering, Hellenic Mediterranean University, Heraklion, Greece. [3]Université Grenoble Alpes, CNRS, Grenoble INP, LMGP, Grenoble, France. [4]Halocell Europe SRL- Viale Castro Pretorio 122, Rome, Italy. [5]Energy Technologies and Renewable Sources Department, ENEA, C.R. Casaccia, Roma, Italy. [6]3SUN - Enel Green Power (EGP) SpA, Catania, Italy. [7]BeDimensional S.p.A., Genova, Italy. [8]Institute of Emerging Technologies, Hellenic Mediterranean University Research Center, Heraklion, Greece. [9]Istituto di Struttura della Materia, CNR-ISM, Roma, Italy. [10]These authors contributed equally: Antonio Agresti, Sara Pescetelli. ✉e-mail: antonio.agresti@uniroma2.it; sara.pescetelli@uniroma2.it; aldo.dicarlo@uniroma2.it

are few, indicating a competitive landscape for scaling these technologies.

Scaling up tandem architecture requires overcoming efficiency losses related to material processing, electrical interconnection, and stability. Among various tandem configurations[5], four-terminal (4T) architectures offer significant advantages, including independent maximum power point (MPP) tracking for the top and bottom cells, minimal modifications to existing Si fabrication lines[6], and compatibility with bifacial Si cells, which can capture reflected sunlight to enhance energy yield, although independent MPP trackers adds system-level cost[7,8]. Moreover, developing high-efficiency, scalable, and stable 4T perovskite/Si tandem modules presents several critical challenges. Firstly, a certain loss in PCE is inevitable when the typical opaque metal electrode used in top PSCs is replaced by a semi-transparent top electrode. Among the best-efficient semi-transparent PSCs (ST-PSCs), Yu and coworkers[9] succeeded to fabricate ST-PSCs with PCE of 20.25% by introducing $AZO/SnO_x$ as a sputtering buffer layer in an inverted configuration. Additionally, the absence of the metal contact acting as a back-reflector reduces the light-harvesting efficiency of ST-PSCs[10]. Moreover, optimizing the perovskite composition for tandem configurations often compromises light harvesting efficiency and current generation[11,12]. As a second challenge, the minimization of PCE losses when moving from small area ST-PSCs to semi-transparent perovskite solar modules (ST-PSMs) remains a key target for the practical realization of efficient tandem devices. Encouragingly, a 25.8% PCE has been reported for a large-area 4-terminal (4T) silicon/perovskite tandem module (2054 cm²) fabricated by LONGi Green Energy Technologies, where the top perovskite sub cells contribute 15.9% absolute to the overall efficiency[13]. Finally, addressing electrical connections, encapsulation, and lamination processes is crucial, as these have only been demonstrated with opaque devices[14]. Therefore, enhancing the PCE of ST-PSCs and ensuring their dimension scaling are pivotal for advancing perovskite/Si technology.

Commonly in 4T tandem devices perovskite absorber, wide-bandgap (WBG)[15] formulations are employed, but the resulting PCE of the ST-PSCs is still hampered by a significant open-circuit voltage ($V_{OC}$) deficit. This issue is mainly caused by non-radiative recombination due to deep-level acceptor defects that can be strongly hampered by the use of ammonium salts[16], such as phenylethyl ammonium iodide (PEAI) derivatives via anti-solvent additive engineering[17,18]. Among them, owing to its elevated electro-positivity, para-fluorinated PEAI (4-FPEAI) demonstrated stronger interaction with critical defects (iodide-lead (I-Pb) and iodide-cation antisite defects) present in WBG perovskites, resulting in $V_{OC}$ losses mitigation. Here, a surface gradient passivation (SGP) can be achieved when 4-FPEAI is introduced into the perovskite film during the anti-solvent step, resulting in a concentration gradient decreasing from the surface toward the perovskite bulk[19]. Moreover, 4-FPEAI confers a gradual p-type character to the film by favoring the energy level alignment across the PSC structure[19]. Indeed, tailoring the perovskite work function (WF) has emerged as a key strategy, alongside the design of homojunctions to modulate the absorber's electronic structure [20].

However, creating p-n homojunctions generally requires the deposition of two successive perovskite layers, increasing fabrication complexity. In our previous work, we demonstrated the use of two-dimensional (2D) materials, in particular MXenes, to dope the perovskite precursor solution, obtaining its WF tunability in a broad range without altering its optoelectronic properties [21].

Here, we propose a original approach to fabricate perovskite field-effect junctions arising from a modified interfacial electronic environment with a local n-type character at the buried perovskite layer, induced by chlorine-based MXene doping[21], and a surface-localized p-type shift promoted by 4-FPEAI based SGP. This intentional spatial asymmetry induces dipole-driven band bending at the perovskite interfaces, enabling more efficient carrier extraction and reduced

recombination. A 2D perovskite overlayer further strengthens the p-type character of 3D perovskite surface, thereby minimizing charge recombination at perovskite/hole transport layer interface. Notably, this strategy is fully scalable, enabling the fabrication of 4T large-area (44 × 44 cm²) perovskite/silicon tandem PV panels, incorporating bifacial Si technology and developing panel lamination protocols compatible with silicon industry standards.

Finally, the performance of tandem panels has been evaluated under real-world conditions in Crete, as laboratory results often differ from actual outdoor performance[22], underscoring the importance of field testing. The results demonstrated durability, with the tandem panel retaining over 95% of its initial maximum power output, primarily showing a slow fill factor (FF) degradation of 1.27% per month, expressed as a relative decrease respect to the initial FF. Additionally, leveraging the bifacial nature of the silicon heterojunction (Si-HJT) bottom cell, the system efficiently harnessed ground-reflected light, further enhancing power generation density (PGD) up to 23.6 mWcm⁻². These findings highlight the real-world viability of the proposed technology, bridging the gap between laboratory efficiency and large-scale deployment in diverse environmental conditions.

## Results

### Small area cells

Our reference n-i-p perovskite top cell (Fig. 1a) employs an electron transporting layer (ETL) based on graphene nanoflakes (G) to improve both charge transfer at the interface and cell stability[23,24]. Additional strategies implemented to further enhance the performance of the ST-PSCs include: (i) tuning the perovskite energy band gap; (ii) replacing hazardous anti-solvent with environmentally friendly alternatives and (iii) utilizing tailored 2D materials to enhance the absorber's optoelectronic properties. A detailed description of the fabrication process for the four structures of tested PSCs (Fig. 1) is reported in the experimental section.

The proposed $Cs_{0.18}FA_{0.82}Pb(I_{0.8}Br_{0.2})_3$ perovskite formulation features an energy gap ($E_g$) of 1.68 eV (see Supplementary Information (S.I.) Supplementary Fig. 1) which is particular crucial for the performance of perovskite/silicon tandem devices operating at temperature above $T > 55 °C$[25]. This consideration is especially relevant in our case, given the installation conditions on Crete Island. To reduce the environmental impact and human health risk, ethyl acetate (EA) is selected as green anti-solvent replacing chlorobenzene (CB) for the perovskite production[26,27]. The fabricated cells (Supplementary Fig. 2a) incorporating graphene nanoflakes and EA (G/EA) demonstrated comparable performance achieving a +2% increase of PCE compared to those utilizing CB (G/CB, as evidenced by Fig. 2 and Table 1). Furthermore, replacing CB with to EA, the perovskite post-deposition annealing time was reduced by half (30 min), effectively halving the energy required for perovskite absorber realization. SGP strategy[19] of the perovskite absorber was implemented by dissolving an optimized amount of 2D perovskite precursor (4-FPEAI) within EA anti-solvent step during perovskite deposition. SGP permits (i) to passivate deep-level trap states in Br-enriched WBG perovskite films caused by faster crystallization rate during perovskite solution process[28] and (ii) to reduce the energy-level mismatch between the WBG perovskite layer and charge transporting layers[19]. The SGP-based small area (0.1 cm²) opaque devices (G/4-FPEAI) showed superior PCE of +8% (+5.8%) with respect to G/CB (G/EA) cells (Fig. 2d), mainly due to improved $V_{OC}$ (+1.6% and +2% in G/EA and G/CB, respectively, Fig. 2a).

Finally, PCE up to 22.12% (+9.6% versus G/4-FPEAI best cell, Fig. 1d was obtained by doping perovskite/4-PEAI active layer with $Ti_3C_2Cl_2$ MXenes (MX-Cl) by carefully tuning the MXene dopant amount (Supplementary Fig. 2b). MX-Cl synthesis is reported in the experimental section while their morphological and chemical characterizations are reported in S.I. section S.I. 3 (Supplementary Figs. 3–6). The high performance of MX-Cl cells (G/MX/4-FPEAI) can be related to high $V_{OC}$

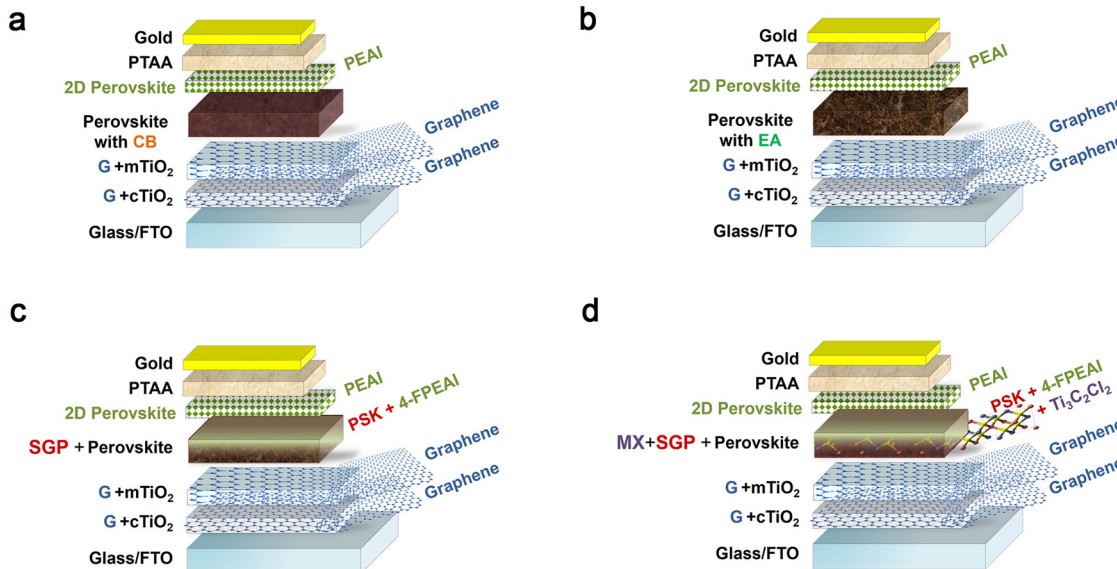

**Fig. 1 | Device architectures and 2D-material engineering strategies for mesoporous perovskite solar cells.** Four tested mesoporous solar cell structures: The reference structure realized employing graphene (G) within the ETL and by using of chlorobenzene-CB (**a**) or greener ethyl acetate-EA in view of the large scale production process (**b**) as anti-solvent during the perovskite layer deposition; (**c**) optimized structure realized with graphene based ETL, EA as green solvent for perovskite and surface gradient passivation-SGP strategy based on 4-FPEAI passivation to deposit the perovskite layer; (**d**) complete 2D materials engineered structure using G-modified ETL, EA as green solvent for perovskite with SGP strategy based on 4-FPEAI passivation and chlorine-based MXenes (MX-Cl) as dopant for perovskite layer.

(>1.19 V), significant enhancement in short circuit current density ($J_{SC}$) about +5% and +8% compared with G/4-FPEAI or G/EA respectively and minimized hysteresis in the I-V characteristics (Supplementary Fig. 7a–d) reflecting in a stabilization of the MPP under prolonged illumination (Supplementary Fig. 7e).

The MX-Cl role on perovskite films morphology was clarified by scanning electron microscopy (SEM). SGP has minimal impact on domain size and surface morphology (Supplementary Fig. 8), whereas MX-Cl doping, resulted in a noticeable increase in perovskite domains compared to the layer without MX-Cl addition (Fig. 3a, d). The presence of small domains around the enlarged ones in the MX-Cl-doped film indicated that MXene flakes locally influenced perovskite morphology (see Fig. 3b, e for the statistical analysis of domain's dimensions).

This is confirmed by the cross-section SEM images acquired on complete devices stacks for G/MX/4-FPEAI (Fig. 3f) and G/4-FPEAI (Fig. 3c). As shown by Liu et al.[29], the strong coordination between Cl atoms and $Pb^{2+}$ ions reduces metallic lead clusters ($Pb^0$), which are known to induce deep defects and trap free charge carriers in the perovskite films, leading to non-radiative recombination.

X-ray diffraction (XRD) analysis (Fig. 3g) reveals improved crystallinity in MX-Cl/4-FPEAI films, evidenced by stronger diffraction peaks for the (110), (220), and (310) lattice planes of cubic perovskite structure, reflecting in reduces lattice distortion, thereby lowering defect density and minimizing trap states. The introduction of MX-Cl improves perovskite crystallinity and promotes preferred orientation growth (section S.I. 5, Supplementary Table 1) due to the interaction between the Cl terminations and the $Pb^{2+}$ ions forming adducts act as heterogeneous nucleation site for the perovskite film[29]. Complementarily, TRPL analysis (Fig. 3h) shows a marked increase in average carrier lifetime in MXene-modified films, pointing to efficient trap-state passivation, likely due to both improved crystallinity and reduced interfacial defect density (see Supplementary Table 2 in the S.I. for full fitting parameters).

The impact of both MX-Cl and 4-FPEAI on the trap assisted recombination and on perovskite crystallinity was assessed by transient photovoltage (TPV), Photoluminescence (PL), $V_{OC}$ vs Light

intensity, XRD and UV–Vis measurements as reported in S.I. (section S.I. 5). TPV measurements (Supplementary Fig. 9a) reveal that G/MX/4-FPEAI significantly extends charge recombination lifetime versus G/EA and G/4-FPEAI. This indicates that, beyond creating a p-doping gradient from ETL to HTL, SGP also provides residual passivation, enhancing $J_{SC}$ (Fig. 2b). Additionally, devices with MX-Cl exhibit higher External Quantum Efficiency (EQE) values across entire spectral range versus 4-FPEAI devices with enhance EQE-calculated $J_{SC}$ (from 21.01 mA cm$^{-2}$ for 4-FPEAI to 22.3 mA cm$^{-2}$ for G/MX/4-FPEAI), reflecting enhanced charge dynamics (Supplementary Fig. 9b) and absorbance (Supplementary Fig. 10a). Furthermore, MX-doping reduces defect state density by enlarging perovskite domains and mitigating grain boundary-induced trap states. Steady-state PL measurements (Supplementary Fig. 10b) confirm the impact of MX-Cl, showing a 75% increase in PL intensity and a slight red shift (from 758 to 763 nm) for MX-Cl/4-FPEAI films, assessing the reduced impact of the non-radiative recombination. Trap-assisted recombination at mTiO$_2$/perovskite interface was also studied via $V_{OC}$ vs. incident light power ($P_{inc}$) measurements (Supplementary Fig. 9d) and dark-$J$-$V$ analysis (Supplementary Fig. 10c). Adding MX-Cl in G/MX/4-FPEAI lowers the $V_{OC}(P_{inc})$ slope, reducing ETL/perovskite recombination at this interface, and boosting $J_{SC}$[30], while the decrease of both reverse leakage current and ideality factor in dark $J$-$V$ curves is in line with the longer TRPL lifetimes and higher PL intensity.

The reduction of trap states in MX-Cl/4-FPEAI-modified perovskites can be attributed to the strong interaction between Cl terminations of MX-Cl and $Pb^{2+}$ ions, which mitigates non-radiative recombination caused by uncoordinated lead sites. This interaction, stronger than that one between carbonyl-groups and $Pb^{2+}$ as in case of Ti$_3$C$_2$T$_x$-based MXenes, makes its effect on crystallization and domain size here more pronounced, compared to our previous work[21], yielding a more stable perovskite lattice.

From SEM images of Fig. 3f, clearly MX-Cl flakes are mainly distributed at bottom interface between perovskite/mTiO$_2$ layer. The presence of MX-Cl at the ETL/perovskite interface calls for a possible tuning of the buried perovskite interface WF [21].

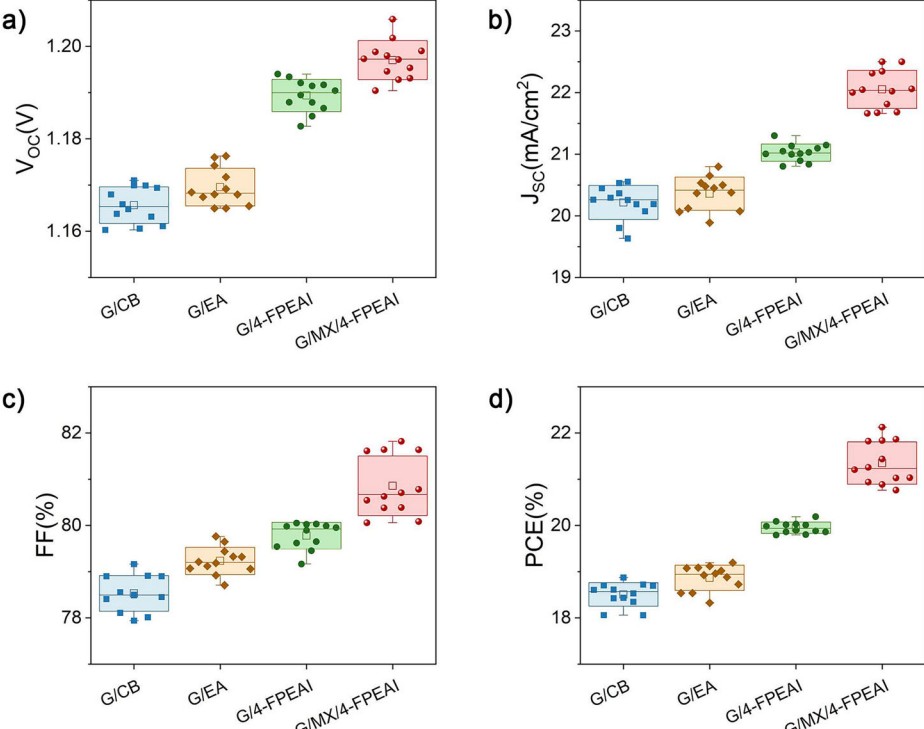

**Fig. 2 | Electrical performance statistics of opaque perovskite solar cells with different interface engineering strategies.** Electrical parameter statistics (12 samples) (**a**) open circuit voltage ($V_{OC}$); (**b**) short circuit current density ($J_{SC}$); (**c**) fill factor (FF); (**d**) power conversion efficiency (PCE), for the four investigated opaque PSC structures extracted by the current-voltage (I-V) characteristics acquired under 1 SUN irradiation. In each box plot, the square indicates the mean value, the horizontal line represents the median, the box corresponds to the standard deviation, and the whiskers extend to the minimum and maximum values. Individual symbols represent independent devices. Forward and reverse scan J–V curves for all four opaque device architectures are provided in the Supplementary Information- S.I. (Supplementary Fig. 7a–d). The minimized hysteresis in the G/MX/4-FPEAI device highlights the improved stability and reduced interfacial recombination enabled by the device engineering via field-effect junction strategy.

Building on this, ultraviolet photoelectron spectroscopy (UPS) was employed to elucidate the progressive modulation of the perovskite energy levels depending on the different treatments. The measurements were carried out on ad-hoc fabricated samples (see section S.I. 6 in S.I.). Notably, the combination of MXene and 4-FPEAI induces a graded shift in WF and valence band maximum (VBM) position across the series, consistent with dipole-induced band bending localized at the perovskite interfaces. A full discussion and corresponding UPS spectra and energy level diagram reconstructed from the UPS data are provided in section S.I. 6 in S.I., Supplementary Fig. 11, while the extracted values of WF and VBM are summarized in Table 2, clearly highlighting the energetic alignment changes induced by each modification. To exclude charging or chemical artefacts, X-ray photoemission spectroscopy (XPS) analyzes were performed, confirming that the observed WF and VBM shifts arise from genuine interfacial electronic modifications. (see Supplementary Fig. 12 for detailed discussion).

To confirm this effect, Kelvin probe force microscopy (KPFM) measurements were carried out on the same samples, allowing us to extract WF values (Supplementary Table 3) and to distinguish the contributions of MXene doping and SGP (see Supplementary Fig. 13).

Density functional theory (DFT) simulations of the MX-Cl/perovskite/4-FPEAI system can elucidate the passivation capability of 4-FPEAI and the impact of MX-Cl on the electronic characteristics of perovskite. As a trade-off between accuracy and computability, single cation MAPbI₃ perovskite was considered, as discussed in section S.I. 7 in S.I. along with computational details about DFT simulations (see Supplementary Fig. 14). The structure of MX-Cl/perovskite/4-FPEAI (Fig. 4a), as obtained by structural optimization, shows that no covalent bonds are involved at perovskite/MXenes interface. However, the

interaction results in a reduction of the WF versus undoped perovskite (−0.39 eV, Supplementary Table 4), as highlighted by the slope of the vacuum potential, indicated with red arrow in the figure. The WF reduction results from band bending induced by the surface dipole not from a shift of the Fermi level, here assumed to be in the middle of the perovskite gap. Moreover, the projected density of states (PDOS) analysis underlines perovskite $E_g$ remains mostly unaffected, and MX-Cl additive introduces donor states near perovskite conduction band edge (Supplementary Fig. 15), confirming the locally induced perovskite n-character at the MX-Cl/perovskite interface. Notably, same qualitative outcomes result for MAPbBr₃ interacting with MX-Cl (Supplementary Table 4, and Supplementary Fig. 16). Conversely, the interaction of perovskite with 4-FPEAI results in the formation of covalent bonds between iodine atoms of the additive molecules and lead atoms on the perovskite surface, while the intercalation of 4-FPEA acts as a cation for the perovskite inorganic cage. The WF shift caused by 4-FPEAI (0.14 eV, Supplementary Table 4) is opposite to that of the MX-Cl additive, as shown by the opposite slopes of the vacuum level and red arrow directions. Finally, the PDOS analysis of perovskite/4-FPEAI interface (Supplementary Fig. 15) shows that the passivation of the states contributed by the Pb ions on the perovskite surface results in an up-shift of the conduction band edge of perovskite, giving a p-type behavior of the doped perovskite with respect to the pristine material.

Although enlarging the domain size leads to a slight increase in UV–Vis absorption (Supplementary Fig. 10a), it only partially explains the $J_{SC}$ enhancement observed in G/MX/4-FPEAI devices. Indeed, the dipole induced by the MX-Cl addition at the mTiO₂/perovskite interface improves charge-transfer efficiency and consequently $J_{SC}$. Thus, the reduction of J-V curve hysteresis (Supplementary Fig. 7) through

**Table 1 | Electrical photovoltaic parameters for the four investigated opaque PSC structures extracted by the I-V characteristics acquired under 1 SUN irradiation in reverse scan reported as averaged values±standard error obtained on 12 samples for each cell typology and for the best performing devices**

| Device | | $V_{OC}$(V) | $J_{SC}$(mA cm$^{-2}$) | FF(%) | PCE(%) |
|---|---|---|---|---|---|
| G/CB | Champion | 1.17 | 20.55 | 79.16 | 18.87 |
| | Average | 1.166 ± 0.001 | 20.22 ± 0.08 | 78.53 ± 0.11 | 18.51 ± 0.07 |
| G/EA | Champion | 1.176 | 20.79 | 79.76 | 19.19 |
| | Average | 1.170 ± 0.001 | 20.36 ± 0.08 | 79.23 ± 0.09 | 18.87 ± 0.08 |
| G/4-FPEAI | Champion | 1.193 | 21.30 | 80.05 | 20.18 |
| | Average | 1.190 ± 0.0006 | 21.03 ± 0.04 | 79.78 ± 0.09 | 19.95 ± 0.04 |
| G/MX/ 4-FPEAI | Champion | 1.205 | 22.5 | 81.82 | 22.12 |
| | Average | 1.197 ± 0.001 | 22.05 ± 0.09 | 80.86 ± 0.19 | 21.34 ± 0.13 |

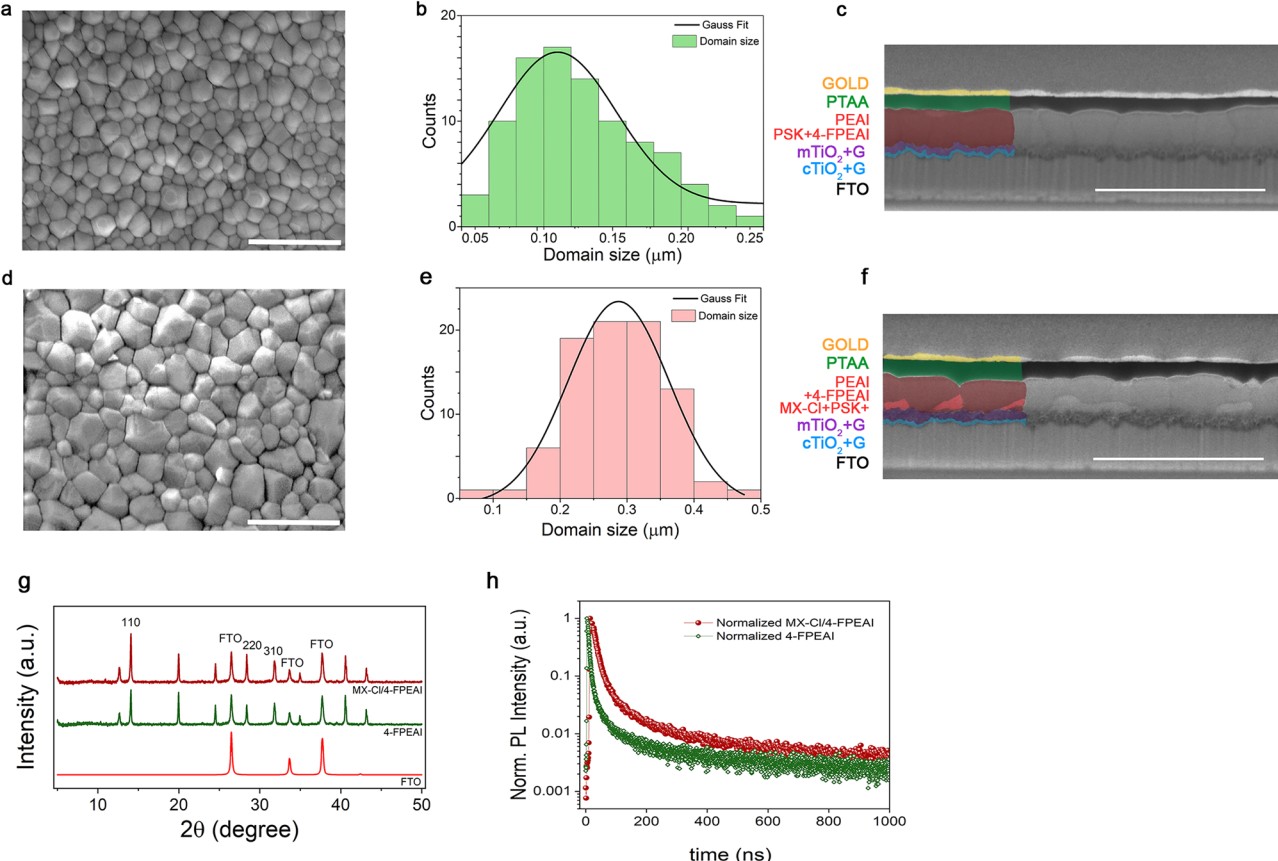

**Fig. 3 | Structural, morphological and optoelectronic effects of MXene incorporation in perovskite films. a** SEM image of a perovskite layer without MX-Cl addition and (**d**) MX-Cl doped perovskite film surface obtained by applying the SGP strategy during the anti-solvent step. The cross-section SEM images for the same samples are reported in (**c**) and (**f**) respectively. Scale bars: 1 μm (**a**, **d**) and 2 μm (**c**, **f**). In the colored part of (**f**) the incorporation of MX-Cl flakes is highlighted with a light-red color showing that MX flakes are sitting at the mTiO₂/perovskite interface, while they are not present in the case of pristine perovskite film in (**c**). The presence of small domains still appearing around the enlarged domains in the case of MX-Cl-doped film elucidated that the presence of MXene flakes induced a localized effect on the perovskite morphology, acting as a template for the perovskite crystals growing on top of them. **b**, **e** Statistical distribution of domain sizes extracted from SEM images for the corresponding films. The histograms are fitted with Gaussian functions, indicating the average domain size and size dispersion for the the perovskite modified with only 4-FPEAI film (**b**) and the treated with both 4-FPEAI and

MXenes film (**e**). Domain size distribution reveals significant differences between the two films. In the sample containing only 4-FPEAI, the average domain size is 0.125 μm, with a standard deviation of 0.047 μm, while in the MX-Cl/4-FPEAI sample the mean increases to 0.286 μm, with a standard deviation of 0.070 μm. Furthermore, the maximum domain size rises from 0.242 μm (4-FPEAI) to 0.452 μm (MX-Cl/4-FPEAI), indicating a more pronounced domain growth. These results suggest that the incorporation of MXenes promotes more efficient crystal growth and greater grain coalescence, which is consistent with a reduction in grain boundary defect density and potentially improved charge transport within the film. **g** XRD patterns for the perovskite films with and without MX-Cl. **h** Normalized transient photoluminescence (TRPL) decay curves of the 4-FPEAI perovskite film and the film incorporating Cl-functionalized MXene. The decay profiles were fitted with a bi-exponential model. The MXene-treated film exhibits a significantly longer average carrier lifetime ($\tau_{avg}$ = 1.11 × 10$^{-7}$ s) compared to the 4-FPEAI ($\tau_{avg}$ = 6.88 × 10$^{-8}$ s), indicating reduced trap-assisted recombination and improved defect passivation.

MX-Cl engineering is mainly attributed to the decrease of charge accumulation at the interface.

We made use of physical device simulations to account for the WF shift, grading and charge recombination changes solar induced by MXenes and 4-FPEAI in the cell electro-optical properties (Section S.I. 8 in S.I., Supplementary Table 5). Figure 4b shows the simulated J-V curves for the three PSC typologies: reference, 4-FPEAI and 4-FPEAI + MX-Cl. The reference curve nicely reproduces the experimental $J_{SC}$ and $V_{OC}$, (insets Fig. 4b). In agreement with measurements, the 4-FPEAI additive boosts $V_{OC}$ to 1.19 eV, and MX-Cl further increases it to 1.20 eV due to dipole-induced electron accumulation at the buried interface and reduced recombination. This also improves FF, evidenced in Fig. 4b from blue J-V characteristic versus the green and black ones. The field-effect junctions arising from spatially separated interfacial dipoles improve $V_{OC}$, especially with 4-FPEAI additive, enhancing hole extraction, as demonstrated by calculated hole density at the p-side perovskite interface (Supplementary Fig. 17).

## Four-terminal tandem panels

The ST-PSCs for 4T tandem were obtained replacing the opaque top-electrode with an ultra-thin gold layer and a sputtered indium thin-oxide (ITO, Supplementary Fig. 18). By employing the same 2D material-engineering strategy, MXene doping and 4-FPEAI-based SGP, previously validated in opaque devices, the ST-PSCs used in the 4T tandem architecture demonstrated enhanced PCEs up to 18.3% (av. PCE = 17.63%) due to a higher $V_{OC}$ (av. $V_{OC}$ = 1.2 V) and $J_{SC}$ (av. $J_{SC}$ = 19.92 mA cm$^{-2}$) compared to the G/EA (REF) sample (see Supplementary Table 6, and Supplementary Figs. 19–22a, 20 and 21 for a detailed analysis), confirming the effectiveness of 2D material-engineering approach even for ST structures[31,32]. Moreover, the combination among the MX-Cl addition and the 4-FPEAI SGP strategy not only translated into superior device performance but also conferred superior robustness on the perovskite film against light-driven degradation, as demonstrated by prolonged light cycling test (Supplementary Fig. 22c) and combined heating + light soaking test (Supplementary Fig. 22d).

ST-PSCs were scaled up to module and to 4T PSK/Si panel detailed as follows. The module layout (Fig. 5a) consists of 24 series connected solar cells on a 9.5 × 9.5 cm$^2$ substrate area (cell width of 3.1 mm). Laser ablation parameters (P1-P2-P3 processes) have been optimized for maximizing the module active area by ensuring a geometrical FF > 96% (Supplementary Figs. 23 and 24)[33,34].

Four parallel-connected ST-PSMs covering one M2 Si-HJT bottom cell were employed to realize a first PSK/Si tandem panel demonstrator (DEM1) (Fig. 5, and Supplementary Fig. 25). The total active area of the four perovskite top modules was matched to the Si cell active area (240 cm$^2$). The four ST-PSMs and the silicon cell were laminated together by employing an industrial laminator (S.I., section S.I. 12, Supplementary Fig. 26).

Figure 6 shows the picture of the fabricated 4T PSK/Silicon tandem together with the I-V characteristics for the best ST-PSM, obtained with optimized 2D material strategy, as well as the filtered Si-HJT single cell and the perovskite top ST panel. This confirms the scalability and effectiveness of the approach at module level (section S.I. 11 in S.I.).

The best efficient 2D material-engineered ST-PSM reached a PCE > 16%, surpassing the state of art for ST-PSCs used in record 4T large-area tandem cells[13], while strongly limiting the PCE drop (<13%) usually observed in scaling the device size (see S.I. section S.I. 11 and Supplementary Table 7 for detailed discussion). Meanwhile, the 4T-DEM1 device reached a PCE of ~21%, underscoring the effectiveness of both the 2D materials and the tailored lamination procedure. To demonstrate the high reproducibility of the developed ST-PSM fabrication protocol, we produced 150 2D material-engineered ST-PSMs, with their PCE statistics shown in Fig. 7a. Ultimately, incorporating 2D materials yielded high-performing ST-PSMs, reduced performance

**Table 2 | Extracted values of work function (WF), valence band maximum (VBM) with respect to the Fermi energy level (EF), and ionization energy (IE = WF + VBM) for the four analysed perovskite film configurations, measured by UPS**

| Sample | WF (eV) | VBM (eV) | (IE) [eV] |
|---|---|---|---|
| REF | 4.75 ± 0.1 | 0.9 ± 0.1 | 5.65 ± 0.1 |
| MX-Cl | 4.55 ± 0.1 | 0.9 ± 0.1 | 5.45 ± 0.1 |
| MX-Cl/4-FPEAI | 4.70 ± 0.1 | 0.8 ± 0.1 | 5.50 ± 0.1 |
| 4-FPEAI | 4.85 ± 0.1 | 0.7 ± 0.1 | 5.55 ± 0.1 |

The results demonstrate distinct electronic effect induced by MXene and 4-FPEAI treatments, supporting the formation of dipoles and spacially separated charge redistribution.

variability, and enabled reproducible production of large area PSK/Si tandem panels, thereby advancing industrially oriented production of the perovskite/Si tandem technology [35].

As a final demonstrator, a 4T tandem panel (DEM2, Fig. 7b, c) was fabricated by connecting in parallel 16 ST-PSMs, while 4 bifacial Si-HJT cells from 3SUN company were series connected. Both were laminated in "one-step" process, creating a single tandem object electrically accessible via the 4 external terminals (S.I. section S.I. 12). The tandem panel (DEM2) was measured outdoor in Rome (41.85371; 12.63508) in a clear sky day (06/08/2023, Supplementary Fig. 27) and the IV-characteristics are reported in Fig. 7d, e for the perovskite top panel and Si-HJT bottom module, respectively. The top semi-transparent panel (ST-PSP) achieved a PCE ~ 12.5%, with negligible relative PCE reduction (−4.5%) compared to PCE of ST-PSP for DEM1 (+13.1%), despite quadrupling the number of parallel connected ST-PSMs. Thus, the lamination approach developed in this work (S.I., section S.I. 12 and 13) minimally impacts the perovskite panel (PSP) performance once the dimensions of the panel are scaled up. The filtered Si-HJT bottom panel measured at 1 SUN approached 7%, contributing to a final PCE of DEM2 reaching 19.45%. We should point out that, due to the manual stringing (electrical connection among adjacent cells), the bottom Si-HJT module has a reduced efficiency compared to those obtained on an automatized industrial line (S.I., section S.I. 13). DEM2 was finally installed in Crete Island and its performance was monitored in outdoor conditions by following the ISOS-O-2 protocol.

The tandem panel's electrical characteristics were studied under varying irradiance and temperature (S.I., section S.I. 14). Outdoor measurements performed using a metallic mounting base were carried out with the module installed at a fixed inclination of 30° and a height of ~100 cm above ground (Fig. 7g). From the outdoor performance monitoring at open circuit conditions, the I-V curves for top ST-PSP were acquired during a 3-months period (from August to November 2023) and the main electrical parameters were normalized in STC (1000 W m$^{-2}$ irradiance and 25 °C temperature) considering both the temperature and irradiance dependency of the main electrical parameters (S.I., section S.I. 14). Notably, the ST-PSP constituting DEM2 retained over 95% of its initial $P_{MPP}$ value (Fig. 7f) during outdoor exposure, while mainly its FF was showing a slow decreasing trend over time (−1.27%/month). No evident performance losses were observed for the Si-HJT bottom module. Despite degradation was not the focus of this study, the gradual decline in performance for ST-PSP can be ascribed to the well-known intrinsic degradation mechanisms affecting ST-PSM[36], since no evident lamination failure of the tandem panel was detected upon dismounting.

The bifaciality of the Si-HTJ bottom panel allowed us to estimate the final PGD of the 4T tandem perovskite/bifacial Si-HJT under varying reflective irradiance (albedo) from different ground materials (S.I., section S.I. 14). To date, the natural albedo of soil in solar installations ranges from 10% to 30%; the PGD for DEM2 was estimated to range from 21.15 mW cm$^{-2}$ to 23.66 mW cm$^{-2}$ (Table 3).

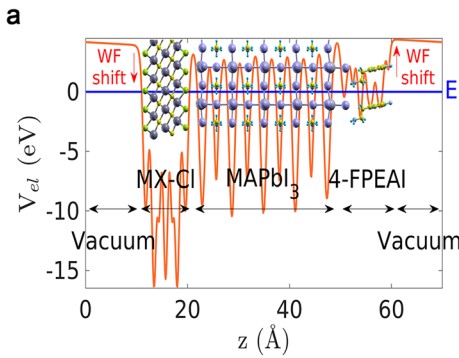
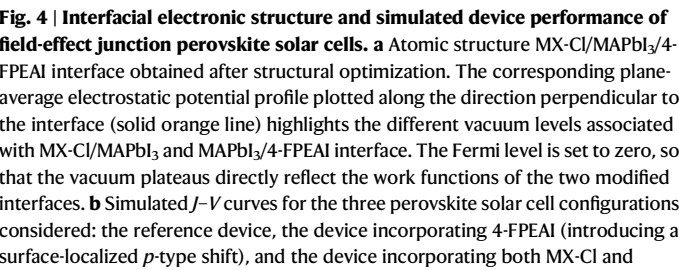
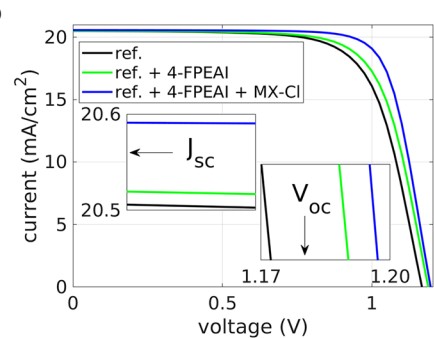

**Fig. 4 | Interfacial electronic structure and simulated device performance of field-effect junction perovskite solar cells. a** Atomic structure MX-Cl/MAPbI$_3$/4-FPEAI interface obtained after structural optimization. The corresponding plane-average electrostatic potential profile plotted along the direction perpendicular to the interface (solid orange line) highlights the different vacuum levels associated with MX-Cl/MAPbI$_3$ and MAPbI$_3$/4-FPEAI interface. The Fermi level is set to zero, so that the vacuum plateaus directly reflect the work functions of the two modified interfaces. **b** Simulated $J–V$ curves for the three perovskite solar cell configurations considered: the reference device, the device incorporating 4-FPEAI (introducing a surface-localized $p$-type shift), and the device incorporating both MX-Cl and

4-FPEAI. In the latter case, MX-Cl induces a modified electronic environment at the buried interface (localized $n$-type character), manifested as enhanced electron accumulation and reduced recombination, while 4-FPEAI ($p$-type doping profile) modulates the electronic structure at the top surface. Together, these interfacial modifications improve carrier extraction across the perovskite layer. The two insets display the corresponding $J_{SC}$ and $V_{OC}$ values for the three simulated cases. The $J–V$ curves are obtained by coupling a transfer-matrix method (TMM) for the optical response with Poisson/drift-diffusion (DD) calculations for the electrical properties (TMM/DD simulations).

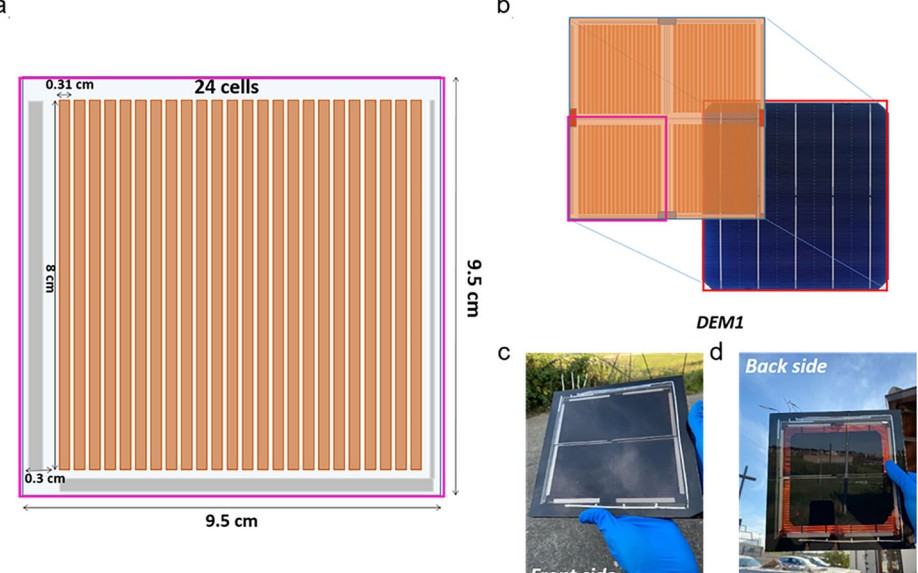

**Fig. 5 | Design and realization of semi-transparent perovskite modules and tandem demonstrators. a** Layout of the semi-transparent 2D material-based PSMs. Each module is composed by 24 series-connected solar cells with an active area of 2.49 cm$^2$. The total active area is 60 cm$^2$ while the aperture area (comprising the

interconnection areas) is about 63 cm$^2$. **b** Demonstrator 1 (DEM1) perovskite/Si tandem panel. Each building block is composed of four parallel-connected semi-transparent perovskite modules stacked above the M2 Si-HJT bifacial cell (provided by 3SUN). **c, d** Pictures of the front and back side of the laminated tandem DEM1.

This study explores the optimization and scalability of 4T perovskite/silicon tandem devices. A field-effect junction within the perovskite layer by combining the SGP strategy with Cl-based MXene doping yielded to ST 60 cm$^2$ active area modules with top PCE > 16%, following integrated into a 0.2 m$^2$ 4T perovskite/silicon tandem panel. The as-developed all-in-one lamination process is compatible with the existing Si-HJT production lines, marking a significant step toward commercialization. Moreover, outdoor panel performance at STC (PCE of 19.45%) and in presence of 30% ground albedo conditions (PGD > 23 mW cm$^{-2}$) showed remarkable stability during three operative months.

Crucially, this work introduces a perovskite field-effect junctions, enabled by the spatially resolved integration of Cl-terminated MXenes

and 4-FPEAI, representing an advanced approach to interface and energy landscape engineering in perovskite photovoltaics.

Future efforts will focus on device cost reduction (i.e., Au/ITO transparent electrode replacement with graphene[37]), refining production methods to enhance scalability and economic viability. Optimizations like 2T voltage-matched architecture could further reduce costs and improve spectral resilience. Post-operation diagnostic analysis of field-deployed modules could offer valuable insight into the intrinsic degradation pathways and stability limitations of the proposed device architecture. Overall, this study demonstrates significant progress toward efficient, scalable, and commercially viable tandem solar solutions, underscoring the transformative potential of integrating 2D materials into photovoltaic technology.

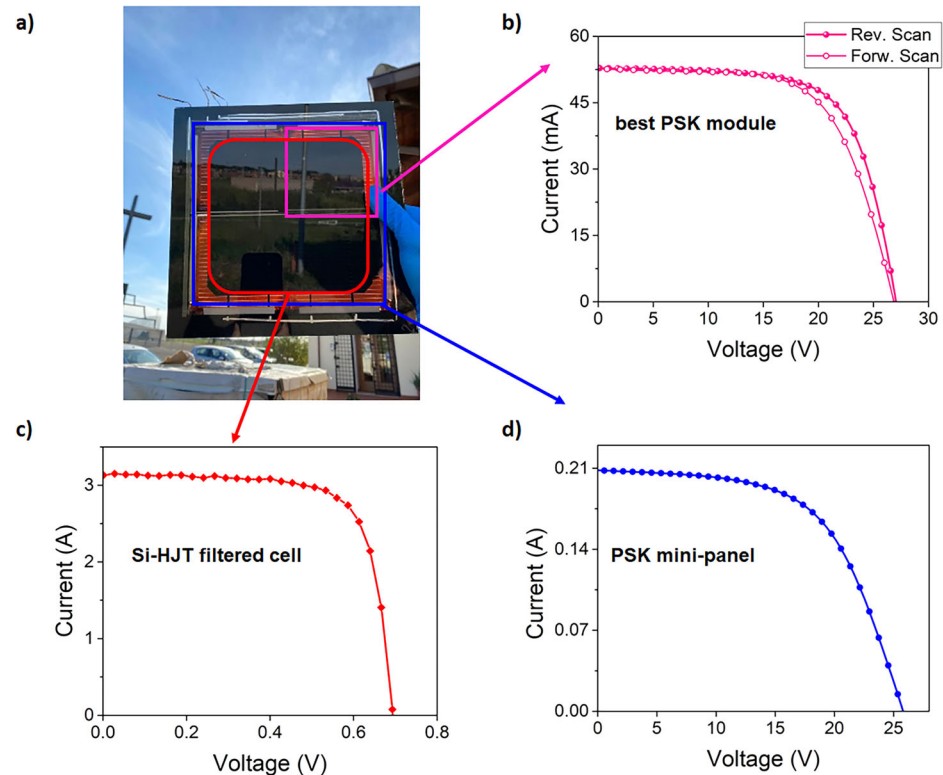

**Fig. 6 | Electrical characterization of the semi-transparent perovskite modules and tandem demonstrator. a** Front picture of a 2D material-engineered tandem demonstrator 1 (DEM1). **b** *I-V* curves acquired under forward (Forw) $V_{OC}$ = 26.8 V, $I_{SC}$ = 52.66 mA, $P_{MPP}$ = 9.03E-1W, FF = 63.96%, PCE = 15.1% and reverse (Rev) $V_{OC}$ = 26.98 V, $I_{SC}$ = 52.7 mA, $P_{MPP}$ = 9.66E-1 W, FF = 67.9%, PCE = 16.16% voltage scan under 1 Sun irradiation for the best performing ST-PSM based on 2D materials. **c** *I-V* curve for the Si-HJT M2 sized cells filtered by the perovskite top panel $V_{OC}$ = 6.9E-1V, $I_{SC}$ = 3.12 A, $P_{MPP}$ = 1.6 W, FF = 74.3%, PCE = 7.78% **(d)** *I-V* curve for the ST perovskite top panel composed of four parallel connected 2D material-engineered ST-PSMs $V_{OC}$ = 25.75 V, $I_{SC}$ = 2.08E-1A, $P_{MPP}$ = 3.12 W, FF = 58.24%, PCE = 13.1%.

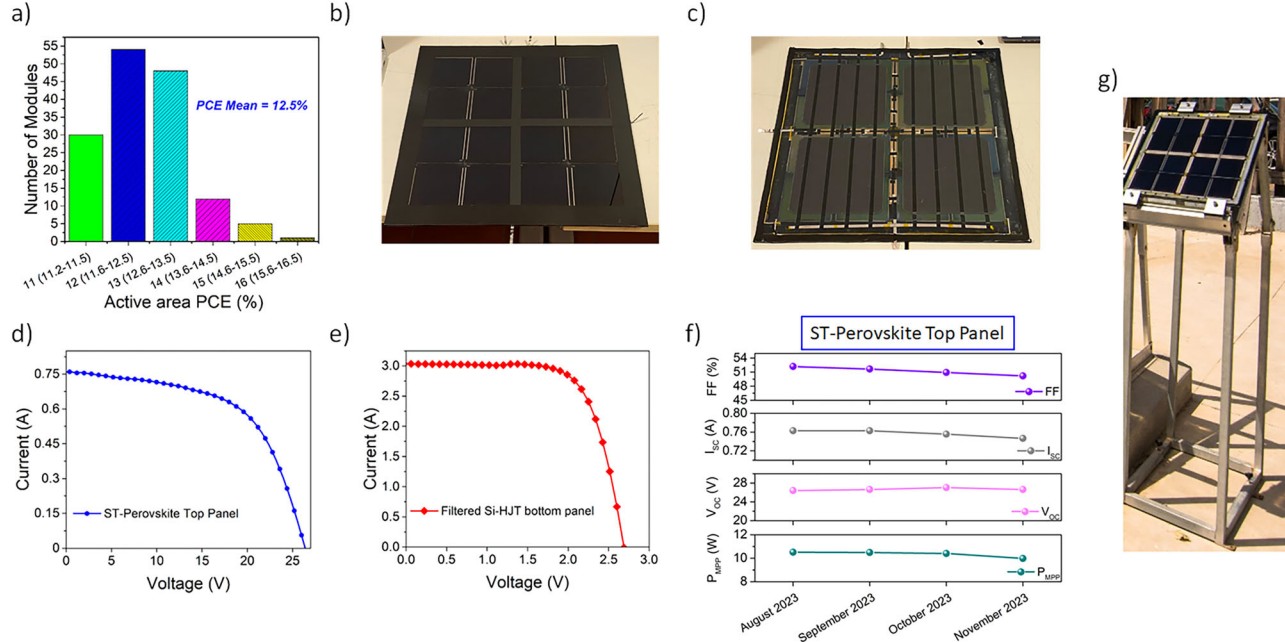

**Fig. 7 | Large-area tandem panel performance and outdoor stability. a** PCE statistics over 150 2D material-engineered ST-PSMs; **(b)** Front and **(c)** back side of the large area tandem panel DEM2. The panel covers an area approximately equal to 0.2 m². **(d)** *I-V* characteristics for ST-perovskite top (circle symbol blue curve) panel $V_{OC}$ = 26.39 V, $I_{SC}$ = 7.62E-1 A, $P_{MPP}$ = 11.51 W, FF = 59.24%, PCE = 12.49% and **(e)** *I-V* characteristics for Si-HJT bottom (diamond symbol red curve) $V_{OC}$ = 2.69 V, $I_{SC}$ = 3.042 A, $P_{MPP}$ = 5.73 W, FF = 70.21%, PCE = 6.95%; **(f)** preliminary outdoor stability of the ST-perovskite top panel above 3 operating months (from August 2023 to November 2023); **(g)** picture of DEM2 mounted in the facilities of HMU in Crete Island in outdoor conditions.

## Methods

### Materials

All the materials were used as received, unless specified otherwise. All the following materials, including titanium(IV) isopropoxide (TTIP), diisopropoxytitanium bis(acetylacetonate) (Ti(AcAc)$_2$), acetyl acetone (AcAc), ethyl acetate (EA), lithium bis(trifluoromethanesulfonyl)imide (Li-TFSI), ethanol (EtOH), isopropanol (IPA), acetone, dimethylformamide (DMF), dimethyl sulfoxide (DMSO), acetonitrile (ACN), tert-butylpyridine (tBP), chlorobenzene (CB), toluene (T), N-methyl-2-pyrrolidone (NMP) and graphite flakes were purchased from Sigma-Aldrich. Mesoporous transparent titania paste (30 NR-D), formamidinium iodide (FAI) and methylammonium bromide (MABr) were purchased from GreatCell Solar. Lead(II) iodide (PbI$_2$), lead(II) bromide (PbBr$_2$) were purchased from TCI, while caesium iodide (CsI) from GmbH. Poly (triarylamine) (PTAA) Medium Mw: 20–75 kDa (SOL2426M) from Solaris Chem. The graphene flakes ink in EtOH (≥99.8%) (concentration of 0.9 mg mL$^{-1}$) was prepared by liquid phase exfoliation of graphite flakes (Sigma–Aldrich) in NMP and exchanged into EtOH (see our previous paper for further details)[38].

**Ti$_3$C$_2$Cl$_2$ MXene synthesis.** Ti$_3$C$_2$Cl$_2$ MXenes were synthesized by etching of Al layers from Ti$_3$AlC$_2$ MAX phase precursor using Lewis acid. In detail, Ti$_3$AlC$_2$ MAX phase powder was mixed with anhydrous salts of ZnCl$_2$, NaCl, and KCl in a molar ratio of 1: 3: 2: 2 under argon atmosphere. The mixture was placed in an alumina crucible covered with a lid and heated to 650 °C for 5 h. After synthesis, the mixture was washed with deionized (DI) water to dissolve and remove the residual salts, and then with HCl to remove elemental Zn formed by the reduction of Zn$^{2+}$. Delamination of multilayer Ti$_3$C$_2$Cl$_2$ MXenes was carried out under Ar atmosphere by intercalation of LiCl salt dissolved in anhydrous polar organic solvents as described in detail in ref. 39. The delaminated suspension was collected as a film by vacuum filtration followed by drying at 80 °C under vacuum.

**Ti$_3$C$_2$Cl$_2$ MXene structural analysis.** The MAX phase precursor and resulted delaminated film were studied by XRD using a D8 Bruker diffractometer (Bruker, Massachusetts, USA) operating under the Bragg-Brentano geometry and equipped with Cu anode providing X-rays with a wavelength of 1.5406 Å. The morphology and chemical composition of the synthesized Ti$_3$C$_2$Cl$_2$ multilayer powder were investigated using a field-emission SEM (ZEISS Gemini 300, Carl Zeiss AG, Oberkochen, Germany) equipped with a Si(Li) detector (Oxford Instruments) for Energy Dispersive Spectroscopy. The combined photoemission/microscopy Omicron setup ESCA-STM was used to carry out XPS and UPS of Ti$_3$C$_2$Cl$_2$ MXenes. For the measurements, the delaminated suspension of Ti$_3$C$_2$Cl$_2$ was drop-casted on ITO substrates. XPS was performed using monochromatic Al-Kα radiation (1486.6 eV) and deconvolution of the high-resolution (HR) XPS spectra was performed in CasaXPS software. UPS measurements were performed using a He I source (21.22 eV). During the measurements, a bias of 9 V was applied between the sample and the analyzer.

### Electro-optical characterizations: SEM, UV-Vis, PL, XRD, IPCE, Transient measurements, I-V characteristics, XPS, UPS, stability tests

SEM images were performed by using a TESCAN MIRA equipment. SEM images are acquired through an in-beam secondary electrons (in beam-SE) detector by using a Schottky field electron emitter providing an electron beam with an energy of 5 Kev and a probe current of 100pA.

The UV–Vis absorption spectra of the perovskite layer were recorded using UV–Vis 2550 spectrophotometer from Shimadzu. The spectra were collected using a scan rate of 480 nm min$^{-1}$ and a resolution of 1 nm. Steady state PL measurements were performed with a commercial apparatus (Arkeo–Cicci Research s.r.l.) composed of a

### Table 3 | Efficiency of the 4T tandem perovskite/bifacial silicon under various irradiance percentage on backside based on ground reflectivity in STC

| 2D material-engineered 4 T tandem panel (DEM2) | No albedo | Albedo | | | |
|---|---|---|---|---|---|
| Irradiance percentage on backside based on ground reflectivity (STC) | 0% | **15%** | **20%** | **25%** | **30%** |
| Efficiency without albedo (%) and PGD in case of albedo (mW cm$^{-2}$) | **19.45** | 21.15 | 22 | 22.82 | **23.66** |

0.3 m focal length spectrograph with a photon counting unit. The substrates were excited by a green (532 nm) laser at 45° of incidence with a circular spot diameter of 1 mm. The optical coupling system is composed of a lens condenser and a long-pass filter. The PL spectra were elaborated by an in-house developed Matlab script. In detail, perovskite emission spectra were fitted by employing the best fitting between single, double, or triple exponential Gaussian line-shape by minimizing the root mean square error.

XRD measurements were collected with a Rigaku SmartLab SE 1D Type diffractometer working in Bragg−Brentano geometry equipped with a Cu Kα source and a D/teX Ultra250 detector. Current-Voltage (I-V) characteristics of masked and encapsulated devices were acquired in air by using a solar simulator (ABET Sun 2000, class A) calibrated at AM1.5 and 100 mW cm$^{-2}$ illumination with a certified reference Si Cell (RERA Solutions RR-1002). Incident power was measured with a Skye SKS 1110 sensor. The class was measured with a BLACK-Comet UV–vis spectrometer. Both reverse and forward I-V scans were performed by using a scan rate of 20 mV s$^{-1}$ and 100 mVs$^{-1}$ for masked small-area PSCs and unmasked large-area ST-PSMs, respectively.

Combined heating+light soaking stress test was performed using a class-B solar simulator (SolarConstant 1200, K.H. Steuernagel Lichttechnik GmbH, Germany). The system employs a xenon arc lamp coupled with an optical AM 1.5G filter to reproduce the standard solar spectrum over a broad spectral range (300–2500 nm). According to manufacturer specifications and IEC 60904-9 classification, the simulator exhibits class-B performance in spectral match, irradiance uniformity, and temporal stability. In particular, the spectral distribution in the near-infrared region shows the characteristic enhancement associated with xenon-based sources, consistent with the class-B spectral mismatch factor. The output irradiance is stabilized and set to 1000 W m$^{-2}$ at the sample plane under ambient laboratory conditions, and the lamp intensity is calibrated before each measurement session using a certified silicon reference cell.

The relatively broad IR output, combined with continuous illumination, results in a device operating temperature representative of realistic stress conditions, while remaining fully compliant with the IEC tolerances for photovoltaic testing.

Illumination intensity dependence of $V_{OC}$ and dark I-V measurements were performed with a modular testing platform (Arkeo – Cicci Research s.r.l.) composed by a white LED array (4200 K) tunable up to 200 mW cm$^{-2}$ of optical power density and a high-speed source meter unit (600,00 samples s$^{-1}$) in a four-wire configuration. A spring contact-based sample holder was used to improve the repeatability of the experiments. TPV measurements were performed in a high perturbation configuration by acquiring the entire $V_{OC}$ rise profile after switching the light intensity from 0 to 1 Sun. Incident Photon to current Conversion Efficiency (IPCE) spectra acquisitions were carried out by means of an Arkeo system (Cicci Research s.r.l.) with a 150 W xenon lamp and a double grating (300 to1400 nm). A Si photodiode was used for incident light calibration prior to the IPCE measurement. Photoelectron spectroscopy measurements were performed in a Thermo-Scientific ESCALAB QXi XPS apparatus in ultra-high vacuum conditions

($10^{-10}$ mbar base pressure). The valence band region of the samples was investigated by UPS using the 21.2 eV radiation of the HeI line and biasing the samples at −5 V, in order to determine both the work function and the position of the valence band edge[40]. The electronic band structure of the perovskites was characterized by XPS using monochromatic 1486.8 eV photons of the Al Kα line and analyzed with the Thermo-Fischer Avantage Software. XPS measurements were performed on small devices ($14 \times 14$ mm) applying the system charge compensation, while the work function was measured without charge compensation, by providing an additional contact to an edge of the device front surface.

### Device fabrication

A preliminary patterning step on the glass/FTO substrate (Pilkington, 8 $\Omega/\square$, 110 $\times$ 110 mm$^2$) is carried out using a raster-scanning Nd:YVO$_4$ laser (pulsed at 80 kHz, average fluence 700 mJ cm$^{-2}$, $\lambda = 350$ nm) to electrically isolate adjacent cells on the same substrate. In the case of modules, this step (P1 process) is used to isolate the individual cells that compose the final device. For both cells and modules, after a deep cleaning sequence, consisting in a triple step of ultra sonic bath with cleaning liquid dissolved in deionized water, acetone and 2-propanol for 10 min each one, a patterned fluorine-doped tin oxide (FTO) coated glasses $2.5 \times 2.5$ cm$^2$ ($9.5 \times 9.5$ cm$^2$ for modules) are spray-pyrolised with a solution of acetylacetone (1 mL), titanium diisopropoxide (1.5 mL) and ethanol (22.5 mL) at 460 °C.

The subsequent step of small area (and large area) device fabrication consists in a thin mesoporous TiO$_2$ (mTiO$_2$) film (-130 nm) deposited by spin coating. The TiO$_2$ paste (Dyesol 30 NRD paste diluted in ethanol 1:6 in wt.) was spun at 3000 rpm for 20 s (2000 rpm for 20 s) and followed by the subsequent sintering at 480 °C for 30 min in air. The graphene doped solutions, cTiO$_2$ + G and mTiO$_2$+G, are obtained by doping with 1 vol% of graphene ink (from IIT) the pristine solution. Then the substrates were transferred into a N$_2$ glove box.

The perovskite films were deposited by combining a one-step deposition with the anti-solvent method. The perovskite layer was deposited in a N$_2$-filled glove box after cooling down the samples to room temperature. The perovskite solution was obtained by mixing FAI, PbI$_2$, PbBr$_2$ and CsI in a mixture of anhydrous DMF/DMSO (4:1 vol/vol) in a proper ratio to obtain Cs$_{0.18}$FA$_{0.82}$Pb(I$_{0.8}$Br$_{0.2}$)$_3$. After 30 min of stirring at room temperature, the perovskite (100 µl for small area devices, 1 ml for large area PSMs) was spin-coated onto the samples, using an anti-solvent method. When using CB as anti-solvent, a two-step program at 1000 rpm for 10 s and 5000 rpm for 30 s was used to deposit the perovskite precursor solution.

During the second step, 150 µl of CB was poured on the spinning substrate 7 s before the end of the program. Immediately after spin coating, the substrates were annealed at 100 °C for 1 h to form a perovskite crystal structure resulting in a compact perovskite layer with a thickness of 450 nm.

Alternatively, in the case of EA anti-solvent a two-step program at 1000 rpm for 10 s and 4000 rpm for 45 s (same ramp used in the case of ST-PSMs) was used for the deposition of the perovskite precursor solution. During the second step, 250 µl of EA (2.5 ml in the case of ST-PSMs) was poured on the spinning substrate 17 s before the end of the program. The deposited perovskite film was annealed at 100 °C for 30 min, in the case of both small area cells and modules. It is important to note that, given its lower boiling point and optimal polarity, ethyl acetate promotes faster crystallization and more efficient solvent removal during film formation, which enables shorter annealing times compared to chlorobenzene under identical thermal conditions [41].

For device structure G/4-FPEAI and G/MX/4-FPEAI the 4-FPEAI was dissolved in the EA anti-solvent in a concentration of 0.13 mg ml$^{-1}$ and shaken prior the use. Once 3D perovskite-absorber was ready, a 3.73 mg ml$^{-1}$ solution of 2-phenylethylammonium iodide (PEAI) in 2-propanol was prepared in glovebox and deposited by spin-coater

atop the perovskite-absorber at 5000 rpm for 30 s (4000 rpm for 30 s for modules). For the incorporation of chlorine-terminated MXenes (MX-Cl) into the perovskite layer, a stock solution of MX-Cl is first prepared by dispersing the desired amount of material in anhydrous acetonitrile (ACN) using 1 h of sonication, followed by vigorous stirring to ensure a well-dispersed suspension. This solution is then added to the perovskite precursor to achieve a final MX-Cl concentration of 0.016 mg mL$^{-1}$. The resulting mixture is deposited onto the substrate using the same spin-coating and annealing parameters as the pristine perovskite formulation. To complete the 2D perovskite layer formation, the samples were annealed at 100 °C for 10 min. Subsequently, the device photo-electrode was covered by a PTAA (10 mg ml$^{-1}$) solution in toluene, doped with tBP 7 µl ml$^{-1}$ and Li-TFSI salt (170 mg ml$^{-1}$ in ACN) 10 µl ml$^{-1}$ and spin coated at 3700 rpm for 30 s (same spin parameters for modules).

At this stage, in the case of ST-PSMs, a second laser process (P2) was carried out to clean the FTO interconnection areas[42]. Finally, a high vacuum chamber ($10^{-6}$ mbar) was used to thermally evaporate the 150 nm Au back electrode in case of opaque device or alternatively 3 nm gold buffer layer in case of semi-transparent (ST) devices. In this latter case, about 100 nm of ITO was deposited onto PTAA layers in a linear magnetron radio frequency (RF) sputtering system (Kenosistek).

Lastly, in the case of ST-PSMs, adjacent cell isolation from the counter-electrode (CE) side was achieved by a third laser ablation (P3 process). The detailed optimization of the P1, P2, P3 laser step together with the description and the optimization of ST-PSM layout are reported in S.I. section S.I. 10.

Regarding the bottom module, commercial c-Si HJT solar cells were provided by 3Sun company, embedding the silver metal grids on both sides. The cell was produced in standard industrial conditions and used without further modifications.

### Tandem panel fabrication

An ultra-clear 4 mm thick tempered glass was used to give robustness to full panel, keeping a good transmission rate. Atop the primary glass, a 100 µm ionomer foil as primary sealer was posed followed by the positioning of the ST-PSMs one next to the other (4 column × 4 lines in DEM2). The parallel electrical connection among the ST-PSMs composing the perovskite top panel was realized employing charge collector tapes from 3M + tabbing ribbons (see S.I. section S.I. 12-a for further details). As the following step, a 400 µm ionomer foil as insulator was positioned atop the backside (ITO side) of the ST-PSMs as an electrical insulator avoiding electrical shunt formation between the perovskite top panel and the silicon bottom module once laminated together. Then, the Si-HJT cells are positioned over the isomer foil and electrically connected by using 6 mm silver charge collector tapes. Subsequently, a 300 µm ionomer foil as primary sealer was deposited atop the device stack while an edge sealer was placed all around the ST-PSM area. (see S.I. section S.I. 12-b for further details). Finally, a 150 µm polymeric transparent foil was posed atop the tandem device stack as a back sheet (from 3 M, WVTR < $6 \times 10^{-5}$ g/m$^2$/day, 90% transparency in the 400−1400 nm range) prior lamination procedure carried out by an industrial hot vacuum laminator (see SI section S.I. 12-c for further details).

### Reporting summary

Further information on research design is available in the Nature Portfolio Reporting Summary linked to this article.

## Data availability

All the needed information to interpret the findings reported herein can be found within the manuscript and its Supplementary Files. Further experimental validation results and the source data used to generate the Figures and Supplementary Figs. are not publicly available for commercial confidentiality reasons but can be made available from the

corresponding author upon request, subject to signing an NDA document. A copy of the latter can be obtained from the corresponding author upon request.

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

## Acknowledgements

The work has been partially supported by European Union's Horizon 2020 research and innovation programme Graphene Core3 under grant agreement number 881603. A.D.C., S.P., and A.A. acknowledge the support of the RdS project 2025-2027 (working agreement with ENEA). A.A. acknowledged financial support under the National Recovery and Resilience Plan (NRRP), Mission 4, Component 2, Investment 1.1, Call for tender No. 104 published on 2.2.2022 by the Italian Ministry of University and Research (MUR), funded by the European Union – NextGenerationEU– Project Title ELDORADO – CUP - 2022K9PFSJ Grant Assignment Decree No. 957 adopted on 30/06/2023 by the Italian Ministry of Ministry of University and Research (MUR).

## Author contributions

AA, SP, and ADC conceived the work. S.P. and A.A. designed, realized and optimized 2D material-based perovskite solar cells and modules by performing the electrical characterizations. A.A and S.P. designed and optimized the tandem panel architecture. S.P., A.A, F.D.G. optimized and performed laser scribe ablation for perovskite module realization. A.P. and F.B. produced and characterized MXenes and Graphene, respectively. F.M. and S.D.G. performed and analyzed UPS and XPS measurements. G.V. and E.K. designed and built the solar farm infrastructure and performed outdoor electrical panel characterizations in situ. E.L. and L.S. laminated the panels. C.C., G.B., and M.F. provided the Si HJT cells. A.D.V., P.A., and M.A.D.M. performed theoretical studies. A.A., S.P., and A.D.C supervised the work. All authors contributed to the discussion of the results and to the writing of the manuscript.

## Competing interests

F.B. is a co-founder and Chief Scientific Officer of BeDimensional S.p.A., a company commercializing graphene-based materials, which supplied the graphene used in this study. L.S. and E.L. are employees of Halocell Europe SRL, a company focused on the commercialization of perovskite solar technologies. G.B., C.C., and M.F. are employees of 3SUN – Enel Green Power (EGP) S.p.A., a company active in the industrial production of silicon heterojunction solar cells. The authors declare no competing non-financial interests. A.A., S.P., G.V., A.P., A.D.V., P.A., M.A.D.M., F.M., S.D.G., F.D.G., E.K., A.D.C. declare no competing interests.
