## [Transparent Peer Review file · Nature Communications]

MXene-driven nanoscale field-effect junction for advanced 4-terminal perovskite/silicon tandem solar panels

Corresponding Author: Professor Antonio Agresti

Version 0:

Reviewer comments:

Reviewer #1

(Remarks to the Author)

The manuscript reports progress in MXene-driven nanoscale graded homojunctions for perovskite top cell in 4-terminal perovskite/silicon tandem solar panels.

The manuscript is technically well written and sound. However, I have one major and several minor remarks that need to be considered before I can recommend the manuscript for publication.

MAJOR remark:

1) l. 303: "monitored in outdoor conditions" must be specified in detail. Were top cells and bottom cells in 4-T tandems in MPP or open-circuit or short-circuit? You need to show the behaviour for the 4 months in Supplement! How did determine data in Fig. 7f? Did you bring tandems indoor and measure them under STC? Fig. 7f shows only period of 3-months (and not 4 months!).

In line 318 you state FF degradation -1.05%/month, which in 4 months end in 4,20% which means a 8% degradation from initial FF value of ~51% (estimated from Fig. 7f, since there is no table of exact values – add a table in Supplement!) In Fig. 7f also Isc degradation is seen, so the total degradation of 7% (over 93% retained after 4 months) is undermined and UNDER QUESTION.

No metastability behaviour of perovskite cells on day-night basis is nor mentioned not discussed, but data should be recorded and such analysis would clarify the stability of the MXene-driven nanoscale graded homojunction perovskite solar cells.

Minor remarks:

1) l. 36: "under albedo conditions" needs to be more specific, e.g. "under 30% ground albedo conditions".

2) l. 36: "retained over 93%" should be corrected according to l. 96 and l. 318 (see below remark l.318)

3) l.53 "significant advantages, including independent MPPT" brings also additional cost that must be added to this too optimistic statement

3) l. 54 I miss reference to 3-terminal architecture of perovskite/silicon tandem cells that also exhibits flexibility to bifaciality

4) l. 57: "typical opaque electrode" - not clear for which cell (top PK or bottom Si)?

5) l. 63: "the scaling-up PCE losses" is not a challenge, but rather minimation of PCE losses upon area up-scaling

6) l. 96: degradation of -1.05%/month * 4 months = -4,20% is this absolute or relative value of FF change?

7) l. 120: comma is missing between ref 28 and 29 (28,29)

8) l. 135: 8 samples in Fig. 2 deserve to have a photo of a glass substrate with the 8 cells in Supplement.

9) l. 137: you show only results in reverse scan, although in Supplement you show both reverse and forward scan. Since there is quite a difference between reverse and forward scan, you should at least mention this difference (not only FF, but even Voc is quite different).

10) l. 312: "optimal distance" deserves an argumentation WHY? As Fig. 7g reveals the ground albedo is very high and the module under test is nor surrounded by neighboring modules or shades in plane-of-array.

Reviewer #2

(Remarks to the Author)

The manuscript concisely reports the steps taken to work on the efficiency of perovskite cells by manipulating both interfaces of the absorber layer. The working principles on how this leads to improved device performance are well backed by a thorough analysis of additional experimental results trying to identify the opto-electronic build-up of the energetics in the layer stack.

Additionally, fabrication of perovskite semi-transparent minimodules, 4T tandem with a Si solar cell and finally also a larger area perovskite multi-minimodule/Si minimodule 4T tandem assembly is demonstrated, with outdoor operating data for few months.

While these results can have high relevance to proof the progress made in this research domain of perovskite photovoltaics, it remains a too fragmented report overall. There's no clear link, with for example a cell-to-module loss analysis, between the advancements seen in the (opaque) perovskite cell developments and the (ST) minimodules. Also, no stability results on cell level are presented to support any early interpretation on the potential degradation mechanisms that can play in the outdoor tested samples.

Moreover, claims repeatedly being made in the manuscript that this work shows scalability of the perovskite and tandem PV technology, even alluding on the low LCoE that can be obtained, are not well placed here. The process is still very lab scale oriented (still spin coating based), highly manual - both on minimodule manufacturing as on tandem assembly - and at sizes still well below m2 dimensions.

Reviewer #3

(Remarks to the Author)

The manuscript titled "MXene-driven nanoscale graded homojunctions for advanced 4-terminal perovskite/silicon tandem solar panels" by Agresti et al. demonstrates and studies chlorine-terminated MXene (Ti₃C₂Cl₂) ETL and PEAI passivated HTL contacts for NIP perovskite solar cells and 4T silicon/perovskite tandems. The authors claim that this architecture creates a graded homojunction within the perovskite bulk that suppresses non-radiative recombination and facilitates carrier extraction for high-efficiency devices. That strategy is scaled to a 19.45% efficient 0.2m² tandem modules and fielded outdoors in Crete for 4 months. The module fabrication steps and performance characterization were well documented with the inclusion of temperature and irradiance dependent data on reasonably sized tandem modules as well as details on module scribing, interconnects, and lamination processes. The scalability of this approach and field data could be of high interest to the field, however, I have some significant concerns limiting the potential impact of this work.

First, I do not find sufficient evidence to support the claim for the formation of a graded homojunction. Second, while the authors clearly advance device demonstration of halogen MXene doping for tuning perovskite interfaces, this concept has been reported before (e.g. interface calculations of Ti₃C₂Cl₂ DOI: 10.1016/j.vacuum.2023.112381; MXene work function tuning for perovskite solar cells DOI: 10.1039/D2NR02799B). Third, the reported tandem module efficiency of 19.45% is not above the efficiency of an average single junction silicon module, even though the tandem pairing resulted in a net 1.05% absolute efficiency increase over the Si HJT cells used in this study and the reported field stability is not at the state-of-the-art of other public reports for perovskite-based tandem modules (<https://pvpact.sandia.gov/results-and-data/>). I have included additional comments below, but unfortunately cannot recommend publication of this manuscript in nature communications in its current form.

Additional comments:

- 1) The authors report UPS and Kelvin probe CPD data on Ti₃C₂Cl₂ MXenes in Fig S6 and additional CPD measurements in Fig S13 and half-stacks in Tab. 2. These experiments appear to be film surface measurements but not a depth profile of an actual perovskite film. As far as I can tell, the perovskite work function and band structure primarily comes from DFT simulations and TiberCAD simulations in Figs S15-S17. A work function change due to doping of a contact layer can induce an electric field at a contact interface, but that is insufficient to claim the formation of a graded homojunction. Is there any experimental evidence to suggest that there are doping changes within the perovskite layer as a result of the new contact strategy that are aiding in the suppression of non-radiative recombination and carrier collection?
- 2) The authors should be careful about making claims of grain sizes based on SEM methods. It has been shown in the literature that these techniques cannot identify true grain size or grain boundaries (DOI: 10.1021/acseenergylett.8b01704, DOI: 10.1016/j.joule.2018.12.011). It is reasonable to say "apparent grains", "perovskite domain", or to describe the "morphology", but this should be taken into consideration when attributing film properties to SEM images.
- 3) The authors claim "Moreover, the hydrophobic nature of MX-Cl further boosts both device efficiency and operational stability, as confirmed by Maximum Power Point (MPP) tracking of the best-performing cell (Fig. S12)." However, figure S12 only shows 180s of maximum power point data. 180s is not sufficient to claim operational stability and does not compare against a control. Furthermore, I do not follow the reasoning as to how the MPP data is related to the hydrophobic nature of MX-Cl.
- 4) In the SI, it is claimed that drawing from Table S5, the 4T tandem pairing resulted in a 2.74% PCE improvement over the silicon HJT module alone. However, it appears that DEM2 is a 19.45% 4T module fabricated with 3SUN. HJT silicon (18.4%; Table S5). Wouldn't this indicate an efficiency improvement of 1.05% over the single junction silicon?
- 5) The authors should provide accelerated stability data (with at a minimum light + heat, preferably 85C or higher) for the 4F-PEAI/MX-Cl architecture vs. control.
- 6) The authors should consider extending the outdoor field test and providing a post-mortem characterization of the fielded module to identify apparent degradation modes of this architecture.

Reviewer #4

(Remarks to the Author)

In this manuscript, the authors demonstrate a synergistic approach of doping a two-dimensional material, MXene Ti₃C₂Cl₂, in the perovskite absorber layer and adding 4-FPEAI to the antisolvent in anticipation of forming a n-p junction, to enhance

the performance of perovskite solar cells. And the method is extended to semi-transparent cells and four-terminal perovskite/silicon tandem devices. Finally PCE of 19.45% was achieved on a 0.2 m² 4T tandem device and field tested for stability. However, I don't see much correlation with the methodology described in this manuscript in the section of the tandem devices, and it doesn't adequately describe the characterization of tandem devices. In addition, there are some problems with the article as follows, so I don't recommend this manuscript for publication in Nature Communications.

Q1) Researchers have already revealed that 4-FPEAI and MXene play important roles in improving the performance of perovskite solar cells. The highlights and novelty of this work should be highlighted.

Q2) In Fig. 2 and Fig.S19, the statistics of the electrical parameters are all obtained from 8 samples, which is not a sufficient amount of data to support the experimental conclusions. For example, in lines 122 "...and exhibiting reduced PCE dispersion ..." and lines 123 "... Improvements highlights enhance device reproducibility ...", 8 samples are clearly not enough to account for these.

Q3) In lines 124, "Furthermore, replacing CB with to EA, the perovskite post-deposition annealing time was reduced by half (30 minutes), effectively halving the energy required for perovskite absorber realization." Is there any experiment or reference that proves it, I don't understand.

Q4) Lines 140 "...with Ti₃C₂Cl₂ MXenes (MX-Cl) by carefully tuning the MXene dopant amount (Fig. S2)." In Fig. S2, what are the units of MX-Cl concentration? The interval between 0.016 and 0.15 is too large, are there conditions for better concentrations in between?

Q5) Lines 146 and 147, "SGP has minimal impact on grain size and surface morphology (Fig. S8), whereas MX-Cl doping, resulted in a noticeable increase in perovskite grain size compared to the pristine layer (Fig. 3a and 3c)." Statistical distribution and comparison of average grain size should be given.

Q6) Lines 162, "XRD analysis (Fig. 3e) reveals ... and minimizing trap states." Enhancement of the XRD peak intensity does not necessarily directly indicate a reduction in defects in the film, and enhancement of the diffraction peaks may reflect increased crystallinity or increased grain size, which corresponds to SEM. Increased grain size results in sharper peaks and increased intensity, but this may be directly related to a reduction in grain boundaries rather than point defects. A reduction in point defects or dislocations may reduce lattice distortion, resulting in increased peak intensity, but changes in defect density need to be confirmed in conjunction with other means such as TEM and SCLC.

Q7) The peak intensity ratios in the XRD pattern in Fig.3e clearly do not correspond to the values in section SI 5, Tab. S1, e.g., the ratio of 4-FPEAI (110)/(310) is clearly greater than 1 in the figure, but 0.92 in the table.

Q8) In lines 179, "...slight blue shift (767 nm vs. 763 nm) for MX-Cl/4-FPEAI films...", the optical band gap of Cs_{0.18}FA_{0.82}Pb_(10.8Br0.2)₃ perovskite was 1.68 eV in Fig.S1, and the band gap is calculated to be 1.62 eV (~760 nm) based on the emission peaks of PL. Does the introduction of 4-FPEAI and MX-Cl change the band gap of perovskite?

Q9) In lines 189, "...the hydrophobic nature of MX-Cl further boosts both device efficiency and operational stability, ..." There's no pristine group here in Fig.S12. And a 3-minute mpp tracking is too short to indicate stability. Contact angle and surface tension calculations are needed to show that MXene doping leads to an increase in the hydrophobicity of perovskite films.

Q10) In lines 206 and 207, "...could be concluded that the combination of 4-FPEAI and MX-Cl doping forms a graded n-p junction along the thickness of the perovskite absorber from ETL to HTL side." The variation of the work function of CPD is not a sufficient proof of the existence of n-p junction, which can be further demonstrated by calculating the energy level structure based on UPS of different films.

Q11) In lines 258, "...confirming the effectiveness of 2D material-engineering approach even for semi-transparent structures." The cell structure changes from opaque to semi-transparent, where the only change seems to be the change of the top electrode from Au to ultrathin Au/ITO. Whereas the effect of MX-Cl/4-FPEAI is significantly diminished (the average increase of 30 mV is diminished to 11 mV), and is it possible that due to the change in work function of electrode that leads to a diminution of the effect of the n-p junction as described in the manuscript.

Version 1:

Reviewer comments:

Reviewer #2

(Remarks to the Author)

The authors have addressed all previous comments and therefore publication of the manuscript is agreed.

Reviewer #3

(Remarks to the Author)

The authors have made a significant effort to revise the manuscript with additional experiment and discussion. While there have been improvements in many areas, I still have serious concerns about the graded homojunction claim. Considering how this is central to the proposed mechanisms in this paper, I cannot support publication this claim included. I'll discuss in more detail below.

Graded homojunction: The authors add additional UPS spectra in Fig S11 and summarize the extracted WF and VBM positions (relative to the WF) in Tab. 2.

Concern 1: UPS is highly surface sensitive (probe depth ~1-2nm), but there is no discussion on how samples were handled or if they ever saw air. These details are important to the interpretation of the results. For example, differences in hydrophobicity of samples surfaces (e.g. with 4-FPEAI treatment) can alter water uptake. In addition, the 4-FPEAI treatment is additive to the surface and therefore the measurement may not be representative of the perovskite band structure.

Concern 2: Doping of a semiconductor inherently moves the fermi level relative the valence band and conduction band of the material. There is no characterization of the material bandgaps provided. In order to interpret where the fermi level lies within the bandgap, one must know two of the three: VBM, conduction band minimum (CBM), and bandgap. The VBM has been measured, but subtle bandgap changes from MX-Cl and/or 4-FPEAI processes could impact the interpretation of the doping of the semiconductor.

Concern 3: Assuming that the bandgap remains constant for all tested samples (REF, MX-Cl, MX-Cl/4-FPEAI, 4-FPEAI), there is no change in the doping level of the perovskite between the REF and MX-Cl samples (0.9eV between the VBM and the Fermi level for both samples). There is no enhancement of n-type character with MX-Cl addition (stated in the discussion), but it is consistent with the MX-Cl dipole that the authors mention. However, that implies that the "homojunction grading" of the band structure arises purely from the 4-FPEAI surface treatment. This would theoretically be true for both the 4-FPEAI and MX-Cl/4-FPEAI samples and the homojunction does not arise from a synergy of MX-Cl and 4-FPEAI.

Concern 4: The CPD and UPS measurements are only measuring surface properties of individual films. None of these methods directly measure the graded depth dependence that is being claimed. It would be more appropriate to characterize the claimed effect with a method such as cross-sectional scanning KPFM, for example DOI: 10.1038/s41560-018-0324-8.

Concern 5: In Tab. 2. the authors provide an error of +/- 0.1eV on the VBM-work function energy offsets. The VBM position relative to the Fermi level is 0.9 +/-0.1 eV, 0.9 +/-0.1eV, and 0.8 +/-0.1eV for the REF, MX-Cl, and MX-Cl/4F-PEAI samples, respectively. Therefore, all of these measurements are within error and there is no discernable difference in doping between these three samples and the graded homojunction conclusion should not be made.

In addition to my above concerns, I do not believe that MPP under cycled light is a sufficient alternative to the combined stressors of light + heat and reiterate my comment that accelerated testing under 85C and 1 sun illumination should be included.

Reviewer #4

(Remarks to the Author)

I have carefully reviewed the revised manuscript and am pleased to inform you that I am satisfied with the revisions. The manuscript has addressed the concerns raised previously, and I believe it is now suitable for publication in Nature Communications. I recommend its acceptance.

Thank you for the opportunity to review this work.

Version 2:

Reviewer comments:

Reviewer #3

(Remarks to the Author)

I appreciate the authors' detailed responses to the concerns that have been raised. The authors included the following statement in their response:

"However, our system does not rely on bulk doping but on two spatially separated induced p or n character:

- MX-Cl creates a dipole and downward vacuum-level shift at the buried perovskite/m-TiO₂ interface, producing localized n-type interfacial character and electron accumulation.
- 4-FPEAI induces a p-type shift at the top perovskite surface, modifying the electronic landscape only at the upper interface."

"This validates the interpretation provided in the main text and supports the formation of a dipole-induced interfacial band bending rather than any bulk-doping-driven shift of the band edges."

I generally agree with the above proposed mechanisms by the authors and believe the data are consistent with the

conclusion that the observed effects are independent surface dipoles generated by the MX-CI at the buried interface and 4-FPEAI at the top surface. The specifics of the 4-FPEAI passivation in these statements are a bit vague, but the accepted mechanisms in the field are i) surface passivation of undercoordinated lead sites, ii) the formation of a 2D/3D heterostructure, and iii) surface dipole passivation.

A graded homojunction and field-effect passivation are not the same mechanism. A graded homojunction implies an internal electric field generated by a doping profile within the material. Field-effect passivation is achieved through the application of an external electric field or fixed charges/dipoles from an adjacent layer. I believe the authors and I are in agreement that a bulk doping mechanism is not supported by the data and therefore field-effect passivation is a more appropriate description of the mechanism.

I would support this manuscript for publication if the claim of a graded homojunction is removed and is instead described as field-effect or surface dipole passivation.

Answers to the Reviewers' comments:

We would like to sincerely thank the reviewers for their thorough and constructive evaluation of our manuscript. We are grateful for their recognition of the technical quality, clarity, and overall structure of our work. We particularly appreciate the positive comments highlighting the scientific soundness of our study, the well-documented fabrication and characterization of the tandem modules, and the relevance of the scalability and outdoor stability data presented.

We are pleased that the reviewers acknowledged the novelty of our synergistic interface engineering approach, combining chlorine-terminated MXene doping and 4-FPEAI passivation, to form a graded homojunction within the perovskite absorber. We also thank them for appreciating the demonstration of high-efficiency four-terminal perovskite/silicon tandem devices, including large-area modules tested under real outdoor conditions.

Below, we provide detailed point-by-point responses to the reviewers' comments.

Reviewer #1 (Remarks to the Author):

The manuscript reports progress in MXene-driven nanoscale graded homojunctions for perovskite top cell in 4-terminal perovskite/silicon tandem solar panels. The manuscript is technically well written and sound. However, I have one major and several minor remarks that need to be considered before I can recommend the manuscript for publication.

MAJOR remark:

- 1) I. 303: "monitored in outdoor conditions" must be specified in detail. Were top cells and bottom cells in 4-T tandems in MPP or open-circuit or short-circuit? You need to show the behaviour for the 4 months in Supplement! How did determine data in Fig. 7f? Did you bring tandems indoor and measure them under STC?**

We thank the reviewer for underlining this point that indeed requires some clarifications. Regarding the prolonged outdoor testing, we decided to test our DEM2 panel at open circuit condition, following the ISOS-O-2 protocol. Thus, the panel was not "continuously monitored" in a strict sense, but was left mounted on the rack in the operative position at open-circuit and measured once per month. The parameters reported in **Fig. 7f** were extracted from the I-V curves, acquired for the perovskite top panel after a stabilization time of 10 minutes, since we observed an increase in the delivered power till reaching a stabilized value after 10 minutes (see the new **Fig. S29a** in SI). Since the outdoor characterizations are typically subjected to the instantaneous variation of the irradiance and the panel temperature, we selected a clear sky day in the first half of the month performing the measurements when the irradiance was supposed to be maximum (at 12:30 am). The irradiance daily curves for Hellenic Mediterranean University site in Heraklion (35.31959; 25.10244) recorded during the day when the outdoor I-V curve were acquired, are reported in **Fig. S29b**. Once acquired the I-V curves, the extracted electrical parameters were normalized in STC (1000 W/m² irradiance and 25°C temperature) considering i) the panel temperature and the irradiance values recorded at the measurement time (at 12:30 am, see values reported in the new **Tab. S10**) ii) both temperature and irradiance dependency of the main electrical parameters reported in **Fig. S28**. What is stated above is now included in **SI section S.I. 14** and following reported:

The prolonged outdoor testing was performed on DEM2 panel at open circuit condition, following the ISOS-O-2 protocol. Thus, the panel was not "continuously monitored" in a strict sense, but was left mounted on the rack in the operative position at open-circuit and measured once per month. The parameters reported in **Fig.7f** were extracted from the I-V curves, acquired for the perovskite top panel after a stabilization time of

10 minutes, since we observed an increase in the delivered power till reaching a stabilized value after 10 minutes (see Fig. S29a). Since the outdoor characterizations are typically subjected to the instantaneous variation of the irradiance and the panel temperature, we selected a clear sky day in the first half of the month performing the measurements when the irradiance was supposed to be maximum (at 12:30 am). The irradiance daily curves for Hellenic Mediterranean University site in Heraklion (35.31959; 25.10244) recorded during the day when the outdoor I-V curve were acquired, are reported in Fig. S29b. Once acquired the I-V curves, the extracted electrical parameters were normalized in STC (1000 W/m² irradiance and 25°C temperature) considering i) the panel temperature and the irradiance values recorded at the measurement time (at 12:30 am, see values reported in Tab. S10) ii) both temperature and irradiance dependency of the main electrical parameters reported in Fig. S28.

Fig. S29: a) Normalized power at maximum power point (PMPP) stabilization prior the acquisition of the monthly I-V curves for the DEM2 left at V_{oc} in outdoor conditions; b) daily irradiance for the days selected during the outdoor test for the acquisition of the I-V curves.

2) Fig. 7f shows only period of 3-months (and not 4 months!). In line 318 you state FF degradation - 1.05%/month, which in 4 months end in 4,20% which means a 8% degradation from initial FF value of ~51% (estimated from Fig. 7f, since there is no table of exact values x 2013; add a table in Supplement!) In Fig. 7f also I_{sc} degradation is seen, so the total degradation of 7% (over 93% retained after 4 months) is undermined and UNDER QUESTION.

We thank the reviewer for this careful observation. The reviewer is right, the stability test shown in Fig. 7f covers a period of three months rather than four. We have corrected the text accordingly.

Thanks to the suggestion of the reviewer, with the aim to ensure full data transparency and to avoid potential ambiguities in data interpretation, in the new version of the SI we have included a table named Tab. S10 reporting the electrical parameters (for the perovskite top panel of DEM2) related to Fig. 7f, allowing a quantitative assessment of the parameter degradation over the testing period. Considering a three-months period, the extracted parameter values (Tab. S10) indicate that FF decreases from 52.2% to 50.2% (\approx -3.83% relative degradation over three months), I_{sc} decreases from 0.76 A to 0.74 A (\approx -2.6% relative degradation), while V_{oc} remains largely stable (\approx +0.8%), resulting in an overall panel power decreases from 10.5 W to 10.0 W (\approx -4.8% relative degradation). Thus, the perovskite top panel retained approximately 94–95% of its initial performance over the three-months period. Notably, the observed modest degradation is consistent with the metastability behaviour (now reported in SI, section S.I. 9) of graded homojunction perovskite cells subjected to accelerated light-cycling tests, showing limited performance loss (T_{91} =600 hours at MPP) over time. We added a discussion in SI, section S.I. 14, that now sounds as follows reported:

Tab. S10: Photovoltaic parameters of the perovskite top panel (DEM2) during the three-month outdoor stability test and quantitative evaluation of performance degradation over time. The electrical data were obtained by acquiring the I-V characteristics in outdoor condition in selected days (one per each stress month; panel temperature and irradiance values at the I-V curve acquisition time are reported in the first two columns) at 12:30 am and subsequently normalized in STC (1000 W/m² irradiance and 25°C temperature).

	Panel T (°C)	Irradiance (W/m ²)	V _{oc} (V)	I _{sc} (A)	FF(%)	Power(W)
8 th August 2023	61.3	986	26.4	0.76	52.2	10.5
9 th September 2023	53.9	829	26.6	0.76	51.6	10.5
9 th October 2023	50.4	782	27.1	0.75	50.9	10.4
9 th November 2023	43.5	608	26.6	0.74	50.2	10.0
Total degradation over three months			0.76%	-2.63	-3.83%	-4.76%

Tab. S10 summarizes the photovoltaic parameters of the perovskite top panel (DEM2) during outdoor testing from August to November 2023. The results show a gradual decrease in FF (~-3.1%) and I_{sc} (~-2.6%) over the three-month period, while V_{oc} remains nearly constant. The overall power output decreases by ~-4.8%, corresponding to a retained performance of about 95% of the initial value.

3) No metastability behaviour of perovskite cells on day-night basis is nor mentioned not discussed, but data should be recorded and such analysis would clarify the stability of the MXene-driven nanoscale graded homojunction perovskite solar cells.

We thank the reviewer for his insightful suggestion that pushed us to add the analysis of the metastable behavior of our perovskite semi-transparent top cell. With the aim of investigating the metastability of our graded homojunction cell, we performed light-cycling tests now inserted in SI as follows reported.

Indoor photo-stability under MPP tracking of semi-transparent encapsulated cells upon exposure to cycled light. As recently discussed in literature, the most effective way to perform a reliable stability analysis consists in light cycling test at constant temperature, representing a good trade-off between simplicity of execution in lab conditions while taking into account phenomena observed only in case of outdoor testing (that cannot be considered in case of constant light experiments).¹⁵ Indeed, transient behavior occurs in PSCs as a result of the coexistence of several dynamics with characteristic times spanning from time scales of seconds to hours. Thus, day–night cycling of devices can consider such slow dynamics as they occur on similar timescales. Following this idea, we performed a light–dark cycling test as recommended by the ISOS-LC-1 protocol, exposing the cells to simulated sunlight turned on and off with cycle periods of 24 h and duty cycles (light:dark) of 1:2, mimic an accelerated diurnal sun cycle (the solar cell is maintained at ambient conditions while the temperature is monitored and RH is maintained at 30%).¹⁶ The recorded maximum power (P_{MPP}) is reported in **Fig. S7e** for both G/EA and fully optimized G/MX/4-FPEAI semi-transparent cells, normalized at the initial value. The combined stabilization effect of the 4-FPEAI based SGP together with the presence of MX-Cl results in a T₉₁=600 hours vs a T₈₀=574 hours of the untreated device, demonstrating the superior stability (T₉₁ 6.5 times larger) of the device employing the proposed graded homojunction approach. Despite more in depth analyses are required to elucidate the role of MX-Cl and 4-FPEAI SGP, we can confidently ascribe the improved stability to a synergistic effect of the two treatments. On one side, the addition of MX-Cl resulted in a perovskite film with improved crystallinity eventually reducing lattice distortion, lowering defect density and minimizing trap states, as demonstrated by XRD, transient PL and SEM characterizations. This, combined with the enlarged perovskite domains and mitigated grain boundary-induced trap states, makes the trapped charge driven degradation less effective,¹⁷ resulting in an overall enlargement of device lifetime. We cannot

exclude a side role of the Cl in reducing the tensile strain within the perovskite layer, recently pointed out as one of the main degradation channel when light-dark cycling is applied.¹⁸ On the other side, the phenylalkylammonium compounds as 4-FPEAI has been proven to stabilize the perovskite through suppression of iodide ion migration, likely responsible for metastable perovskite behavior under light cycling observed for the G/EA device, while passivating the surface defects.¹⁹ In addition, the 4-FPEAI SGP strategy has been already proven to improve the long-term operational stability of the encapsulated devices under the continuous illumination in ambient condition.¹² Thus, the superior light cycling stability showed by the graded homojunction cell can be confidently ascribed to a combined stabilizing effect from 4-FPEAI and MX-Cl, definitively reducing the device metastable behavior.

Fig. S22: a) J-V curve in forward and reverse scan for the best-performing 2D material-engineered ST-PSC (G/MX/4-FPEAI structure); b) IPCE spectra with the integrated photocurrent density (Integrated J_{sc}) related to the best efficient optimized cell (G/MX/4-FPEAI) for both opaque and ST proposed device structure. c) Indoor photo-stability under MPP tracking of G/EA and G/MX/4-FPEAI semi-transparent encapsulated cells upon exposure to cycled light (ISOS-LC-1).

- (12) Yan, N.; Gao, Y.; Yang, J.; Fang, Z.; Feng, J.; Wu, X.; Chen, T.; Liu, S. F. Wide-Bandgap Perovskite Solar Cell Using a Fluoride-Assisted Surface Gradient Passivation Strategy. *Angew. Chem. Int. Ed.* **2023**, *62*, e202216668. <https://doi.org/10.1002/anie.202216668>.
- (15) Khenkin, M.; Köbler, H.; Remec, M.; Roy, R.; Erdil, U.; Li, J.; Phung, N.; Adwan, G.; Paramasivam, G.; Emery, Q.; et al. Light Cycling as a Key to Understanding the Outdoor Behaviour of Perovskite Solar Cells. *Energy Environ. Sci.* **2023**, *17* (2), 602–610. <https://doi.org/10.1039/d3ee03508e>.
- (16) Khenkin, M. V.; Katz, E. A.; Abate, A.; Bardizza, G.; Berry, J. J.; Brabec, C.; Brunetti, F.; Bulović, V.; Burlingame, Q.; Di Carlo, A.; et al. Consensus Statement for Stability Assessment and Reporting for Perovskite Photovoltaics Based on ISOS Procedures. *Nat. Energy* **2020**, *5* (1), 35–49. <https://doi.org/10.1038/s41560-019-0529-5>.
- (17) Ahn, N.; Kwak, K.; Jang, M. S.; Yoon, H.; Lee, B. Y.; Lee, J.; Pikhitsa, P. V.; Byun, J.; Choi, M. Trapped Charge-Driven Degradation of Perovskite Solar Cells. *Nat. Commun.* **2016**, *8* (May), 1–9. <https://doi.org/10.1038/ncomms13422>.
- (18) Shen, Y.; Zhang, T.; Xu, G.; Steele, J. A.; Chen, X.; Chen, W.; Zheng, G.; Li, J.; Guo, B.; Yang, H.; et al. Strain Regulation Retards Natural Operation Decay of Perovskite Solar Cells. *Nature* **2024**, *635* (8040), 882–889. <https://doi.org/10.1038/s41586-024-08161-x>.
- (19) Guo, Y.; Aperi, S.; Li, N.; Chen, M.; Yin, C.; Yuan, Z.; Gao, F.; Xie, F.; Brocks, G.; Tao, S.; et al. Phenylalkylammonium Passivation Enables Perovskite Light Emitting Diodes with Record High-Radiance Operational Lifetime: The Chain Length Matters. *Nat. Commun.* **2021**, *12* (1). <https://doi.org/10.1038/s41467-021-20970-6>.

Moreover, we modified the main text as follows:

“Moreover, the combination among the MX-Cl addition and the 4-FPEAI SGP strategy not only translated into superior device performance but also conferred superior robustness to the perovskite film against light-driven degradation, as demonstrated by prolonged light cycling test (Fig. S22c).”

Minor remarks:

- 1) I. 36: "under albedo conditions" needs to be more specific, e.g. "under 30% ground albedo conditions".

We appreciate the reviewer's comment regarding the need for clarity in the phrase "under albedo conditions." We agree that specifying the exact albedo value improves the precision and readability of the manuscript.

Accordingly, we have revised the text to read "under 30% ground albedo conditions" wherever applicable. This change has been implemented throughout the manuscript to ensure consistency and to accurately reflect the simulation setup used in our study.

2) *l. 36: "retained over 93%" should be corrected according to l. 96 and l. 318 (see below remark l.318)*

Thank you for your observation. We acknowledge the inconsistency between the statement in line 36 and the more precise data provided in lines 96 and 318. We have now revised the sentence in the abstract to reflect the exact performance metric, specifying that the perovskite top panel of DEM2 retained over 95% of its initial P_{MPP} after three months at open circuit, with a slow FF degradation of 1.27% per month. This ensures consistency and accuracy throughout the manuscript.

We change the sentence that now sounds as: "The tandem panel, installed in Crete, retained over 95% of its initial P_{MPP} after three months."

3) *l.53 "significant advantages, including independent MPPT" brings also additional cost that must be added to this too optimistic statement*

We appreciate the reviewer's valuable comment. We agree that the implementation of independent MPPT in 4-terminal architecture entails additional costs related to the required power electronics and system complexity. However, these added costs are often justified by the performance benefits, especially in non-uniform irradiance conditions or bifacial configurations, where independent MPPT allows for optimized energy harvesting from each sub-cell.

To acknowledge this important aspect, we have revised the manuscript by adding in the main text the following sentence, to clarify that while independent MPPT offers significant operational advantages, it also introduces added costs at the system level, which must be considered in techno-economic evaluations:

"and compatibility with bifacial Si cells, which can capture reflected sunlight to enhance energy yield, although independent MPPT adds system-level cost.^{6,7}

(6) Robles-Algarín, C., Olivero-Ortíz, V., & Restrepo-Leal, D. (2022). *Techno-Economic Analysis of MPPT and PWM Controllers Performance in Off-Grid PV Systems*. *International Journal of Energy Economics and Policy*, 12(6), 370–376.

(7) Faranda, R. S., Leva, S., & Maugeri, V. (2008). *MPPT Techniques for PV Systems: Energetic and Cost Comparison*.

4) *l. 54 I miss reference to 3-terminal architecture of perovskite/silicon tandem cells that also exhibits flexibility to bifaciality*

Thanks for your observation. We agree that the 3-terminal perovskite/silicon tandem architecture also allows for bifacial operation and represents a promising configuration. However, the partial electrical coupling between the subcells still requires precise current balancing and complex interconnection schemes, which can hinder independent energy yield optimization and large-scale integration compared to fully decoupled 4-terminal configurations. We have now included a reference and a brief discussion on this architecture at the relevant point in the manuscript (previous line 54) to provide a more complete overview of tandem options.

"Additionally, 3-terminal architectures have also been proposed as a compromise solution, combining reduced complexity with bifacial compatibility,⁸ despite their implementation still requires careful current matching and interconnection design, which may limit scalability compared to 4-terminal configurations."

(8) Gota, F.; Langenhorst, M.; Schmager, R.; Lehr, J.; Paetzold, U. W. Energy Yield Advantages of Three-Terminal Perovskite-Silicon Tandem Photovoltaics. *Joule* 2020, 4 (11), 2387–2403. <https://doi.org/10.1016/j.joule.2020.08.021>.

5) *l. 57: "typical opaque electrode" - not clear for which cell (top PK or bottom Si)?*

Thank you for your comment. We acknowledge that the sentence “typical opaque electrode” could be ambiguous. We confirm that it refers specifically to the top perovskite solar cell, which typically uses a reflective metal back electrode (e.g., Au or Ag) in its standard opaque configuration. To avoid confusion, we have revised the sentence to make this clear in the updated manuscript. The new sentence is now:

“a certain loss in PCE is inevitable when the typical opaque metal electrode used in top PSCs is replaced by a semi-transparent top electrode.”

6) I. 63: “the scaling-up PCE losses” is not a challenge, but rather minimation of PCE losses upon area up-scaling

We thank the reviewer for this valuable clarification. We agree that the correct formulation should emphasize the minimization of PCE losses upon area upscaling as the actual challenge, rather than describing the losses themselves as the challenge. We have accordingly revised the sentence in the manuscript to better reflect this point. The modified new sentence is:

“As a second challenge, the minimization of PCE losses when moving from small area ST-PSCs to semi-transparent perovskite solar modules (ST-PSM) remains a key target for the practical realization of efficient tandem devices.”

Moreover, with the aim of providing further insight into the PCE losses in the scaling from opaque small area cells to semi-transparent large area modules, we added a discussion in the SI section S.I: 11 that now sounds as follows reported:

Cell-to-module performance gap analysis.

To clarify the link between small-area opaque devices and the performance of large-area semi-transparent (ST) modules, we carried out a loss analysis identifying the main contributing factors. The best-performing opaque device (G/MX/4-FPEAI) exhibited a PCE of 22.12%, while the corresponding 60 cm² ST module reached 16.2%, resulting in an absolute performance drop of ~5.96 percentage points.

This gap can be attributed to three main sources:

(i) Optical losses due to the absence of the reflective metal electrode in ST configurations, which reduces photon recycling and light harvesting. To quantify the effect of transparency on photovoltaic performance, we compared our optimized opaque device (G/MX/4-FPEAI) with its semi-transparent (ST) counterpart featuring an Au/ITO top electrode. The opaque device exhibited a PCE of 22.12%, while the ST cell reached 18.3%, under identical illumination and processing conditions (see **Tab. S7**). This ~17.2 % relative efficiency drop (~3.82 percentage points absolute) can be attributed primarily to optical losses due to the removal of the reflective metal electrode, which otherwise enhances internal light trapping and photon recycling. With the aim to quantify the impact of the ST electrode onto the device photocurrent, we performed IPCE measurements on both the best performing opaque and semi-transparent cell, engineered with graded homojunction strategy. By computing the integrated current density, we can conclude that the replacement of opaque electrode with a ST one (gold/ITO) contributed to a J_{sc} loss of ~2.0 mA/cm². This is in line with what observed by De Wolf, S. and co-workers,^{20,21} which employed optical modeling (Transfer Matrix Method (TMM)) of the device stack to predict a 10–12% of incident photons loss due to front-surface reflection and incomplete absorption in the perovskite because of the reduced optical path length. These factors jointly reduce the short-circuit current density (J_{sc}) by ~ 1.8–2.2 mA/cm². In our case the reflection at the glass substrate is the same for both the ST and opaque devices, concluding that the observed J_{sc} drop can be attributed to the absence of photon recycling in the 580-730 nm region of the visible spectra.

(ii) Electrical losses from the transparent electrode stack (Au/ITO), including increased sheet resistance and contact resistance, resulting in an FF drop of ~6.32%. Nonetheless, the retained high V_{OC} (~1.22 V) and FF (~75%) in the ST cell confirm that our band alignment strategy remains effective under reduced optical confinement.

(iii) Scaling-related effects, including increased surface non-uniformity, edge shunting, and series resistance due to larger interconnection paths, contributing to a further loss in FF of -7.5% moving from the ST-cell to the ST-module. FF loss is the main responsible for the observed drop in PCE moving from ST-cell to ST-modules, since V_{OC} and J_{SC} for each large area cells (LAC, with an active area of 2.5 cm²) composing the module did not undergo a significant drop (V_{OC_LAC} ~ 1.25 V and J_{SC_LAC} ~ 20.8 mA/cm², see **Tab. S7**)

Despite these inevitable losses, the use of a vertically graded homojunction via Cl-MXene doping and 4-FPEAI passivation proved critical in maintaining high V_{OC} (1.226 V) and FF (67-69%) at the module level. This highlights the effectiveness of the interface and band alignment engineering in mitigating efficiency degradation upon scaling and semi-transparency.

Tab. S7:Electrical parameters for best performing opaque cell, ST cell and ST module.

Device Type	Area (cm ²)	PCE (%)	V_{OC} (V)	J_{SC} (mA/cm ²)	FF (%)
Opaque cell (G/MX/4-FPEAI)	0.1	22.12	1.21	22.5	81.82
ST cell (Au/ITO)	0.1	18.3	1.22	20.5	75.5
	Area (cm ²)	PCE (%)	V_{OC} (V)	I_{sc} (mA)	FF (%)
ST module	60.0	16.16	26.98	52.7	67.9

And the following statement in the main text:

“while strongly limiting the PCE drop (<13%) usually observed in scaling the device size (S.I. **section S.I. 11** and **Tab. S7** for detailed discussion).”

7) *I. 96: degradation of -1.05%/month * 4 months = -4,20% is this absolute or relative value of FF change?*

We thank the reviewer for this pertinent observation. The reported fill factor degradation rate of 1.27% per month refers to a relative decrease respect to the initial FF value. Moreover, as we point out in the answer to the second question in the major remarks, we underline that the stability test shown in **Fig. 7f** covers a period of three months rather than four, thus the total degradation for FF is -3.8%. We have revised the text to clarify this point accordingly. The new sentence in the main text now sounds as:

“primarily showing a slow fill factor (FF) degradation of 1.27% per month, expressed as a relative decrease respect to the initial FF.”

This

8) *I. 120: comma is missing between ref 28 and 29 (28,29)*

Thank you for pointing this out. We have corrected the citation by adding the missing comma between references 28 and 29, which now correctly appears as (31, 32) in the revised manuscript.

9) *I. 135: 8 samples in Fig. 2 deserve to have a photo of a glass substrate with the 8 cells in Supplement.*

Thank you for the suggestion. We agree that including a photograph of the substrate containing the tested cells enhances the clarity of the experimental section. We would like to clarify that in the new version of the manuscript we extended the device statistics to 12 samples fabricated on three glass substrates, each hosting 4 perovskite solar cells. As an example, we have now added a photograph of one of these substrates to the Supporting Information shown as panel a) in Fig. S2. This integration provides visual context for the sample layout used in the statistical analysis.

Fig. S2: a) Photograph of one of the two glass substrates used for device fabrication, each hosting four small-area perovskite solar cells. A total of 12 devices were fabricated for the statistical evaluation of photovoltaic parameters (Fig. 2,) while 8 devices were used for the optimization of the MXene-Cl doping concentration, b) PCE of PSCs realized by varying the amount of MXene-Cl added to 1 ml of perovskite precursor solution.

10) I. 137: you show only results in reverse scan, although in Supplement you show both reverse and forward scan. Since there is quite a difference between reverse and forward scan, you should at least mention this difference (not only FF, but even V_{oc} is quite different).

We thank the reviewer for the valuable observation. As correctly pointed out, in the main manuscript we reported only the reverse scan results for the opaque devices shown in Fig. 2. However, we agree that the difference between forward and reverse scans should be explicitly addressed, especially for key parameters such as V_{oc} and FF. To this end, we now clarified in the revised manuscript that forward and reverse J–V characteristics for the opaque devices are provided in Fig. S7 of the Supporting Information by adding the following sentence in the caption of the Fig. 2.

“Forward and reverse scan J–V curves for all four opaque device architectures are provided in the Supporting Information (Fig. S7 a), b), c), d)). The minimized hysteresis in the G/MX/4-FPEAI device highlights the improved stability and reduced interfacial recombination enabled by the device engineering via graded homojunction strategy.”

These measurements confirm the presence of scan direction-dependent hysteresis in the reference devices, which is progressively reduced in the engineered devices thanks to MXene doping and surface gradient passivation (SGP). We added in the SI the following text:

From Figs. S7 a), b), c), d), we extracted the forward and reverse scan values of V_{oc} , FF, J_{sc} and PCE for each of the four opaque PSC structures. As shown by the data, the difference in open-circuit voltage between reverse and forward scan (ΔV_{oc}) ranges from 0.04 V in the G/CB reference device down to 0.002 V in the optimized G/MX/4-FPEAI device. Similarly, the fill factor (ΔFF) decreases from 6.3% in G/CB to 1.4% in G/MX/4-FPEAI. This clearly demonstrates that the combined MXene and 4-FPEAI strategy significantly reduces hysteresis, in terms of PCE between forward and reverse scan directions. The ΔPCE decreases from 1.94% in G/CB devices to just 0.4% in G/MX/4-FPEAI.

This trend supports the conclusion that our dual strategy (SGP + MX-Cl) effectively mitigates interfacial recombination and ionic migration, leading to more stable and symmetric device performance under J–V scanning.

11) I. 312 “optimal distance” deserves an argumentation WHY? As Fig. 7g reveals the ground albedo is very high and the module under test is not surrounded by neighboring modules or shades in plane-of-array.

We acknowledge the reviewer’s comment and agree that the term “optimal” may be misleading in this context. To avoid confusion, we have rephrased the sentence as follows:

“Outdoor measurements performed using a metallic mounting base were carried out with the module installed at a fixed inclination of 30° and a height of approximately 100 cm above ground (see Fig. 7g).”

The choice of these parameters follows internationally recognized practices for outdoor photovoltaic performance measurements. In our setup, the module under test was mounted at ~1 m above ground and tilted by 30°, in accordance with the procedures described in IEC 61853-2¹ and IEC 61215². These standards provide representative mounting conditions but do not prescribe a unique “optimal” distance, leaving the configuration to be adapted to the specific experimental environment (see also: NREL Best Practices Handbook for the Collection and Use of Solar Resource Data, 2017;³ IEA-PVPS Task 13 Report on Outdoor Performance Testing, 2020⁴). The mounting height of ~1 m was selected to ensure a representative exposure to ground albedo while still avoiding possible shading effects from nearby structures, which are present in the experimental field even if only partially visible in the Fig. 7g. Furthermore, the inclination of 30° is consistent with the optimal yearly tilt for Heraklion, Crete (latitude ≈ 35°N), where the optimum fixed-tilt angle is generally close to the site latitude. Thus, the chosen configuration not only aligns with international testing practice but also provides realistic operating conditions for the geographical location and device architecture under study.

1. **IEC 61853-2:2016.** Photovoltaic (PV) Module Performance Testing and Energy Rating – Part 2: Spectral Response, Incidence Angle and Module Operating Temperature Measurements; International Electrotechnical Commission, Geneva, Switzerland, 2016.
2. **IEC 61215-1:2021.** Terrestrial Photovoltaic (PV) Modules – Design Qualification and Type Approval – Part 1: Test Requirements; International Electrotechnical Commission, Geneva, Switzerland, 2021.
3. **Sengupta, M.; Habte, A.; Lopez, A.; Gueymard, C.; Wilbert, S.** *Best Practices Handbook for the Collection and Use of Solar Resource Data for Solar Energy Applications*; NREL/TP-5D00-78489; National Renewable Energy Laboratory: Golden, CO, 2017.
4. **IEA-PVPS Task 13.** *Outdoor Performance Analysis of Photovoltaic Systems: Report IEA-PVPS T13-20:2020*; International Energy Agency Photovoltaic Power Systems Programme, 2020.

Reviewer #2 (Remarks to the Author):

The manuscript concisely reports the steps taken to work on the efficiency of perovskite cells by manipulating both interfaces of the absorber layer. The working principles on how this leads to improved device performance are well backed by a thorough analysis of additional experimental results trying to identify the opto-electronic build-up of the energetics in the layer stack.

Additionally, fabrication of perovskite semi-transparent modules, 4T tandem with a Si solar cell and finally also a larger area perovskite multi-module/Si module 4T tandem assembly is demonstrated, with outdoor operating data for few months.

1. While these results can have high relevance to proof the progress made in this research domain of perovskite photovoltaics, it remains a too fragmented report overall. There's no clear link, with for example a cell-to-module loss analysis, between the advancements seen in the (opaque) perovskite cell developments and the (ST) minimodules.

We thank the Reviewer for these valuable observations. We partially agree with the Reviewer that the manuscript may appear somewhat fragmented at first glance, due to the broad scope of the study. However, we respectfully believe that this work represents a comprehensive and methodologically coherent progression, from the optimization of opaque single-junction devices, through the development of semi-transparent architectures, to the fabrication of large-area modules fully integrated in a 4-terminal tandem panel. This stepwise approach reflects the practical route required to translate laboratory-scale advances into scalable and commercially relevant photovoltaic technologies. Furthermore, we believe that the strength of our approach lies in the careful co-optimization of materials and processing techniques across multiple device architectures. By tailoring interfacial properties and energy alignment at the material level, first in small-area opaque devices and then in semi-transparent configurations, we were able to ensure compatibility and reproducibility at the module and panel scale. This integrated development strategy is essential to bridge the gap between material innovation and scalable device implementation in the perovskite/Si tandem field. Nevertheless, we agree that clearly establishing the link between the opaque cell developments and the final performance of the semi-transparent (ST) modules is crucial to better frame the progress achieved. To address this, we have now included in the SI of the revised manuscript a more explicit cell-to-module loss analysis. This discussion outlines the main factors contributing to the performance gap between lab-scale opaque devices and ST modules, including: i) the expected optical losses due to the absence of the back metal reflector, ii) the impact of the transparent electrode stack on parasitic absorption and series resistance, and iii) the influence of scaling-related non-uniformities in film morphology and interface quality.

We further highlight how the graded homojunction architecture, developed and validated at the cell level, contributes to maintaining high open-circuit voltage and fill factor even in the ST configuration, thereby mitigating the typical losses associated with transparency and upscaling.

We added the following comments and **Fig. S22 b)** in the SI in Section S.I. 9

Fig. S22: a) J-V curve in forward and reverse scan for the best-performing 2D material-engineered ST-PSC (G/MX/4-FPEAI structure); b) IPCE spectra with the integrated photocurrent density (Integrated J_{sc}) related to the best efficient optimized cell (G/MX/4-FPEAI) for both opaque and ST proposed device structure. c) Indoor photo-stability under MPP tracking of G/EA and G/MX/4-FPEAI semi-transparent encapsulated cells upon exposure to cycled light (ISOS-LC-1).

External quantum efficiency (IPCE) spectra of opaque and semi-transparent perovskite solar cells. The opaque device exhibits a higher IPCE across the entire spectral range, particularly between 580 and 730 nm, indicating enhanced photon harvesting due to increased optical path length and efficient photon recycling. In contrast, the semi-transparent cell, incorporating a gold/ITO rear electrode, shows a reduced IPCE response in this region, consistent with optical losses from the transparent electrode and limited back reflection. This reduction contributes to J_{sc} loss of approximately 2.0 mA/cm^2 , in case of the optimized G/MX/4-FPEAI device structure.

And the following discussion in section S.I. 11

Cell-to-module performance gap analysis.

To clarify the link between small-area opaque devices and the performance of large-area semi-transparent (ST) modules, we carried out a loss analysis identifying the main contributing factors. The best-performing opaque device (G/MX/4-FPEAI) exhibited a PCE of 22.12%, while the corresponding 60 cm² ST module reached 16.2%, resulting in an absolute performance drop of ~5.96 percentage points.

This gap can be attributed to three main sources:

(i) Optical losses due to the absence of the reflective metal electrode in ST configurations, which reduces photon recycling and light harvesting. To quantify the effect of ST electrode on photovoltaic performance, we compared our optimized opaque device (G/MX/4-FPEAI) with its ST counterpart featuring an Au/ITO top electrode. The opaque device exhibited a PCE of 22.12%, while the ST cell reached 18.3%, under identical illumination and processing conditions (see **Tab. S7**). This ~17.2 % relative efficiency drop (~3.82 absolute percentage points) can be attributed primarily to optical losses due to the removal of the reflective metal electrode, which otherwise enhances internal light trapping and photon recycling. With the aim to quantify the impact of the ST electrode onto the device photocurrent, we performed IPCE measurements on both the best performing opaque and semi-transparent cell, engineered with graded homojunction strategy. By computing the integrated current density, we can conclude that the replacement of opaque electrode with a ST one (gold/ITO) contributed to a J_{sc} loss of ~2.0 mA/cm². This is in line with what observed by De Wolf, S. and co-workers,^{20,21} which employed optical modeling (Transfer Matrix Method (TMM)) of the device stack to predict a 10–12% of incident photons loss due to front-surface reflection and incomplete absorption in the perovskite because of the reduced optical path length. These factors jointly reduce the short-circuit current density (J_{sc}) by ~ 1.8–2.2 mA/cm². In our case the reflection at the glass substrate is the same for both the ST and opaque devices, concluding that the observed J_{sc} drop can be attributed to the absence of photon recycling in the 580-730 nm region of the visible spectra.

(ii) Electrical losses from the transparent electrode stack (Au/ITO), including increased sheet resistance and contact resistance, resulting in an FF drop of ~6.32%. Nonetheless, the retained high V_{oc} (~1.22 V) and FF (~75%) in the ST cell confirm that our band alignment strategy remains effective under reduced optical confinement.

(iii) Scaling-related effects, including increased surface non-uniformity, edge shunting, and series resistance due to larger interconnection paths, contributing to a further loss in FF of -7.5% moving from the ST-cell to the ST-module. FF loss is the main responsible for the observed drop in PCE moving from ST-cell to ST-modules, since V_{oc} and J_{sc} for each large area cells (LAC, with an active area of 2.5 cm²) composing the module did not undergo a significant drop (V_{oc_LAC} ~ 1.25 V and J_{sc_LAC} ~ 20.8 mA/cm², see **Tab. S7**)

Despite these inevitable losses, the use of a vertically graded homojunction via Cl-MXene doping and 4-FPEAI passivation proved critical in maintaining high V_{oc} (1.226 V) and FF (67-69%) at the module level. This highlights the effectiveness of the interface and band alignment engineering in mitigating efficiency degradation upon scaling and semi-transparency.

Tab. S7:Electrical parameters for best performing opaque cell, ST cell and ST module.

Device Type	Area (cm ²)	PCE (%)	V_{oc} (V)	J_{sc} (mA/cm ²)	FF (%)
Opaque cell (G/MX/4-FPEAI)	0.1	22.12	1.21	22.5	81.82
ST cell (Au/ITO)	0.1	18.3	1.22	20.5	75.5
	Area (cm ²)	PCE (%)	V_{oc} (V)	I_{sc} (mA)	FF (%)

Device Type	Area (cm ²)	PCE (%)	V _{oc} (V)	J _{sc} (mA/cm ²)	FF (%)
ST module	60.0	16.16	26.98	52.7	67.9

And the following statement in the main text:

“while strongly limiting the PCE drop (<13%) usually observed in scaling the device size (S.I. section S.I. 11 and Tab. S7 for detailed discussion).”

2. Also, no stability results on cell level are presented to support any early interpretation on the potential degradation mechanisms that can play in the outdoor tested samples.

We fully agreed with the concern raised by reviewer about the possibility to support an early interpretation of the outdoor stability behavior by performing indoor stress test. On this basis, following what was recently suggested by authoritative works in the literature^{17,18}, we performed light-cycling tests that are now inserted and commented in SI, Fig. S22c, as follows reported:

Indoor photo-stability under MPP tracking of semi-transparent encapsulated cells upon exposure to cycled light. As recently discussed in literature, the most effective way to perform a reliable stability analysis consists in light cycling test at constant temperature, representing a good trade-off between simplicity of execution in lab conditions while taking into account phenomena observed only in case of outdoor testing (that cannot be considered in case of constant light experiments).¹⁵ Indeed, transient behavior occurs in PSCs as a result of the coexistence of several dynamics with characteristic times spanning from time scales of seconds to hours. Thus, day–night cycling of devices can consider such slow dynamics as they occur on similar timescales. Following this idea, we performed a light–dark cycling test as recommended by the ISOS-LC-1 protocol, exposing the cells to simulated sunlight turned on and off with cycle periods of 24 h and duty cycles (light:dark) of 1:2, mimic an accelerated diurnal sun cycle (the solar cell is maintained at ambient conditions while the temperature is monitored and RH is maintained at 30%).¹⁶ The recorded maximum power (P_{MPP}) is reported in Fig. S7e for both G/EA and fully optimized G/MX/4-FPEAI semi-transparent cells, normalized at the initial value. The combined stabilization effect of the 4-FPEAI based SGP together with the presence of MX-Cl results in a T_{91} =600 hours vs a T_{80} =574 hours of the untreated device, demonstrating the superior stability (T_{91} 6.5 times larger) of the device employing the proposed graded homojunction approach. Despite more in depth analyses are required to elucidate the role of MX-Cl and 4-FPEAI SGP, we can confidently ascribe the improved stability to a synergistic effect of the two treatments. On one side, the addition of MX-Cl resulted in a perovskite film with improved crystallinity eventually reducing lattice distortion, lowering defect density and minimizing trap states, as demonstrated by XRD, transient PL and SEM characterizations. This, combined with the enlarged perovskite domains and mitigated grain boundary-induced trap states, makes the trapped charge driven degradation less effective,¹⁷ resulting in an overall enlargement of device lifetime. We cannot exclude a side role of the Cl in reducing the tensile strain within the perovskite layer, recently pointed out as one of the main degradation channel when light-dark cycling is applied.¹⁸ On the other side, the phenylalkylammonium compounds as 4-FPEAI has been proven to stabilize the perovskite through suppression of iodide ion migration, likely responsible for metastable perovskite behavior under light cycling observed for the G/EA device, while passivating the surface defects.¹⁹ In addition, the 4-FPEAI SGP strategy has been already proven to improve the long-term operational stability of the encapsulated devices under the continuous illumination in ambient condition.¹² Thus, the superior light cycling stability showed by the graded homojunction cell can be confidently ascribed to a combined stabilizing effect from 4-FPEAI and MX-Cl, definitively reducing the device metastable behavior.

Fig. S22: a) J-V curve in forward and reverse scan for the best-performing 2D material-engineered ST-PSC (G/MX/4-FPEAI structure); b) IPCE spectra with the integrated photocurrent density (Integrated J_{SC}) related to the best efficient optimized cell (G/MX/4-FPEAI) for both opaque and ST proposed device structure. c) Indoor photo-stability under MPP tracking of G/EA and G/MX/4-FPEAI semi-transparent encapsulated cells upon exposure to cycled light (ISOS-LC-1).

- (12) Yan, N.; Gao, Y.; Yang, J.; Fang, Z.; Feng, J.; Wu, X.; Chen, T.; Liu, S. F. Wide-Bandgap Perovskite Solar Cell Using a Fluoride-Assisted Surface Gradient Passivation Strategy. *Angew. Chem. Int. Ed.* **2023**, *62*, e202216668. <https://doi.org/10.1002/anie.202216668>.
- (15) Khenkin, M.; Köbler, H.; Remec, M.; Roy, R.; Erdil, U.; Li, J.; Phung, N.; Adwan, G.; Paramasivam, G.; Emery, Q.; et al. Light Cycling as a Key to Understanding the Outdoor Behaviour of Perovskite Solar Cells. *Energy Environ. Sci.* **2023**, *17* (2), 602–610. <https://doi.org/10.1039/d3ee03508e>.
- (16) Khenkin, M. V.; Katz, E. A.; Abate, A.; Bardizza, G.; Berry, J. J.; Brabec, C.; Brunetti, F.; Bulović, V.; Burlingame, Q.; Di Carlo, A.; et al. Consensus Statement for Stability Assessment and Reporting for Perovskite Photovoltaics Based on ISOS Procedures. *Nat. Energy* **2020**, *5* (1), 35–49. <https://doi.org/10.1038/s41560-019-0529-5>.
- (17) Ahn, N.; Kwak, K.; Jang, M. S.; Yoon, H.; Lee, B. Y.; Lee, J.; Pikhitsa, P. V.; Byun, J.; Choi, M. Trapped Charge-Driven Degradation of Perovskite Solar Cells. *Nat. Commun.* **2016**, *8* (May), 1–9. <https://doi.org/10.1038/ncomms13422>.
- (18) Shen, Y.; Zhang, T.; Xu, G.; Steele, J. A.; Chen, X.; Chen, W.; Zheng, G.; Li, J.; Guo, B.; Yang, H.; et al. Strain Regulation Retards Natural Operation Decay of Perovskite Solar Cells. *Nature* **2024**, *635* (8040), 882–889. <https://doi.org/10.1038/s41586-024-08161-x>.
- (19) Guo, Y.; Aperi, S.; Li, N.; Chen, M.; Yin, C.; Yuan, Z.; Gao, F.; Xie, F.; Brocks, G.; Tao, S.; et al. Phenylalkylammonium Passivation Enables Perovskite Light Emitting Diodes with Record High-Radiance Operational Lifetime: The Chain Length Matters. *Nat. Commun.* **2021**, *12* (1). <https://doi.org/10.1038/s41467-021-20970-6>.

Moreover, we modified the main text as follows:

“Moreover, the combination among the MX-Cl addition and the 4-FPEAI SGP strategy not only translated into superior device performance but also conferred superior robustness to the perovskite film against light-driven degradation, as demonstrated by prolonged light cycling test (Fig. S22c).”

3. Moreover, claims repeatedly being made in the manuscript that this work shows scalability of the perovskite and tandem PV technology, even alluding on the low LCoE that can be obtained, are not well placed here. The process is still very lab scale oriented (still spin coating based), highly manual - both on minimodule manufacturing as on tandem assembly - and at sizes still well below m² dimensions.

We agree with the reviewer in arguing that a new photovoltaic technology aiming to enter in the market must show a low LCOE. Achieving low LCOE is possible if industrially compatible and low-cost large-area deposition methods are proposed, such as large-area printing methods. Therefore, there are 2 levels of technology development to be considered. The first, intended as “scalability” of the device dimensions, consists in proposing a cell/module/panel structure that can minimize the performance decrease and the device processability issues when moving from a lab size to a dimension compatible with industrial standards. This can also be done using spin-coating techniques once optimizing the deposition of the cell constituting layers on the desired substrate size. The second level consists in trying to reduce the costs associated with the

production of the device, therefore using printing techniques that reduce material waste as much as possible and that are as compatible as possible with a high-yield production line. Both development levels contribute to a low LCOE of the new technology that is intended to be proposed on the market. In our work, we aim to provide a manufacturing strategy for semi-transparent perovskite devices that can be used as top cells in the 4-terminal tandem architecture while demonstrating great resilience to the performance losses usually experienced in the size scaling process. The manufacturing processes are lab-oriented because they are carried out in a research laboratory, which is currently not equipped with a pilot production line for perovskite modules. For this reason, a massive production of semi-transparent perovskite modules was reported in the manuscript, demonstrating a reasonable module performance repeatability not guaranteed when using the spin-coater, and obtained in our case thanks to the use of two-dimensional materials. Moreover, we already demonstrated that 2D materials-based approach combined with the spin-coating technique is effective in guaranteeing excellent performance repeatability at module and panel level, realizing the first ever showed in literature perovskite solar farm of 4.5 m², composed by 9 panels of 0.5 m², each of them realized by electrically connecting 40 opaque perovskite modules (10x10 cm² each one).¹ Thus, in our opinion, the use of spin-coater does not preclude the "scalability" of the device dimensions. As further confirmation, our research group has already demonstrated the scalability of spin-coated perovskite monolithic modules beyond the 9.5x9.5 cm² size, covering a surface of about 300 cm² (15.7x15.7 cm²) and demonstrating efficiencies above 17% for opaque devices.² While these dimensions are exactly compatible with those of the M2 silicon wafer size, in tandem technology it is necessary to consider the presence of the semi-transparent electrode, usually made with sputtered ITO, whose series resistance significantly limits the maximum dimensions of the module. In fact, in our work, we proposed to use module dimensions limited to 9.5x9.5 cm², size also strongly compatible with more "industrially oriented" deposition techniques such as blade-coating or slot dye coating, as extensively demonstrated in literature.³ However, such "out of glove box" techniques require a careful control of the environmental conditions during and after the deposition process, whose variability could preclude acceptable device manufacture and performance reproducibility. For these reasons, our choice fell on the spin-coating technique in inert environment, using an industrial spin-coater with the possibility to spin up to 25x25 cm² substrate size. We therefore believe that the use of spin-coating does not compromise in any way the scalability of our tandem technology, that have been designed with a "modular approach" for further extending the dimension of the tandem in the next future (up to several m² size), without compromise too much the final performance, while being compatible also with other bottom cell technologies available in the market, such as CIGS or CdTe PV. For these reasons, we strongly believe that our work provides a step forward the industrialization of the tandem technology, proposing a modular approach where perovskite top panel based on graded homojunction concept, can be realized employing both lab-scale or industrially oriented deposition techniques. This provides also a way for broadening the audience of our work in accordance with the broad audience of Nature Communication Journal, giving the chance to scientists and researchers to repeat and verify the proposed approach in their lab, without the need to have industrial facilities. We agree instead on the impact of the spin-coating technique can have on production costs, and therefore on the overall LCOE.

(1) Pescetelli, S.; Agresti, A.; Viskadourous, G.; Razza, S.; Rogdakis, K.; Kalogerakis, I.; Spiliarotis, E.; Leonardi, E.; Mariani, P.; Sorbello, L.; et al. Integration of Two-Dimensional Materials-Based Perovskite Solar Panels into a Stand-Alone Solar Farm. *Nat. Energy* **2022**, *7*, 597–607.

(2) H. Nikbakht, P. Mariani, L. Vesce, F. Di Giacomo, E. Leonardi, G. Viskadourous, E. Spiliarotis, K. Rogdakis, S. Pescetelli, A. Agresti, S. Bellani, F. Bonaccorso, E. Kymakis, . Di Carlo Upscaling Perovskite Photovoltaics: from 156 cm² Modules to 0.73 M² Panels. *Advanced Science* **2025**, *22,12*, 2416316.

(3) Geistert, K., Ternes, S., Ritzer, D. B. & Paetzold, U. W. Controlling Thin Film Morphology Formation during Gas Quenching of Slot-Die Coated Perovskite Solar Modules. *ACS Appl. Mater. Interfaces* **15**, 52519-52529 (2023)

For this reason, following the reviewer's suggestion, we have modified the text in the main as follows:

“Ultimately, incorporating 2D materials yielded high-performing ST-PSMs, reduced performance variability, and enabled reproducible production of large area PSK/Si tandem panels, thereby advancing industrially oriented production of the perovskite/Si tandem technology.³⁷”

(37) Čulík, P.; Brooks, K.; Momblona, C.; Adams, M.; Kinge, S.; Maréchal, F.; Dyson, P. J.; Nazeeruddin, M. K. Design and Cost Analysis of 100 MW Perovskite Solar Panel Manufacturing Process in Different Locations. *ACS Energy Lett.* **2022**, 3039–3044.

Regarding the “manual” techniques for manufacturing the tandem panel, these were dictated by the fact that the panels were assembled in the laboratory, without the possibility to employ very expensive and customized tools, typical of a PV industry. As an example, the stringer machine used to make the series connections between silicon cells costs from a few hundred thousand euros up to several million euros for the most advanced and automated production lines, making their use impossible in a standard research lab.

In our revised work we have now underlined this aspect, also making considerations in terms of efficiency loss when using manual rather than industrial stringing methods (see SI, SECTION S.I. 13). However, we believe that, looking at the results shown in this work obtained using lab-scale production processes, these are really promising and surely improvable once transferred to an industrial environment. In this view, our work represents one of the very few demonstrations present in the literature on the possibility to scale the 4T tandem technology at panel level, without asking for any modifications of the silicon production line. We are pretty confident that, with all of this in mind, the reviewer could appreciate the tremendous efforts displayed in this work for providing one of the first demonstration of a working large-area perovskite/silicon tandem panel.

Reviewer #3 (Remarks to the Author):

The manuscript titled “MXene-driven nanoscale graded homojunctions for advanced 4-terminal perovskite/silicon tandem solar panels” by Agresti et al. demonstrates and studies chlorine-terminated MXene ($\text{Ti}_3\text{C}_2\text{Cl}_2$) ETL and PEAI passivated HTL contacts for NIP perovskite solar cells and 4T silicon/perovskite tandems. The authors claim that this architecture creates a graded homojunction within the perovskite bulk that suppresses non-radiative recombination and facilitates carrier extraction for high-efficiency devices. That strategy is scaled to a 19.45% efficient 0.2m^2 tandem modules and fielded outdoors in Crete for 4 months. The module fabrication steps, and performance characterization were well documented with the inclusion of temperature and irradiance dependent data on reasonably sized tandem modules as well as details on module scribing, interconnects, and lamination processes. The scalability of this approach and field data could be of high interest to the field; however, I have some significant concerns limiting the potential impact of this work.

First, I do not find sufficient evidence to support the claim for the formation of a graded homojunction.

Thank you for your valuable comment. We appreciate the opportunity to clarify the experimental basis for our conclusions regarding the electronic structure of the perovskite films and the evidence supporting the graded homojunction formation. To address this point, we performed UPS measurements on four perovskite films specifically fabricated for this analysis, each with a reduced thickness of approximately 200 nm. This thickness is comparable to the lateral size of the MXene flakes (~200 nm) and was selected to ensure more uniform incorporation of the additives and to allow UPS signals to better reflect the bulk of the perovskite film, not just the topmost surface. In this way, the UPS measurements can provide useful insight on these four distinct thin perovskite films, each designed to isolate the effect of individual and combined additive treatments:

1. **REF:** untreated reference perovskite;

2. **MX-Cl**: perovskite containing $\text{Ti}_3\text{C}_2\text{Cl}_2$ MXenes;
3. **4-FPEAI**: perovskite post-treated with 4-FPEAI;
4. **MX-Cl / 4-FPEAI**: perovskite containing both additives.

Tab. 2. Extracted values of work function (WF), valence band maximum (VBM) with respect to the Fermi level, and ionization energy (IE = WF + VBM) for the four analyzed perovskite film configurations, measured by UPS. The results demonstrate distinct electronic effects induced by MXene and 4-FPEAI treatments, supporting the formation of dipoles and doping gradients.

Sample	WF (eV)	VBM (eV)	(IE) [eV]
REF	4.75	0.9	5.65
MX-Cl	4.55	0.9	5.45
MX-Cl / 4-FPEAI	4.70	0.8	5.50
4-FPEAI	4.85	0.7	5.55

From these results, we observe two different behaviors:

- In the MX-Cl sample, the VBM position remains unchanged with respect to the REF sample (0.9 eV from the Fermi energy level- E_F), while the WF decreases from 4.75 eV to 4.55 eV. This suggests the formation of a surface dipole at the interface, which modifies the vacuum level without altering the valence band position—a signature of interface dipole formation. This is consistent with our DFT calculation, which shows a vacuum level shift of -0.39 eV due to MXene-induced dipole formation.
- Conversely, in the 4-FPEAI sample, both the WF increases (to 4.85 eV) and the VBM shifts closer to E_F (from 0.9 to 0.7 eV). This implies a more complex interaction, where not only vacuum level bending occurs, but also an actual shift in the electronic band structure, possibly due to surface passivation and local electronic doping. DFT results support this observation, indicating a +0.14 eV vacuum level shift with 4-FPEAI relative to the REF.
- The MX-Cl/4-FPEAI sample shows an intermediate behavior, further supporting the scenario of a graded energy alignment across the film thickness, where the MXene dominates the bottom interface and 4-FPEAI modifies the top surface.

These findings strongly support the graded electronic structure scenario. The selective dipole formation and VBM modulation induced by the two additives at opposite sides of the perovskite layer result in a quasi-homojunction or gradient doping profile, which facilitates carrier separation and extraction, thereby reducing recombination.

We added the following text and table in the main text and in the experimental section, while we moved the discussion on Kelvin probe (KP) measurement in the SI.

From SEM images of **Fig. 3f**, clearly MX-Cl flakes are mainly distributed at bottom interface between perovskite/ mTiO_2 layer. The presence of MX-Cl at the ETL/perovskite interface calls for a possible tuning of the buried perovskite interface WF.²¹

Building on this, ultraviolet photoelectron spectroscopy (UPS), was employed to elucidate the progressive modulation of the perovskite energy levels depending on the different treatments. Notably, the combination of MXene and 4-FPEAI induces a graded shift in work function and valence band position across the series, consistent with the formation of a built-in electric field within the absorber. A full discussion and corresponding UPS spectra and energy level diagram reconstructed from the UPS data are provided in **section S.I. 6 in SI Fig. S11 and Fig. S12** respectively, while the extracted values of work function and valence band maximum (VBM) are summarized in **Tab. 2**, clearly highlighting the energetic alignment changes induced by each modification.

Tab. 2. Extracted values of work function (WF), valence band maximum (VBM) with respect to the Fermi level, and ionization energy (IE = WF + VBM) for the four analyzed perovskite film configurations, measured by UPS. The results demonstrate distinct electronic effects induced by MXene and 4-FPEAI treatments, supporting the formation of dipoles and doping gradients.

Sample	WF (eV)	VBM (eV)	(IE) [eV]
REF	4.75±0.1	0.9±0.1	5.65±0.1
MX-Cl	4.55±0.1	0.9±0.1	5.45±0.1
MX-Cl / 4-FPEAI	4.70±0.1	0.8±0.1	5.50±0.1
4-FPEAI	4.85±0.1	0.7±0.1	5.55±0.1

To confirm this effect, Kelvin probe force microscopy (KPFM) measurements were carried out on dedicated half-cell samples (FTO/G+cTiO₂/G+mTiO₂/perovskite), allowing us to extract WF values (Tab. S3) and to distinguish the contributions of MXene doping and SGP (see Fig. S13)

In the experimental section we added:

Photoelectron spectroscopy measurements were performed in a ThermoScientific ESCALAB QXi XPS apparatus in ultra-high vacuum conditions (10⁻¹⁰ mbar base pressure). The valence band region of the samples was investigated by UPS using the 21.2 eV radiation of the He I line and biasing the samples at -5 V, in order to determine both the work function and the position of the valence band edge. The electronic band structure of the perovskites was characterized by X-ray photoemission spectroscopy (XPS) using monochromatic 1486.8 eV photons of the Al K α line and analysed with the Thermo-Fischer Avantage Software. XPS measurements were performed on small devices (14x14 mm) applying the system charge compensation, while the work function was measured without charge compensation, by providing an additional contact to an edge of the device front surface.

Moreover, the complete discussion on UPS results with the relative figures (Figs: S11 and S12) are reported in SI

Ultraviolet photoelectron spectroscopy (UPS) measurements

To gain insight into the energetic effects induced by the surface gradient passivation and MXene doping engineering strategies, we performed ultraviolet photoelectron spectroscopy (UPS) measurements on a set of ad-hoc fabricated perovskite thin films with reduced thickness (comparable to the MXene flake size, ~200 nm) and varied interfacial treatments (REF, MX-Cl, 4-FPEAI, and MX-Cl/4-FPEAI). The extracted values of work function (WF) and valence band maximum (VBM) for each cell configuration are summarized in Tab. 2 of the main text.

As shown in Fig. S12, the energy level diagram reconstructed from the UPS data (Fig. S11) clearly reveals a systematic shift in both WF and VBM across the series.

The reference (REF) sample exhibits a WF of 4.75 eV and a VBM located 0.9 eV below the Fermi level (EF), corresponding to an ionization energy (IE) of 5.65 eV. Upon incorporation of Ti₃C₂Cl₂ MXenes (MX-Cl), the WF decreases to 4.55 eV while the VBM remains unchanged at 0.9 eV. This behavior indicates that the addition of MXenes enhances the n-type character of the perovskite, consistent with the formation of a surface dipole at the perovskite/MXene interface. Notably, this interface dipole shifts the vacuum level without affecting the perovskite band structure. Conversely, the use of 4-FPEAI as SGP induces a WF increase to 4.85 eV and a shift in the VBM to 0.7 eV below EF. These results suggest a clear modification of the perovskite electronic structure, indicating a p-type doping effect, in addition to surface dipole formation. The combined MX-Cl/4-

FPEAI treatment results in intermediate values ($WF = 4.70$ eV, $VBM = 0.8$ eV), consistent with a graded band alignment across the film thickness.

These findings are corroborated by DFT simulations, which show a vacuum level shift of -0.39 eV for the MXene interface and $+0.14$ eV for 4-FPEAI, supporting the observed WF modulation. Overall, the UPS data confirm the formation of a vertical energy level gradient across the perovskite layer, driven by the two treatments acting at opposite interfaces.

Fig. S11. UPS spectra for perovskite films modified with different interface strategies: reference (a–b), MX-Cl (c–d), MX-Cl/4-FPEAI (e–f), and 4-FPEAI (g–h). Panels (a, c, e, g) show the secondary electron cutoff region used to extract the WF, while panels (b, d, f, h) display the valence band region used to determine the VBM with respect to the EF. A gradual modulation of the WF and VBM is

observed across the series, highlighting the impact of dipole formation and band structure tuning induced by MXene doping and surface gradient passivation.

Fig. S12: Schematic energy level alignment of perovskite films extracted from UPS measurements for the four configurations: REF (untreated), MX-Cl, 4-FPEAI, and combined MX-Cl/4-FPEAI. The WF and VBM positions are plotted relative to the vacuum level. A progressive modulation of the energy levels is observed, with MX-Cl inducing a downward shift in WF (consistent with dipole-induced vacuum level bending) and 4-FPEAI causing an upward shift in both WF and VBM. The intermediate alignment in the combined treatment suggests the formation of a graded homojunction across the perovskite layer, promoting internal electric field formation and enhanced charge extraction.

Second, while the authors clearly advance device demonstration of halogen MXene doping for tuning perovskite interfaces, this concept has been reported before (e.g. interface calculations of $Ti_3C_2Cl_2$ DOI: 10.1016/j.vacuum.2023.112381; MXene work function tuning for perovskite solar cells DOI: 10.1039/D2NR02799B).

We thank the Reviewer for highlighting key prior work in the field, including our own earlier study (Titanium-carbide MXenes for work function and interface engineering in perovskite solar cells). Indeed, that work introduced the concept of interface dipole formation via MXenes and demonstrated that WF tuning without valence band distortion could be achieved through selected surface terminations (T_x), confirming that MXenes can effectively modulate energy alignment at the perovskite/ETL interface. However, the current study builds significantly on that foundation and introduces several key innovations.

Unlike our previous work that employed mixed-termination $Ti_3C_2T_x$ MXenes, here we use MXenes synthesized with exclusively Cl terminations. This pure halogen termination introduces stronger and more defined interfacial dipoles (confirmed by DFT with a -0.39 eV vacuum shift) and improves film crystallinity and defect passivation, as supported by SEM, XRD, and TRPL.

Moreover, while both MXenes and 4-FPEAI have been studied individually in prior literature, their simultaneous and spatially controlled incorporation is novel. In our architecture, MXenes accumulate at the bottom interface (near the ETL), while 4-FPEAI is applied at the top surface during anti-solvent dripping, resulting in a graded modulation of the perovskite energy landscape along its thickness.

This dual-modifier strategy leads to the formation of an internal n-p gradient across the perovskite layer, a concept not previously demonstrated. UPS measurements clearly show a progression in WF and VBM values across REF, MX-Cl, 4-FPEAI, and MX-Cl/4-FPEAI samples, consistent with a built-in electric field spanning from ETL to HTL interfaces. This experimental validation of the internal band alignment further differentiates our work from prior surface-only studies.

Lastly, beyond materials development, we implement this strategy in large-area modules and semi-transparent tandem top cells, achieving over 16% PCE and high stability under outdoor conditions, demonstrating its practical relevance and scalability.

Taking together, these elements introduce a novel functional architecture that goes beyond the concepts explored in our previous publication and distinguish this work from earlier MXene-related studies.

For greater clarity and to better highlight the novelty of our approach, we have added the following sentence to the main text of the manuscript:

Crucially, this work introduces a vertically graded perovskite homojunction, enabled by the spatially resolved integration of Cl-terminated MXenes and 4-FPEAI, a concept not previously demonstrated, representing a new class of interface and energy landscape engineering in perovskite photovoltaics.

Third, the reported tandem module efficiency of 19.45% is not above the efficiency of an average single junction silicon module, even though the tandem pairing resulted in a net 1.05% absolute efficiency increase over the Si HJT cells used in this study and the reported field stability is not at the state-of-the-art of other public reports for perovskite-based tandem modules

([https://urldefense.com/v3/https://pvfact.sandia.gov/results-and-](https://urldefense.com/v3/https://pvfact.sandia.gov/results-and-data/)

[data/ ;!!O5Bi4QcV!AhStXVfRb4M0c2PywTXMfB IGLzEuLZkOnh8pur96Sf74vNpdeoJ0-](https://urldefense.com/v3/https://pvfact.sandia.gov/results-and-data/;!!O5Bi4QcV!AhStXVfRb4M0c2PywTXMfB IGLzEuLZkOnh8pur96Sf74vNpdeoJ0-16p1095RgyfvE6JrfjxE842t00SPCy7Xv3OcM$)

[16p1095RgyfvE6JrfjxE842t00SPCy7Xv3OcM\\$](https://urldefense.com/v3/https://pvfact.sandia.gov/results-and-data/;!!O5Bi4QcV!AhStXVfRb4M0c2PywTXMfB IGLzEuLZkOnh8pur96Sf74vNpdeoJ0-16p1095RgyfvE6JrfjxE842t00SPCy7Xv3OcM$)).I have included additional comments below but unfortunately cannot recommend publication of this manuscript in nature communications in its current form.

We believe that the main message of our paper does not consist in claiming record efficiency. Such claims should be supported by certified efficiency measurements (as those indicated by the Reviewer). Instead, our study aims to introduce a novel design concept for semi-transparent perovskite top cells in 4-terminal perovskite/Si tandem architectures.

Specifically, this study advances the state of the art in interface and energy landscape engineering by introducing a vertically graded homojunction within the perovskite absorber, realized through the spatially selective integration of chlorine-terminated MXenes and 4-FPEAI. This synergistic combination enables simultaneous work function modulation and defect passivation across the perovskite thickness, establishing an internal electric field that promotes efficient charge extraction and suppressed recombination.

Following your suggestion, to further emphasize this point we added the corresponding statement to the main text of the revised manuscript, as detailed in the previous point.

We would like also to underline to the reviewer that the bifacial Si-HJT technology used in this work is a state of art silicon technology, produced in the industrial facilities of 3SUN company, located in Catania, south of Italy. 3SUN, is the largest high-performance photovoltaic cell and module production factory in Europe, with a production up to 3GW of photovoltaic module per year based on silicon heterojunction (HJT) technology (Investment of 1B€ of which around 200 M€ from EU and national funding; 150,000 m² facility). 3SUN is working on the next perovskite/Si-HJT technology to reach more than 30% efficiency over the next few years and thanks to the demonstration of potentialities carried out during our joint work, the fab hosts now a pilot line for 2D material-based perovskite/silicon tandem panels, aiming to launch tandem panels to the market in the next coming years (see in **Fig. R1** reported below the roadmap released by 3Sun at <https://www.enelgreenpower.com/media/news/2022/09/3sun-gigafactory-evolution-solar-technologies>).

Fig.R1: Roadmap released by 3Sun at <https://www.enelgreenpower.com/media/news/2022/09/3sun-gigafactory-evolution-solar-technologies>, clearly including the PSK/Si technology as expected to be fully up and running at the 3Sun Gigafactory in 2025, promises to further boost the efficiency of photovoltaic modules up to over 30%, while also aiming to ensure a life span of at least 35 years.

We would kindly like to clarify a possible misunderstanding regarding the absolute efficiency gain demonstrated by our “lab-fabricated” tandem panel. In the manuscript, we report a PCE of 19.45% for the tandem panel, compared to 16.7% for the manually stringed silicon bottom cell, resulting in an absolute improvement of approximately **2.74 percentage points**. It seems that the reviewer may have instead compared our tandem panel to an industrially fabricated silicon cell (PCE = 18.4%), leading to a reported difference of 1.05 percentage points.

However, this comparison may not reflect the actual experimental setup used in our work. This point was already addressed in the submitted version of the manuscript, with further clarification provided in the Supporting Information, Section S.I. 13. For your convenience, we report below the relevant paragraph where this aspect is discussed in more detail.

MAIN TEXT: “We should point out that, due to the manual stringing (electrical connection among adjacent cells), the bottom Si-HJT module have a reduce efficiency compared to those obtained on an automatized industrial line (see SI Section 13).”

SI: “In this way, we fabricated a Si-HJT module having the same electrical connections and same lamination process as those employed for Si bottom module composing the tandem all-in-one DEM2. In the as-realized DEM3, the cell stringing process is manually realized using 6 mm width charger collector tapes from 3M. Conversely, in an industrially relevant production environment, Si-HJT cells are usually stringed and laminated using industrial tools (such as an industrial stringing machine), material specifically design for Si-HJT lamination (such as external glass texturized 3 mm thick, polyolefin foil) and higher temperature hot vacuum lamination procedure (performed at 180°C). All these aspects strongly affect the final Si-HJT module performance. Indeed, when considering the lab-produced module (DEM3) it showed maximum PCE of 16.7% once tested at STC while the Si-HJT realized in the 3SUN industrial facilities showed PCE approaching 18.5% (see **Tab. S5**). The main PCE losses at 1SUN conditions in the case of lab-made DEM3 have to be imputed to series path introduced during the manual cell stringing procedure, eventually penalizing FF and I_{sc}. Conversely, when the irradiance is set to 400 W/m² (nearby the irradiance experienced from a filtered Si-HJT module in a tandem panel) the FF gap between the lab-made DEM3 and the 3SUN demonstrator is strongly

reduced with respect to the case of 1 SUN irradiation, leading to a PCE gap of around 1 percentage point. Indeed, the lower generated current under reduced irradiance level minimizes the impact of the series resistive path introduced in case of a manual cell stringing process, leading to a more comparable performance among the lab and industry made Si-HJT module.

Tab. S5: Electrical parameters for both lab-made and 3SUN made Si-HJT module (4 series connected cells) in case of 1SUN and 400 W/m² irradiance level.

Module (4 cells)	V _{oc} (V)	I _{sc} (A)	FF (%)	PCE (%)
Si-HJT (lab-made) @ STC	2.86	8.18	57.57	16.71
Si-HJT (3SUN) @ STC	2.81	9.16	74.8	18.4
Si-HJT (lab-made) @ 400 W/m ²	2.76	3.26	69.8	7.29
Si-HJT (3SUN) @ 400 W/m ²	2.81	3.66	77.5	8.33

Thus, for a fair comparison between the proposed 2D materials-engineered tandem panel (DEM2) and the Si-HJT based single junction technology, DEM2 and DEM3 should be compared. From **Tab. S5**, we can estimate an overall PCE increase of 2.74 percentage points when moving from the Si-HJT to tandem technology.”

Additional comments:

- 1) *The authors report UPS and Kelvin probe CPD data on Ti₃C₂Cl₂ MXenes in Fig S6 and additional CPD measurements in Fig S13 and half-stacks in Tab. 2. These experiments appear to be film surface measurements but not a depth profile of an actual perovskite film. As far as I can tell, the perovskite work function and band structure primarily come from DFT simulations and TiberCAD simulations in Figs S15-S17. A work function changes due to doping of a contact layer can induce an electric field at a contact interface, but that is insufficient to claim the formation of a graded homojunction. Is there any experimental evidence to suggest that there are doping changes within the perovskite layer as a result of the new contact strategy that aids in the suppression of non-radiative recombination and carrier collection?*

We sincerely thank the Reviewer for the thoughtful comment. We acknowledge that in the original submission, the interpretation regarding the formation of a graded homojunction was mainly based on CPD trends and simulations, which alone were not sufficient to provide direct experimental evidence of energy level modulation within the perovskite bulk. In response to the Reviewer’s suggestion, we have now included new Ultraviolet Photoelectron Spectroscopy (UPS) measurements (see **Tab. 2** and Supporting **Figs. S11** and **S12**), carried out on a series of carefully designed perovskite films incorporating individual and combined surface/contact passivation strategies (MX-Cl and/or 4-FPEAI). These films were realized with reduced thickness to ensure that the UPS signal was representative of the electronic structure of the perovskite absorber and sensitive to vertical compositional and electronic modulation induced by the additive distribution. The UPS results show a systematic and progressive shift of both the work function and the valence band maximum (VBM) across the sample series, with MX-Cl leading to a downward vacuum level shift (dipole formation and increased n-type character), while 4-FPEAI induces a larger upward shift and clear band bending (toward p-type). Notably, the sample with both MX-Cl and 4-FPEAI exhibits intermediate energy levels, suggesting the formation of a graded band structure consistent with a vertical n-to-p transition within the perovskite. These experimental observations, now integrated into the main text and SI, support our hypothesis of a graded homojunction induced by the spatial distribution of the interfacial modifiers, which is also corroborated by the enhanced photovoltaic performance and reduced non-radiative recombination

losses. In the revised form of our article, we reported these achievements as detailed in the answer to your above “first” comment.

- 2) *The authors should be careful about making claims of grain sizes based on SEM methods. It has been shown in the literature that these techniques cannot identify true grain size or grain boundaries (DOI: 10.1021/acsenergylett.8b01704, DOI:10.1016/j.joule.2018.12.011). It is reasonable to say “apparent grains”, “perovskite domain”, or to describe the “morphology”, but this should be taken into consideration when attributing film properties to SEM images.*

Thank you for pointing this out. We have revised the text in the main text and Supporting Info to avoid overinterpreting SEM images and now refer to 'perovskite domains' as reported in the following changed version:

“SGP has a minimal impact on grain size and surface morphology (**Fig. S8**), whereas MX-Cl doping, resulted in a noticeable increase in perovskite **domains** compared to the pristine layer (**Fig. 3a** and **3c**). The presence of small **domains** around the enlarged ones in the MX-Cl-doped film indicated that MXene flakes locally influenced perovskite morphology.”

And in SI:

“On the other side, MX-doping is strongly effective in reducing the defect states density thanks to an enlargement in perovskite **domains** where MXenes are present, by reducing the impact of the grain boundaries that are well known to insert deep trap state levels in the perovskite gap.”

- 3) *The authors claim “Moreover, the hydrophobic nature of MX-Cl further boosts both device efficiency and operational stability, as confirmed by Maximum Power Point (MPP) tracking of the best-performing cell (Fig. S12).” However, figure S12 only shows 180s of maximum power point data. 180s is not sufficient to claim operational stability and does not compare against control. Furthermore, I do not follow the reasoning as to how the MPP data is related to the hydrophobic nature of MX-Cl.*

We fully agree with the concern raised by the reviewer about the need to further support our claim on improved operational stability of our graded homojunction cell against control. On this basis, following what was recently suggested^{17,18} by authoritative works in the literature, we performed light-cycling tests that are now inserted and commented in SI, **Fig. S22c**, as follows reported:

Indoor photo-stability under MPP tracking of semi-transparent encapsulated cells upon exposure to cycled light. As recently discussed in literature, the most effective way to perform a reliable stability analysis consists in light cycling test at constant temperature, representing a good trade-off between simplicity of execution in lab conditions while taking into account phenomena observed only in case of outdoor testing (that cannot be considered in case of constant light experiments).¹⁵ Indeed, transient behavior occurs in PSCs as a result of the coexistence of several dynamics with characteristic times spanning from time scales of seconds to hours. Thus, day–night cycling of devices can consider such slow dynamics as they occur on similar timescales. Following this idea, we performed a light–dark cycling test as recommended by the ISOS-LC-1 protocol, exposing the cells to simulated sunlight turned on and off with cycle periods of 24 h and duty cycles (light:dark) of 1:2, mimic an accelerated diurnal sun cycle (the solar cell is maintained at ambient conditions while the temperature is monitored and RH is maintained at 30%).¹⁶ The recorded maximum power (P_{MPP}) is reported in **Fig. S7e** for both G/EA and fully optimized G/MX/4-FPEAI semi-transparent cells, normalized at the initial value. The combined stabilization effect of the 4-FPEAI based SGP together with the presence of MX-Cl results in a T_{91} =600 hours vs a T_{80} =574 hours of the untreated device, demonstrating the superior stability (T_{91} 6.5 times larger) of the device employing the proposed graded homojunction approach. Despite more in depth analyses are required to elucidate the role of MX-Cl and 4-FPEAI SGP, we can confidently ascribe the improved stability to a synergistic effect of the two treatments. On one side, the addition of MX-Cl resulted

in a perovskite film with improved crystallinity eventually reducing lattice distortion, lowering defect density and minimizing trap states, as demonstrated by XRD, transient PL and SEM characterizations. This, combined with the enlarged perovskite domains and mitigated grain boundary-induced trap states, makes the trapped charge driven degradation less effective,¹⁷ resulting in an overall enlargement of device lifetime. We cannot exclude a side role of the Cl in reducing the tensile strain within the perovskite layer, recently pointed out as one of the main degradation channel when light-dark cycling is applied.¹⁸ On the other side, the phenylalkylammonium compounds as 4-FPEAI has been proven to stabilize the perovskite through suppression of iodide ion migration, likely responsible for metastable perovskite behavior under light cycling observed for the G/EA device, while passivating the surface defects.¹⁹ In addition, the 4-FPEAI SGP strategy has been already proven to improve the long-term operational stability of the encapsulated devices under the continuous illumination in ambient condition.¹² Thus, the superior light cycling stability showed by the graded homojunction cell can be confidently ascribed to a combined stabilizing effect from 4-FPEAI and MX-Cl, definitively reducing the device metastable behavior.

Fig. S22: a) J-V curve in forward and reverse scan for the best-performing 2D material-engineered ST-PSC (G/MX/4-FPEAI structure); b) IPCE spectra with the integrated photocurrent density (Integrated J_{sc}) related to the best efficient optimized cell (G/MX/4-FPEAI) for both opaque and ST proposed device structure. c) Indoor photo-stability under MPP tracking of G/EA and G/MX/4-FPEAI semi-transparent encapsulated cells upon exposure to cycled light (ISOS-LC-1).

- (12) Yan, N.; Gao, Y.; Yang, J.; Fang, Z.; Feng, J.; Wu, X.; Chen, T.; Liu, S. F. Wide-Bandgap Perovskite Solar Cell Using a Fluoride-Assisted Surface Gradient Passivation Strategy. *Angew. Chem. Int. Ed.* **2023**, *62*, e202216668. <https://doi.org/10.1002/anie.202216668>.
- (15) Khenkin, M.; Köbler, H.; Remec, M.; Roy, R.; Erdil, U.; Li, J.; Phung, N.; Adwan, G.; Paramasivam, G.; Emery, Q.; et al. Light Cycling as a Key to Understanding the Outdoor Behaviour of Perovskite Solar Cells. *Energy Environ. Sci.* **2023**, *17* (2), 602–610. <https://doi.org/10.1039/d3ee03508e>.
- (16) Khenkin, M. V.; Katz, E. A.; Abate, A.; Bardizza, G.; Berry, J. J.; Brabec, C.; Brunetti, F.; Bulović, V.; Burlingame, Q.; Di Carlo, A.; et al. Consensus Statement for Stability Assessment and Reporting for Perovskite Photovoltaics Based on ISOS Procedures. *Nat. Energy* **2020**, *5* (1), 35–49. <https://doi.org/10.1038/s41560-019-0529-5>.
- (17) Ahn, N.; Kwak, K.; Jang, M. S.; Yoon, H.; Lee, B. Y.; Lee, J.; Pikhitsa, P. V.; Byun, J.; Choi, M. Trapped Charge-Driven Degradation of Perovskite Solar Cells. *Nat. Commun.* **2016**, *8* (May), 1–9. <https://doi.org/10.1038/ncomms13422>.
- (18) Shen, Y.; Zhang, T.; Xu, G.; Steele, J. A.; Chen, X.; Chen, W.; Zheng, G.; Li, J.; Guo, B.; Yang, H.; et al. Strain Regulation Retards Natural Operation Decay of Perovskite Solar Cells. *Nature* **2024**, *635* (8040), 882–889. <https://doi.org/10.1038/s41586-024-08161-x>.
- (19) Guo, Y.; Aperi, S.; Li, N.; Chen, M.; Yin, C.; Yuan, Z.; Gao, F.; Xie, F.; Brocks, G.; Tao, S.; et al. Phenylalkylammonium Passivation Enables Perovskite Light Emitting Diodes with Record High-Radiance Operational Lifetime: The Chain Length Matters. *Nat. Commun.* **2021**, *12* (1). <https://doi.org/10.1038/s41467-021-20970-6>.

We also modified the main text by briefly recalling the abovementioned stability analysis as in the following:

“Moreover, the combination among the MX-Cl addition and the 4-FPEAI SGP strategy not only translated into superior device performance but also conferred superior robustness to the perovskite film against light-driven degradation, as demonstrated by prolonged light cycling test (Fig. S22c).”

Following the reviewer suggestion, we removed from the main text the sentence related to the superior stability of the graded homojunction due to the hydrophobicity of the employed MXenes, since it could be misleading. Indeed, the hydrophobicity of the chlorine-based MXenes can have a role only in slowing down the mTiO_2 /perovskite interface degradation, since they are sitting at this device interface, as demonstrated by the cross-sectional SEM images reported in **Fig. 3d**. In the revised version of the SI we discussed more in detail the possible stabilization effects of MX-Cl once the graded homojunction undergoes prolonged light-cycling stress test. Moreover, as suggested by the referee we also modified the normalized MPP tracking by adding the same plot for the reference device and following reported

Fig. S7: Density of Current-Voltage (J-V) characteristics, acquired in forward and reverse voltage scan directions for the best efficient **a) G/CB, b) G/EA, c) G/4-FPEAI, d) G/MX/4-FPEAI** cells. **e) Normalized Maximum Power Point (MPP) tracking for the best-performing opaque cell employing the 2D material-engineered structure (G/MX/4-FPEAI) vs reference structure (G/EA).**

4) In the SI, it is claimed that drawing from Table S5, the 4T tandem pairing resulted in a 2.74% PCE improvement over the silicon HJT module alone. However, it appears that DEM2 is a 19.45% 4T module fabricated with 3SUN. HJT silicon (18.4%; Table S5). Wouldn't this indicate an efficiency improvement of 1.05% over the single junction silicon?

We would like to point out to the reviewer that absolute efficiency improvement demonstrated by our “lab-fabricated” tandem panel (showing a PCE of 19.45%) is calculated versus the “manually stringed” silicon bottom module (showing a PCE of 16.7%), which is around 2.74 percentage points (and not 1.05 percentage points, obtained by comparing the tandem panel realized in our research lab with the industrially made silicon bottom module showing PCE=18.4%). This aspect has been discussed in the original version of the manuscript

with a more in depth discussion reported in SI. We are now reporting in the following for your convenience the paragraph where this aspect is discussed.

MAIN TEXT: “We should point out that, due to the manual stringing (electrical connection among adjacent cells), the bottom Si-HJT module have a reduce efficiency compared to those obtained on an automatized industrial line (SI, S.I. Section 13).”

SI: “In this way, we fabricated a Si-HJT module having the same electrical connections and same lamination process as those employed for Si bottom module composing the tandem all-in-one DEM2. In the as-realized DEM3, the cell stringing process is manually realized using 6 mm width charger collector tapes from 3M. Conversely, in an industrially relevant production environment, Si-HJT cells are usually stringed and laminated using industrial tools (such as an industrial stringing machine), material specifically design for Si-HJT lamination (such as external glass texturized 3 mm thick, polyolefin foil) and higher temperature hot vacuum lamination procedure (performed at 180°C). All these aspects strongly affect the final Si-HJT module performance. Indeed, when considering the lab-produced module (DEM3) it showed maximum PCE of 16.7% once tested at STC while the Si-HJT realized in the 3SUN industrial facilities showed PCE approaching 18.5% (see **Tab. S5**). The main PCE losses at 1SUN conditions in the case of lab-made DEM3 have to be imputed to series path introduced during the manual cell stringing procedure, eventually penalizing FF and I_{sc} . Conversely, when the irradiance is set to 400 W/m² (nearby the irradiance experienced from a filtered Si-HJT module in a tandem panel) the FF gap between the lab-made DEM3 and the 3SUN demonstrator is strongly reduced with respect to the case of 1 SUN irradiation, leading to a PCE gap of around 1 percentage point. Indeed, the lower generated current under reduced irradiance level minimizes the impact of the series resistive path introduced in case of a manual cell stringing process, leading to a more comparable performance among the lab and industry made Si-HJT module.

Tab. S5: Electrical parameters for both lab-made and 3SUN made Si-HJT module (4 series connected cells) in case of 1SUN and 400 W/m² irradiance level.

Module (4 cells)	V _{oc} (V)	I _{sc} (A)	FF (%)	PCE (%)
Si-HJT (lab-made) @ STC	2.86	8.18	57.57	16.71
Si-HJT (3SUN) @ STC	2.81	9.16	74.8	18.4
Si-HJT (lab-made) @ 400 W/m ²	2.76	3.26	69.8	7.29
Si-HJT (3SUN) @ 400 W/m ²	2.81	3.66	77.5	8.33

Thus, for a fair comparison between the proposed 2D materials-engineered tandem panel (DEM2) and the Si-HJT based single junction technology, DEM2 and DEM3 should be compared. From **Tab. S5**, we can estimate an overall PCE increase of 2.74 percentage points when moving from the Si-HJT to tandem technology.”

5) *The authors should provide accelerated stability data (with at a minimum light + heat, preferably 85C or higher) for the 4F-PEAI/MX-Cl architecture vs. control.*

We fully agree with the concern raised by the reviewer about the need to further support our claim on improved operational stability of our graded homojunction cell against control. On this basis, following what recently suggested by authoritative works in the literature^{17,18}, we performed light-cycling tests that are now inserted and commented in SI, **Fig. S22c**, reported in this document as answer to question 3.

6) *The authors should consider extending the outdoor field test and providing a post-mortem characterization of the fielded module to identify apparent degradation modes of this architecture.*

We thank the reviewer for this thoughtful and insightful suggestion. We fully agree that post-mortem characterization of the fielded module could provide valuable information on the degradation mechanisms specific to this architecture. However, such an in-depth analysis falls outside the scope and objectives of the present work, which focuses primarily on the implementation and initial outdoor validation of the graded homojunction strategy in 4T tandem modules. Given the substantial amount of experimental work and data analysis required, we believe that a comprehensive post-mortem study would be more appropriately addressed in a dedicated follow-up publication. A similar approach was taken in our previous work on all-perovskite solar farms, where degradation analysis was reported in a separate study [E. Spiliarotis et al., 2025, EES Solar, DOI: 10.1039/d5el00042d]. Nonetheless, we appreciate the reviewer's comment and have included a note in the Conclusions to acknowledge this as an important direction for future research.

"Post-operation diagnostic analysis of field-deployed modules could offer valuable insight into the intrinsic degradation pathways and stability limitations of the proposed device architecture."

Reviewer #4 (Remarks to the Author):

In this manuscript, the authors demonstrate a synergistic approach of doping a two-dimensional material, MXene $\text{Ti}_3\text{C}_2\text{Cl}_2$, in the perovskite absorber layer and adding 4-FPEAI to the antisolvent in anticipation of forming a n-p junction, to enhance the performance of perovskite solar cells. And the method is extended to semi-transparent cells and four-terminal perovskite/silicon tandem devices. Finally, PCE of 19.45% was achieved on a 0.2 m² 4T tandem device and field tested for stability. However, I don't see much correlation with the methodology described in this manuscript in the section of the tandem devices, and it doesn't adequately describe the characterization of tandem devices.

We thank the reviewer for the insightful comments. We respectfully address the two main points raised:

1. On the correlation between the methodology and the tandem device section:

We would like to clarify that the same graded homojunction engineering strategy — combining MXene doping and 4-FPEAI-based surface gradient passivation — was employed in both the small-area opaque perovskite solar cells and the semi-transparent top cells used in the four-terminal (4T) tandem architecture. This is explicitly stated in the Results section and detailed in the Supporting Information (Section 10). The improved performance of the ST-PSMs and the resulting 4T tandem modules confirms the scalability and transferability of the strategy. Nevertheless, we have revised the manuscript to better highlight the methodological continuity between the cell-level and tandem-level implementations. The modified text is following reported:

"By employing the same 2D material-engineering strategies, MXene doping and 4-FPEAI-based surface gradient passivation, previously validated in opaque devices, the semi-transparent perovskite top cells used in the 4T tandem architecture demonstrated enhanced PCEs up to 18.3% (av. PCE = 17.63%) due to a higher V_{OC} (av. $V_{OC} = 1.2$ V) and J_{SC} (av. $J_{SC} = 19.92$ mA/cm²) compared to the REF sample (see **Tab. S6, Figs. S19–22a, S20 and S21 for a detailed analysis), confirming the effectiveness of 2D material-engineering approach even for semi-transparent structures."**

and

"This confirms the scalability and effectiveness of the approach at module level."

Moreover, we carried out a detailed analysis onto the performance losses occurring moving from small-area opaque cells to large-area ST modules, highlighting the crucial role our graded homojunction strategy had in limiting the scaling losses. We underlined this aspect in section S.I: 11, as follows reported:

Cell-to-module performance gap analysis.

To clarify the link between small-area opaque devices and the performance of large-area semi-transparent (ST) modules, we carried out a loss analysis identifying the main contributing factors. The best-performing opaque device (G/MX/4-FPEAI) exhibited a PCE of 22.12%, while the corresponding 60 cm² ST module reached 16.2%, resulting in an absolute performance drop of ~5.96 percentage points.

This gap can be attributed to three main sources:

(i) Optical losses due to the absence of the reflective metal electrode in ST configurations, which reduces photon recycling and light harvesting. To quantify the effect of ST electrode on photovoltaic performance, we compared our optimized opaque device (G/MX/4-FPEAI) with its ST counterpart featuring an Au/ITO top electrode. The opaque device exhibited a PCE of 22.12%, while the ST cell reached 18.3%, under identical illumination and processing conditions (see **Tab. S7**). This ~17.2 % relative efficiency drop (~3.82 absolute percentage points) can be attributed primarily to optical losses due to the removal of the reflective metal electrode, which otherwise enhances internal light trapping and photon recycling. With the aim to quantify the impact of the ST electrode onto the device photocurrent, we performed IPCE measurements on both the best performing opaque and semi-transparent cell, engineered with graded homojunction strategy. By computing the integrated current density, we can conclude that the replacement of opaque electrode with a ST one (gold/ITO) contributed to a J_{sc} loss of ~2.0 mA/cm². This is in line with what observed by De Wolf, S. and co-workers,^{20,21} which employed optical modeling (Transfer Matrix Method (TMM)) of the device stack to predict a 10–12% of incident photons loss due to front-surface reflection and incomplete absorption in the perovskite because of the reduced optical path length. These factors jointly reduce the short-circuit current density (J_{sc}) by ~ 1.8–2.2 mA/cm². In our case the reflection at the glass substrate is the same for both the ST and opaque devices, concluding that the observed J_{sc} drop can be attributed to the absence of photon recycling in the 580-730 nm region of the visible spectra.

(ii) Electrical losses from the transparent electrode stack (Au/ITO), including increased sheet resistance and contact resistance, resulting in an FF drop of ~6.32%. Nonetheless, the retained high V_{oc} (~1.22 V) and FF (~75%) in the ST cell confirm that our band alignment strategy remains effective under reduced optical confinement.

(iii) Scaling-related effects, including increased surface non-uniformity, edge shunting, and series resistance due to larger interconnection paths, contributing to a further loss in FF of -7.5% moving from the ST-cell to the ST-module. FF loss is the main responsible for the observed drop in PCE moving from ST-cell to ST-modules, since V_{oc} and J_{sc} for each large area cells (LAC, with an active area of 2.5 cm²) composing the module did not undergo a significant drop (V_{oc_LAC} ~ 1.25 V and J_{sc_LAC} ~ 20.8 mA/cm², see **Tab. S7**)

Despite these inevitable losses, the use of a vertically graded homojunction via Cl-MXene doping and 4-FPEAI passivation proved critical in maintaining high V_{oc} (1.226 V) and FF (67-69%) at the module level. This highlights the effectiveness of the interface and band alignment engineering in mitigating efficiency degradation upon scaling and semi-transparency.

Tab. S7:Electrical parameters for best performing opaque cell, ST cell and ST module.

Device Type	Area (cm ²)	PCE (%)	V_{oc} (V)	J_{sc} (mA/cm ²)	FF (%)
Opaque cell (G/MX/4-FPEAI)	0.1	22.12	1.21	22.5	81.82
ST cell (Au/ITO)	0.1	18.3	1.22	20.5	75.5

Device Type	Area (cm ²)	PCE (%)	V _{oc} (V)	J _{sc} (mA/cm ²)	FF (%)
	Area (cm ²)	PCE (%)	V _{oc} (V)	I _{sc} (mA)	FF (%)
ST module	60.0	16.16	26.98	52.7	67.9

We also added the following statement in the main text:

“while strongly limiting the PCE drop (<13%) usually observed in scaling the device size (see in S.I. section S.I. 11 and **Tab. S7** for detailed discussion).”

2. On the adequacy of tandem device characterization:

We acknowledge that a more detailed description of the tandem characterization could further strengthen the manuscript. We note, however, that the characterization of the semi-transparent top perovskite module and the filtered Si-HJT bottom cell, including IV curves, optical filtering effects, and bifacial power generation density (PGD), are provided in both the main text (**Fig. 6** and **Fig. 7**) and the Supporting Information (**Sections 12–14**). To address the reviewer’s concern, we have further clarified the methodology used to evaluate the outdoor performance and stability and explicitly referred to the relevant SI sections in the revised manuscript. In the following the relevant changes we made following the reviewer’s comments:

MAIN TEXT: Outdoor measurements performed using a metallic mounting base were carried out with the module installed at a fixed inclination of 30° and a height of approximately 100 cm above ground (see **Fig. 7g**). From the outdoor performance monitoring at open circuit conditions, the I-V curves for top ST-PSP were acquired during a 3-months period (from August to November 2023) and the main electrical parameters were normalized in STC (1000 W/m² irradiance and 25°C temperature) considering both the temperature and irradiance dependency of the main electrical parameters (**SI section 14**).

SI: The prolonged outdoor testing was performed on DEM2 panel at open circuit condition, following the ISOS-O-2 protocol. Thus, the panel was not “continuously monitored” in a strict sense, but was left mounted on the rack in the operative position at open-circuit and measured once per month. The parameters reported in **Fig. 7f** were extracted from the I-V curves, acquired for the perovskite top panel after a stabilization time of 10 minutes, since we observed an increase in the delivered power till reaching a stabilized value after 10 minutes (see **Fig. S29a**). Since the outdoor characterizations are typically subjected to the instantaneous variation of the irradiance and the panel temperature, we selected a clear sky day in the first half of the month performing the measurements when the irradiance was supposed to be maximum (at 12:30 am). The irradiance daily curves for Hellenic Mediterranean University site in Heraklion (35.31959; 25.10244) recorded during the day when the outdoor I-V curve were acquired, are reported in **Fig. S29b**. Once acquired the I-V curves, the extracted electrical parameters were normalized in STC (1000 W/m² irradiance and 25°C temperature) considering i) the panel temperature and the irradiance values recorded at the measurement time (at 12:30 am, see values reported in **Tab. S10**) ii) both temperature and irradiance dependency of the main electrical parameters reported in **Fig. S28**.

Fig. S29: a) Normalized power at maximum power point (PMPP) stabilization prior the acquisition of the monthly I-V curves for the DEM2 left at V_{oc} in outdoor conditions; **b)** daily irradiance for the days selected during the outdoor test for the acquisition of the I-V curves.

In addition, there are some problems with the article as follows, so I don't recommend this manuscript for publication in Nature Communications.

Q1) Researchers have already revealed that 4-FPEAI and MXene play important roles in improving the performance of perovskite solar cells. The highlights and novelty of this work should be highlighted.

We thank the reviewer for this important observation. We fully agree that both 4-FPEAI and MXene materials have been individually explored in previous studies and shown to improve the performance of perovskite solar cells through passivation and work function tuning, respectively.

However, to the best of our knowledge, this is the first work that strategically combines both 4-FPEAI and MXene within the same perovskite absorber to intentionally create a graded homojunction with a built-in n-to-p electronic gradient. This approach allows for spatial modulation of the energetic landscape across the absorber thickness, thereby reducing recombination and enhancing charge transport — a mechanism that goes beyond the individual contributions of each additive.

We have now revised the manuscript to more clearly emphasize this point, both in the Introduction and the Conclusions, and to better articulate how our integration of materials and device architecture represents a step forward in perovskite solar cell engineering.

“Here we propose a novel approach to fabricate graded perovskite homojunctions from an n-type character at the buried perovskite layer, induced by chlorine-based MXene doping,² to a p-type character at the surface, enhanced by 4-FPEAI based SGP. This intentional spatial gradient forms a built-in n-to-p homojunction across the absorber, enabling more efficient carrier extraction and reduced recombination.”

“Crucially, this work introduces a vertically graded perovskite homojunction, enabled by the spatially resolved integration of Cl-terminated MXenes and 4-FPEAI, a concept not previously demonstrated, representing a new class of interface and energy landscape engineering in perovskite photovoltaics.”

Q2) In Fig. 2 and Fig.S19, the statistics of the electrical parameters are all obtained from 8 samples, which is not a sufficient amount of data to support the experimental conclusions. For example, in lines “...and exhibiting reduced PCE dispersion “ and lines 123 “... Improvements highlights enhance device reproducibility ...”, 8 samples are clearly not enough to account for these.

We appreciate the reviewer’s insightful comment regarding the statistical validity of our data set. We agree that a larger number of samples would provide stronger statistical power to support claims related to device

reproducibility and PCE dispersion. Our initial set of 8 devices was chosen based on the typical standard in preliminary studies of this nature, aiming to balance experimental throughput with material and time constraints. Nevertheless, we expanded our dataset to twelve samples and included these extended statistics in the revised manuscript accordingly by substituting the figures **Fig. 2** and **Fig. S19** and **Tab. 1** and **Tab. S6** related to opaque and semitransparent devices respectively to avoid any overgeneralized conclusions. Moreover, we have changed in the entire manuscript and SI the discussion on the results obtained accordingly.

In the main text we changed the following **Fig. 2, Tab.1**

Fig. 2: Electrical parameter statistics (12 samples) **a)** open circuit voltage (V_{oc}); **b)** short circuit current density (J_{sc}); **c)** fill factor (FF); **d)** power conversion efficiency (PCE), for the four investigated opaque PSC structures extracted by the current-voltage (I-V) characteristics acquired under 1 SUN irradiation. Forward and reverse scan J–V curves for all four opaque device architectures are provided in the Supporting Information (**Fig. S7 a), b), c), d)**). The minimized hysteresis in the G/MX4-FPEAI device highlights the improved stability and reduced interfacial recombination enabled by the device engineering via graded homojunction strategy.

Tab. 1: Electrical photovoltaic parameters for the four investigated opaque PSCs structures extracted by the I-V characteristics acquired under 1 SUN irradiation in reverse scan reported as averaged values \pm standard error obtained on 12 samples for each cell typology and for the best performing devices.

Device		$V_{oc}(V)$	$J_{sc}(mA/cm^2)$	FF(%)	PCE(%)
CB	Champion	1.17	20.55	79.16	18.87
	Average	1.166 ± 0.001	20.22 ± 0.08	78.53 ± 0.11	18.51 ± 0.07
EA	Champion	1.176	20.79	79.76	19.19
	Average	1.170 ± 0.001	20.36 ± 0.08	79.23 ± 0.09	18.87 ± 0.08
4-FPEAI	Champion	1.193	21.30	80.05	20.18
	Average	1.190 ± 0.0006	21.03 ± 0.04	79.78 ± 0.09	19.95 ± 0.04
MX_4-FPEAI	Champion	1.205	22.5	81.82	22.12

	Average	1.197±0.001	22.05±0.09	80.86±0.19	21.34±0.13
--	---------	-------------	------------	------------	------------

While in the supporting Info the following Fig. S19, Tab. S6:

Fig. S19: Electrical parameter statistics (12 samples) for the four investigated ST-PSC structures extracted by the current-voltage (I-V) characteristics acquired under 1 SUN irradiation.

Tab. S6: Electrical photovoltaic parameters for the four investigated ST-PSCs structures extracted by the I-V characteristics acquired under 1 SUN irradiation in reverse scan reported as averaged values ± standard error obtained on 12 samples for each cell typology and for the best performing devices.

Device		Voc(V)	Jsc(mA/cm ²)	FF(%)	PCE(%)
CB	Champion	1.191	19.5	74.04	16.77
	Average	1.179±0.0019	18.81±0.11	72.5±0.33	16.09±0.10
EA	Champion	1.199	19.42	74.45	17.15
	Average	1.191±0.0017	19.01±0.092	72.93±0.28	16.50±0.10
4-FPEAI	Champion	1.21	20.42	74.4	17.60
	Average	1.196±0.003	19.54±0.10	73.55±0.16	17.19±0.04
MX_4-FPEAI	Champion	1.222	20.51	75.47	18.29
	Average	1.203±0.002	19.92±0.08	73.60±0.29	17.63±0.067

Q3) In lines 124, “;Furthermore, replacing CB with to EA, the perovskite post-deposition annealing time was reduced by half (30 minutes), effectively halving the energy required for perovskite absorber realization” Is there any experiment or reference that proves it, I don't understand.

We thank the reviewer for this question. We would like to clarify that the statement refers to a direct and experimentally observed difference in the annealing time required to fully crystallize the perovskite film, depending on the anti-solvent used during deposition.

In our fabrication process, as specified in device fabrication procedure reported in the experimental section, when chlorobenzene (CB) is used as anti-solvent, a post-deposition annealing time of 60 minutes at 100°C is necessary to achieve complete perovskite crystallization. However, when ethyl acetate (EA) is used, the same crystallization quality is achieved within only 30 minutes of annealing under identical conditions. This reduction in processing time was reported usually in literature considering EA's lower boiling point (77 °C) and moderate polarity lead to faster solvent evaporation and more rapid film formation compared to heavier aromatic anti-solvents like CB,⁴⁴ and consistently observed and confirmed across multiple samples, as reported in our manuscript.

Our intent was to highlight this clear and repeatable experimental outcome, which has a direct implication on process efficiency and energy consumption, since halving the annealing time reduces the energy required during this thermal step. We considered this a straightforward consequence of the processing condition change, but we appreciate the opportunity to make it clearer in the revised text and have added in the experimental section the following sentence for improved clarity.

It is important to note that, given its lower boiling point and optimal polarity, ethyl acetate promotes faster crystallization and more efficient solvent removal during film formation, which enables shorter annealing times compared to chlorobenzene under identical thermal conditions.⁴⁴

(44) Podapangi, S. K.; Jafarzadeh, F.; Mattiello, S.; Korukonda, T. B.; Singh, A.; Beverina, L.; Brown, T. M. Green Solvents, Materials, and Lead-Free Semiconductors for Sustainable Fabrication of Perovskite Solar Cells. *RSC Adv.* **2023**, *13* (27), 18165–18206. <https://doi.org/10.1039/d3ra01692g>.

Q4) Lines 140 “...with $Ti_3C_2Cl_2$ MXenes (MX-Cl) by carefully tuning the MXene dopant amount (Fig. S2).” In Fig. S2, what are the units of MX-Cl concentration? The interval between 0.016 and 0.15 is too large, are there conditions for better concentrations in between?

Thank you for your valuable feedback. Unfortunately, the concentration units for MX-Cl were inadvertently omitted due to image scaling, and we have now reinserted them in the updated version of **Fig. S2**. Additionally, in response to your comment about the large interval between 0.016mg and 0.15mg, we have performed a new experiment to include two new data points at 0.02mg and 0.05mg respectively to better capture the trend in that range. The figure and corresponding caption have been revised accordingly in SI and following reported.

Fig. S2: a) Photograph of one of the two glass substrates used for device fabrication, each hosting four small-area perovskite solar cells. A total of 12 devices were fabricated for the statistical evaluation of photovoltaic parameters (Fig. 2,) while 8 devices were used for the optimization of the Mxene-Cl doping concentration, b) PCE of PSCs realized by varying the amount of MXene-Cl added to 1 ml of perovskite precursor solution.

Q5) Lines 146 and 147, “SGP has minimal impact on grain size and surface morphology (Fig. S8), whereas MX-Cl doping, resulted in a noticeable increase in perovskite grain size compared to the pristine layer (Fig. 3a and 3c)”. Statistical distribution and comparison of average grain size should be given.

We thank the reviewer for his constructive suggestion. In response, we added a detailed statistical analysis of the domain size distribution for both the pristine 4-FPEAI-modified perovskite film and the film incorporating Cl-functionalized MXenes, as now shown in the updated Fig. 3 (panels b and e). This analysis clearly demonstrates a significant increase in average domain size from $0.125 \mu\text{m}$ ($\sigma = 0.047 \mu\text{m}$) in the 4-FPEAI only sample to $0.286 \mu\text{m}$ ($\sigma = 0.070 \mu\text{m}$) in the MXene/4FPEAI sample. Furthermore, the maximum domain size increases from $0.242 \mu\text{m}$ to $0.452 \mu\text{m}$, indicating more pronounced domain growth induced by the presence of MXenes. These findings support our claim that MX-Cl doping leads to noticeable domain enlargement and morphological evolution, consistently with enhanced crystallization and improved film quality. The domain size statistics provide quantitative confirmation of the morphological differences initially observed in the SEM images (panels a and d) and reinforce the interpretation of improved film properties upon MXene incorporation.

The modified Fig. 2 with the new panel b) and e) and the relative caption are reported in the following

Fig. 3: a) SEM image of a) perovskite layer without MX-Cl addition and d) MX-Cl doped perovskite film surface obtained by applying the SGP strategy during the anti-solvent step. The Cross-section SEM images for the same samples are reported in panel c) and f) respectively. In the colored part of panel f) the incorporation of MX-Cl flakes is highlighted with a light-red color showing that MX flakes are sitting at the mTiO₂/perovskite interface, while they are not present in the case of pristine perovskite film in panel c). The presence of small domains still appearing around the enlarged domains in the case of MX-Cl-doped film elucidated that the presence of MXene flakes induced a localized effect on the perovskite morphology, acting as a template for the perovskite crystals growing on top of them. The domain size distribution reveals significant differences between the perovskite modified with only 4FPEAI and that one treated with both 4FPEAI and MXenes. In the sample containing only 4FPEAI, the average domain size is 0.125 μm, with a standard deviation of 0.047 μm, while in the MXene/4FPEAI sample the mean increases to 0.286 μm, with a standard deviation of 0.070 μm. Furthermore, the maximum domain size rises from 0.242 μm (4FPEAI) to 0.452 μm (MXene/4FPEAI), indicating a more pronounced domain growth. These results suggest that the incorporation of MXenes promotes more efficient crystal growth and greater grain coalescence, which is consistent with a reduction in grain boundary defect density and potentially improved charge transport within the film. g) XRD patterns for the perovskite films with and without MX-Cl, h) Normalized transient photoluminescence (TRPL) decay curves of the 4-FPEAI perovskite film and the film incorporating Cl-functionalized MXene. The decay profiles were fitted with a bi-exponential model. The MXene-treated film exhibits a significantly longer average carrier lifetime ($\tau_{avg} = 1.11 \times 10^{-7}$ s) compared to the 4-FPEAI ($\tau_{avg} = 6.88 \times 10^{-8}$ s), indicating reduced trap-assisted recombination and improved defect passivation.

Q6) Lines 162 “XRD analysis (Fig. 3e) reveals ... and minimizing trap states” Enhancement of the XRD peak intensity does not necessarily directly indicate a reduction in defects in the film, and enhancement of the diffraction peaks may reflect increased crystallinity or increased grain size, which corresponds to SEM. Increased grain size results in sharper peaks and increased intensity, but this may be directly related to a reduction in grain boundaries rather than point defects. A reduction in point defects or dislocations may reduce lattice distortion, resulting in increased peak intensity, but changes in defect density need to be confirmed in conjunction with other means such as TEM and SCLC.

We agree with the reviewer that an increase in XRD peak intensity, while indicative of enhanced crystallinity or increased domain size, does not alone confirm a reduction in defect density, especially point defects. To further validate our interpretation regarding defect passivation, we have complemented the XRD analysis with transient photoluminescence (TRPL) measurements, which provide insights into charge carrier dynamics and trap-assisted recombination processes. The TRPL decay curves, reported in the new Fig. 3h, were fitted

using a bi-exponential model, and the extracted lifetimes are summarized in the table below reported in SI as **Tab. S2**. For the sample incorporating 4-FPEAI/MX-Cl, the average carrier lifetime (τ_{avg}) is 1.11×10^{-7} s, notably longer than the 6.88×10^{-8} s observed for the sample with only 4-FPEAI. The fast decay component (τ_1), often associated with trap-assisted recombination, also shows a significant increase from 4.16×10^{-8} s (REF) to 7.57×10^{-8} s (4-FPEAI/MX-Cl), accompanied by a larger amplitude (A_1), suggesting that traps are being effectively passivated in the Mxene-modified film. These results support the hypothesis that the incorporation of Cl-functionalized Mxenes leads to a reduction in non-radiative recombination pathways, likely due to trap state passivation. This improvement in carrier dynamics corroborates the increased crystallinity observed in XRD and the enhanced morphology seen in SEM, thereby offering a more holistic understanding of the film quality improvements.

We added the panel h) in **Fig. 3** by adapting the caption and the following text in the main:

Complementarily, TRPL analysis (**Fig. 3h**) shows a marked increase in average carrier lifetime in MXene-modified films, pointing to efficient trap-state passivation, likely due to both improved crystallinity and reduced interfacial defect density (see **Tab. S2** in the SI for full fitting parameters).

Fig. 3: a) SEM image of a) perovskite layer without MX-Cl addition and d) MX-Cl doped perovskite film surface obtained by applying the SGP strategy during the anti-solvent step. The Cross-section SEM images for the same samples are reported in panel c) and f) respectively. In the colored part of panel f) the incorporation of MX-Cl flakes is highlighted with a light-red color showing that MX flakes are sitting at the mTiO₂/perovskite interface, while they are not present in the case of pristine perovskite film in panel c). The presence of small domains still appearing around the enlarged domains in the case of MX-Cl-doped film elucidated that the presence of MXene flakes induced a localized effect on the perovskite morphology, acting as a template for the perovskite crystals growing on top of them. The domain size distribution reveals significant differences between the perovskite modified with only 4FPEAI and that one treated with both 4FPEAI and MXenes. In the sample containing only 4FPEAI, the average domain size is 0.125 μm, with a standard deviation of 0.047 μm, while in the MXene/4FPEAI sample the mean increases to 0.286 μm, with a standard deviation of 0.070 μm. Furthermore, the maximum domain size rises from 0.242 μm (4FPEAI) to 0.452 μm (MXene/4FPEAI), indicating a more pronounced domain growth. These results suggest that the incorporation of MXenes promotes more efficient crystal growth and greater grain coalescence, which is consistent with a reduction in grain boundary defect density and potentially improved charge transport within

the film. g) XRD patterns for the perovskite films with and without MX-Cl, h) Normalized transient photoluminescence (TRPL) decay curves of the 4-FPEAI perovskite film and the film incorporating Cl-functionalized MXene. The decay profiles were fitted with a bi-exponential model. The MXene-treated film exhibits a significantly longer average carrier lifetime ($\tau_{\text{avg}} = 1.11 \times 10^{-7}$ s) compared to the 4-FPEAI ($\tau_{\text{avg}} = 6.88 \times 10^{-8}$ s), indicating reduced trap-assisted recombination and improved defect passivation.

While in the supporting info in SECTION S.I. 5 we added the following text and table:

To further evaluate the influence of Cl-functionalized MXenes on the optoelectronic properties of the perovskite films, we performed time-resolved photoluminescence (TRPL) measurements. The decay profiles of the reference and MXene-treated films were fitted using a bi-exponential decay model:

$$F(t) = A_1 e^{-t/\tau_1} + A_2 e^{-t/\tau_2}$$

where τ_1 and τ_2 represent the fast and slow decay lifetimes, respectively, and A_1 , A_2 are the corresponding amplitudes. The fast decay component (τ_1) is commonly associated with trap-assisted recombination processes, while the slow component (τ_2) reflects radiative recombination of free carriers.

As shown in **Tab. S2**, the MXene-treated film shows a longer average carrier lifetime ($\tau_{\text{avg}} = 111.5$ ns) compared to the reference (68.76 ns). The fast decay time τ_1 also increases from 41.63 ns to 75.67 ns, indicating fewer shallow trap states. This reduction is likely due to larger domain sizes induced by MXene flakes acting as localized crystallization templates, which decrease the overall grain boundary density and associated trap sites. The amplitude A_1 also increases significantly (from 3.16 to 9.127), further suggesting a dominant role of non-radiative pathways in the control film that are effectively passivated by the MXene layer.

These findings corroborate the beneficial role of Cl-functionalized MXenes in reducing trap-assisted recombination by passivating defect sites—likely through interaction between Cl^- terminations and undercoordinated Pb^{2+} ions. The TRPL results align with structural improvements observed via XRD and SEM, further confirming that the incorporation of MXenes enhances the film's crystallinity, reduces trap densities, and improves charge carrier dynamics.

Tab. S2: Fitting parameters extracted from the bi-exponential fitting of TRPL decay curves for the reference and MX-Cl modified perovskite films. A_1 and A_2 are the amplitudes of the fast and slow decay components, respectively; τ_1 corresponds to trap-assisted (non-radiative) recombination, and τ_2 to radiative recombination. τ_{avg} is the average carrier lifetime calculated as $\tau_{\text{avg}} = (A_1\tau_1^2 + A_2\tau_2^2) / (A_1\tau_1 + A_2\tau_2)$. The increased τ_1 and τ_{avg} in the MXene-modified film suggest reduced trap density and suppressed non-radiative recombination.

Sample	A_1	τ_1 (ns)	A_2	τ_2 (ns)	τ_{avg} (ns)
4-FPEAI	3.617	41.63	0.157	199.3	68.76
MX-Cl/4-FPEAI	9.127	75.67	1.327	203.2	111.5

Q7) The peak intensity ratios in the XRD pattern in Fig.3e clearly do not correspond to the values in section SI 5, Tab. S1, e.g., the ratio of 4-FPEAI (110)/(310) is clearly greater than 1 in the figure, but 0.92 in the table.

Thank you for your careful observation. You are absolutely right, the peak intensity ratios reported in **Tab. S1** (Section SI 5) were affected by a formatting error, which led to the inversion of some values. Specifically, the (110)/(310) ratio for 4-FPEAI was mistakenly listed as lower than 1. We have now corrected the table with the accurate values, which are consistent with the XRD pattern shown in **Fig. 3e**. We appreciate your attention to detail and apologize for the oversight.

	(110)/(310)	(220)/(310)
4-FPEAI	1.62	0.92
MX-Cl/4-FPEAI	2.29	1.31

Q8) In lines 179, “...slight blue shift (767 nm vs. 763 nm) for MX-Cl/4-FPEAI films...”; the optical band gap of $Cs_{0.18}FA_{0.82}Pb(I_{0.8}Br_{0.2})_3$ perovskite was 1.68 eV in Fig.S1, and the band gap is calculated to be 1.62 eV (~760 nm) based on the emission peaks of PL. Does the introduction of 4-FPEAI and MX-Cl change the band gap of perovskite?

We are grateful to the reviewer for underlining this aspect that gives us the opportunity to repeat both absorption and PL spectra analysis, newly acquired on fresh perovskite layers on glass. In particular, we made 4-FPEAI treated perovskite film (named 4-FPEAI) and MXene-doped perovskite film treated with 4-FPEAI (named MX-Cl/4-FPEAI). The band gap (E_g) extracted by the Tauc plots reported in **Fig. S10a** was 1.67 eV for 4-FPEAI film while 1.667 eV for MX-Cl/4-FPEAI film. The slight blue-shift induced by the MXene doping can be ascribed to the enlarged dimension of the perovskite domain size.^{3,4} Moreover, by comparing the Tauc plot of the bare perovskite film reported in SI and that one of 4-FPEAI sample, it's quite evident a slight E_g blue shift induced by the 4-FPEAI treatment, shifting the E_g from 1.68 to 1.67 eV. A similar effect has been already observed in previous publications when using fluorinated aromatic cation-based treatment as passivation strategy for 3D perovskite film.⁵⁻⁷ On the same samples, we acquired PL spectra on three different points of the films. The PL spectra reported in **Fig. S10b** showed averaged emission peaks centered at 758 nm and 763 nm for 4-FPEAI and MX-Cl/4-FPEAI samples respectively. The red-shift in PL observed upon the incorporation of MXenes can be attributed to the growth of larger perovskite domains. This morphological evolution reduces the density of trap states at grain boundaries, thereby significantly enhancing the PL intensity.

- (3) Ummadisingu, A.; Meloni, S.; Mattoni, A.; Tress, W.; Grätzel, M. Crystal-Size-Induced Band Gap Tuning in Perovskite Films. *Angew. Chemie - Int. Ed.* **2021**, *60* (39), 21368–21376. <https://doi.org/10.1002/anie.202106394>.
- (4) Chen, L.; Paillard, C.; Zhao, H. J.; Íñiguez, J.; Yang, Y.; Bellaiche, L. Tailoring Properties of Hybrid Perovskites by Domain-Width Engineering with Charged Walls. *npj Comput. Mater.* **2018**, *4* (1). <https://doi.org/10.1038/s41524-018-0134-3>.
- (5) Zhou, Q.; Liang, L.; Hu, J.; Cao, B.; Yang, L.; Wu, T.; Li, X.; Zhang, B.; Gao, P. High-Performance Perovskite Solar Cells with Enhanced Environmental Stability Based on a (p-FC 6 H 4 C 2 H 4 NH 3) 2 [PbI 4] Capping Layer. *Adv. Energy Mater.* **2019**, *9* (12), 1–11. <https://doi.org/10.1002/aenm.201802595>.
- (6) Zhou, L.; Su, J.; Lin, Z.; Guo, X.; Ma, J.; Li, T.; Zhang, J.; Chang, J.; Hao, Y. Synergistic Interface Layer Optimization and Surface Passivation with Fluorocarbon Molecules toward Efficient and Stable Inverted Planar Perovskite Solar Cells. *Research* **2021**, *2021*. <https://doi.org/10.34133/2021/9836752>.
- (7) Xu, J.; Boyd, C. C.; Yu, Z. J.; Palmstrom, A. F.; Witter, D. J.; Larson, B. W.; France, R. M.; Werner, J.; Harvey, S. P.; Wolf, E. J.; et al. Triple-Halide Wide-Band Gap Perovskites with Suppressed Phase Segregation for Efficient Tandems. *Science (80-.).* **2020**, *367* (6482), 1097–1104. <https://doi.org/10.1126/science.aaz4639>.
- (8) Campanari, V.; Martelli, F.; Agresti, A.; Pescetelli, S.; Nia, N. Y.; Giacomo, F. Di; Catone, D.; Keeffe, P. O.; Turchini, S.; Yang, B.; et al. Reevaluation of Photoluminescence Intensity as an Indicator of Efficiency in Perovskite Solar Cells. *Sol. RRL* **2022**, *6* (8), 2200049–2200059. <https://doi.org/10.1002/solr.202200049>.

We modified the SI accordingly, that now sounds as follows reported:

The enlarged perovskite domain size translates in a slight reduction of the perovskite band gap, shifting from 1.67 eV for perovskite films with SGP (named 4-FPEAI) to 1.667 in case of same film but with the addition of MX-Cl (named MX-Cl/4-FPEAI), as reported in **Fig. S10a**. Moreover, the beneficial role of MX-Cl in reducing the charge recombination is confirmed by the steady state photoluminescence (PL) measurements performed on both 4-FPEAI and MX-Cl/4-FPEAI films (**Fig. S10b**). Regarding PL spectra, generally a reduced PL emission is imputed to the non-radiative recombination acting in the film, while the presence of a red-shift emission can be ascribed to an enlargement of the averaged perovskite domain size.⁸ Both phenomena can be observed in **Fig. S10b** where in the case of optimized MX-Cl/4-FPEAI a higher PL emission intensity (+75.1%) is recorded together with a slightly red shift of the emission peak (763 nm) once compared with 4-FPEAI device (758nm).

Fig. S10: a) Tauc plots and b) steady-state PL spectra for perovskite films with SGP (named 4-FPEAI) and same film but with the addition of MX-Cl (named MX-Cl/4-FPEAI). PL spectra were acquired on three different points of the samples (named _1, _2, _3).

Q9) In lines 189, “...the hydrophobic nature of MX-Cl further boosts both device efficiency and operational stability, ...” There's no pristine group here in Fig.S12. And a 3-minute mpp tracking is too short to indicate stability. Contact angle and surface tension calculations are needed to show that MXene doping leads to an increase in the hydrophobicity of perovskite films.

We thank the reviewer for his insightful observation. Verifying the effect of MX-Cl on the hydrophobic nature of the perovskite surface is not a trivial task, since MXenes flakes are sitting at the mTiO₂/perovskite interface. Therefore, we performed XPS measurements considering O1s peaks for the ad-hoc samples already realized for UPS now reported in the revised version of the SI. As evident from Fig. R2, two bands are detected, one around 532.8 eV, related to the oxygen present in the adsorbed water, and one around 532.4 eV, very close to the first, which can be attributed to C=O groups.¹ In particular, the measurements were performed on sample named 4-FPEAI (4-FPEAI treated perovskite film with a reduced thickness of around 200 nm growth on mTiO₂ scaffold) and sample named MX-Cl/4-FPEAI (4-FPEAI treated perovskite film with MX-Cl with a reduced thickness around 200 nm growth on mTiO₂). The two samples showed pretty similar percentage of adsorbed water onto the surface, (see Tab.R1) confirming the minor role of the MX-Cl in improving the hydrophobicity of the perovskite film surface.

Fig. R2: XPS measurements O1s peaks for a) sample A1 (4-FPEAI treated perovskite film with a reduced thickness of around 200 nm growth on mTiO₂ scaffold) and b) sample B1 (4-FPEAI treated perovskite film with MX-Cl with a reduced thickness around 200 nm growth on mTiO₂).

Sample	Adsorbed water	C=O
Position (eV) / relative amount		
4-FPEAI	532.80/82.2%	532.47/17.8%
MX-Cl/4-FPEAI	533.14/81.5%	532.43/18.5%

Tab. R1 Adsorbed water values a) sample named 4-FPEAI (4-FPEAI treated perovskite film with a reduced thickness of around 200 nm growth on mTiO₂ scaffold) and b) sample named MX-Cl/4-FPEAI (4-FPEAI treated perovskite film with MX-Cl with a reduced thickness around 200 nm growth on mTiO₂).

(1) Larciprete, R.; Agresti, A.; Pescetelli, S.; Pazniak, H.; Liedl, A.; Lacovig, P.; Lizzit, D.; Tosi, E.; Lizzit, S.; Carlo, A. Di. Mixed Cation Halide Perovskite under Environmental and Physical Stress. *Materials (Basel)*. **2021**, *14*, 3954 (1-20)

For this reason, on one side we modify the main text by removing the sentence related to the “hydrophobic nature of MX-Cl”. On the other side, with the aim to deeply investigate the improved stability of our graded homojunction, we performed cycling light test (ISOS-C1), following the recently established stressing protocol.¹⁸ The results are reported and commented on SI. The main modifications carried out on both the main text of the manuscript and the SI are reported following for the reviewer convenience.

In SI: **Indoor photo-stability under MPP tracking of semi-transparent encapsulated cells upon exposure to cycled light.** As recently discussed in literature, the most effective way to perform a reliable stability analysis consists in light cycling test at constant temperature, representing a good trade-off between simplicity of execution in lab conditions while taking into account phenomena observed only in case of outdoor testing (that cannot be considered in case of constant light experiments).¹⁵ Indeed, transient behavior occurs in PSCs as a result of the coexistence of several dynamics with characteristic times spanning from time scales of seconds to hours. Thus, day–night cycling of devices can consider such slow dynamics as they occur on similar timescales. Following this idea, we performed a light–dark cycling test as recommended by the ISOS-LC-1 protocol, exposing the cells to simulated sunlight turned on and off with cycle periods of 24 h and duty cycles (light:dark) of 1:2, mimic an accelerated diurnal sun cycle (the solar cell is maintained at ambient conditions while the temperature is monitored and RH is maintained at 30%).¹⁶ The recorded maximum power (P_{MPP}) is reported in **Fig. S7e** for both G/EA and fully optimized G/MX/4-FPEAI semi-transparent cells, normalized at the initial value. The combined stabilization effect of the 4-FPEAI based SGP together with the presence of MX-Cl results in a T_{91} =600 hours vs a T_{80} =574 hours of the untreated device, demonstrating the superior stability (T_{91} 6.5 times larger) of the device employing the proposed graded homojunction approach. Despite more in depth analyses are required to elucidate the role of MX-Cl and 4-FPEAI SGP, we can confidently ascribe the improved stability to a synergistic effect of the two treatments. On one side, the addition of MX-Cl resulted in a perovskite film with improved crystallinity eventually reducing lattice distortion, lowering defect density and minimizing trap states, as demonstrated by XRD, transient PL and SEM characterizations. This, combined with the enlarged perovskite domains and mitigated grain boundary-induced trap states, makes the trapped charge driven degradation less effective,¹⁷ resulting in an overall enlargement of device lifetime. We cannot exclude a side role of the Cl in reducing the tensile strain within the perovskite layer, recently pointed out as one of the main degradation channel when light-dark cycling is applied.¹⁸ On the other side, the phenylalkylammonium compounds as 4-FPEAI has been proven to stabilize the perovskite through suppression of iodide ion migration, likely responsible for metastable perovskite behavior under light cycling observed for the G/EA device, while passivating the surface defects.¹⁹ In addition, the 4-FPEAI SGP strategy has been already proven to improve the long-term operational stability of the encapsulated devices under the continuous illumination in ambient condition.¹² Thus, the superior light cycling stability showed by the graded homojunction cell can be confidently ascribed to a combined stabilizing effect from 4-FPEAI and MX-Cl, definitively reducing the device metastable behavior.

Fig. S22: **a)** J-V curve in forward and reverse scan for the best-performing 2D material-engineered ST-PSC (G/MX/4-FPEAI structure); **b)** IPCE spectra with the integrated photocurrent density (Integrated J_{SC}) related to the best efficient optimized cell (G/MX/4-FPEAI) for both opaque and ST proposed device structure. **c)** Indoor photo-stability under MPP tracking of G/EA and G/MX/4-FPEAI semi-transparent encapsulated cells upon exposure to cycled light (ISOS-LC-1).

- (12) Yan, N.; Gao, Y.; Yang, J.; Fang, Z.; Feng, J.; Wu, X.; Chen, T.; Liu, S. F. Wide-Bandgap Perovskite Solar Cell Using a Fluoride-Assisted Surface Gradient Passivation Strategy. *Angew. Chem. Int. Ed.* **2023**, *62*, e202216668. <https://doi.org/10.1002/anie.202216668>.
- (15) Khenkin, M.; Köbler, H.; Remec, M.; Roy, R.; Erdil, U.; Li, J.; Phung, N.; Adwan, G.; Paramasivam, G.; Emery, Q.; et al. Light Cycling as a Key to Understanding the Outdoor Behaviour of Perovskite Solar Cells. *Energy Environ. Sci.* **2023**, *17* (2), 602–610. <https://doi.org/10.1039/d3ee03508e>.
- (16) Khenkin, M. V.; Katz, E. A.; Abate, A.; Bardizza, G.; Berry, J. J.; Brabec, C.; Brunetti, F.; Bulović, V.; Burlingame, Q.; Di Carlo, A.; et al. Consensus Statement for Stability Assessment and Reporting for Perovskite Photovoltaics Based on ISOS Procedures. *Nat. Energy* **2020**, *5* (1), 35–49. <https://doi.org/10.1038/s41560-019-0529-5>.
- (17) Ahn, N.; Kwak, K.; Jang, M. S.; Yoon, H.; Lee, B. Y.; Lee, J.; Pikhitsa, P. V.; Byun, J.; Choi, M. Trapped Charge-Driven Degradation of Perovskite Solar Cells. *Nat. Commun.* **2016**, *8* (May), 1–9. <https://doi.org/10.1038/ncomms13422>.
- (18) Shen, Y.; Zhang, T.; Xu, G.; Steele, J. A.; Chen, X.; Chen, W.; Zheng, G.; Li, J.; Guo, B.; Yang, H.; et al. Strain Regulation Retards Natural Operation Decay of Perovskite Solar Cells. *Nature* **2024**, *635* (8040), 882–889. <https://doi.org/10.1038/s41586-024-08161-x>.
- (19) Guo, Y.; Aperi, S.; Li, N.; Chen, M.; Yin, C.; Yuan, Z.; Gao, F.; Xie, F.; Brocks, G.; Tao, S.; et al. Phenylalkylammonium Passivation Enables Perovskite Light Emitting Diodes with Record High-Radiance Operational Lifetime: The Chain Length Matters. *Nat. Commun.* **2021**, *12* (1). <https://doi.org/10.1038/s41467-021-20970-6>.

We also modified the main text by briefly recalling the abovementioned stability analysis as in the following:

“Moreover, the combination among the MX-Cl addition and the 4-FPEAI SGP strategy not only translated in superior device performance but also conferred superior robustness to the perovskite film against light-driven degradation, as demonstrated by prolonged light cycling test (Fig. S22c).”

Finally, we inserted in ex Fig. S12 (now Fig. S7e) the stabilized MPP plot for the G/EA cells, as requested by the reviewer.

Q10) In lines 206 and 207, “...could be concluded that the combination of 4-FPEAI and MX-Cl doping forms a graded n-p junction along the thickness of the perovskite absorber from ETL to HTL side.” The variation of the work function of CPD is not a sufficient proof of the existence of n-p junction, which can be further demonstrated by calculating the energy level structure based on UPS of different films.

We thank the Reviewer for the insightful comment. We agree that the variation of the contact potential difference (CPD) alone is not sufficient to unequivocally demonstrate the formation of a graded n–p junction across the perovskite layer. Following the Reviewer’s suggestion, we have complemented our analysis with Ultraviolet Photoelectron Spectroscopy (UPS) measurements on ad hoc prepared films, allowing for a more accurate evaluation of the energy level structure and its modulation upon MX-Cl doping and 4-FPEAI incorporation.

The revised main text now includes a discussion of the UPS results, which clearly show a progressive tuning of both work function and valence band maximum values across the series of modified films. This supports

the presence of an internal electric field and energy level bending consistent with a graded n–p junction configuration. Full UPS spectra and extracted values are now reported in the Supporting Information (**Figs. S11, S12, and Tab. 2** in main text), and relevant references to these data have been added to the main manuscript.

Below we provide the specific additions made to the main text and the SI to address this point more comprehensively.

From SEM images of **Fig. 3f**, clearly MX-Cl flakes are mainly distributed at bottom interface between perovskite/mTiO₂ layer. The presence of MX-Cl at the ETL/perovskite interface calls for a possible tuning of the buried perovskite interface WF.²¹

Building on this, ultraviolet photoelectron spectroscopy (UPS), was employed to elucidate the progressive modulation of the perovskite energy levels depending on the different treatments. Notably, the combination of MXene and 4-FPEAI induces a graded shift in WF and valence band maximum (VBM) position across the series, consistent with the formation of a built-in electric field within the absorber. A full discussion and corresponding UPS spectra and energy level diagram reconstructed from the UPS data are provided in **section S6 in SI Fig. S11 and Fig. S12** respectively, while the extracted values of WF and VBM are summarized in **Tab. 2**, clearly highlighting the energetic alignment changes induced by each modification.

Tab. 2. Extracted values of work function (WF), valence band maximum (VBM) with respect to the Fermi level, and ionization energy (IE = WF + VBM) for the four analyzed perovskite film configurations, measured by UPS. The results demonstrate distinct electronic effects induced by MXene and 4-FPEAI treatments, supporting the formation of dipoles and doping gradients.

Sample	WF (eV)	VBM (eV)	(IE) [eV]
REF	4.75±0.1	0.9±0.1	5.65±0.1
MX-Cl	4.55±0.1	0.9±0.1	5.45±0.1
MX-Cl / 4-FPEAI	4.70±0.1	0.8±0.1	5.50±0.1
4-FPEAI	4.85±0.1	0.7±0.1	5.55±0.1

To confirm this effect, Kelvin probe force microscopy (KPFM) measurements were carried out on dedicated half-cell samples (FTO/G+cTiO₂/G+mTiO₂/perovskite), allowing us to extract WF values (**Tab. S3**) and distinguish the contributions of MXene doping and SGP (see **Fig. S13**)

In the experimental section we added:

Photoelectron spectroscopy measurements were performed in a ThermoScientific ESCALAB QXi XPS apparatus in ultra-high vacuum conditions (10⁻¹⁰ mbar base pressure). The valence band region of the samples was investigated by UPS using the 21.2 eV radiation of the HeI line and biasing the samples at -5 V, in order to determine both the work function and the position of the valence band edge. The electronic band structure of the perovskites was characterized by X-ray photoemission spectroscopy (XPS) using monochromatic 1486.8 eV photons of the Al K α line and analysed with the Thermo-Fischer Avantage Software. XPS measurements were performed on small devices (14x14 mm) applying the system charge compensation, while the work function was measured without charge compensation, by providing an additional contact to an edge of the device front surface.

Moreover, the complete discussion on UPS results with the relative figures (**Fig. S11 and S12**) are reported in SI

Ultraviolet photoelectron spectroscopy (UPS) measurements

To gain insight into the energetic effects induced by the surface and contact engineering strategies, we performed ultraviolet photoelectron spectroscopy (UPS) measurements on a set of ad-hoc fabricated perovskite thin films with reduced thickness (comparable to the MXene flake size, ~200 nm) and varied interfacial treatments (REF, MX-Cl, 4-FPEAI, and MX-Cl/4-FPEAI). The extracted values of work function (WF) and valence band maximum (VBM) for each cell configuration are summarized in **Tab. 2** of the main text.

As shown in **Fig. S12**, the energy level diagram reconstructed from the UPS data **Fig. S11** clearly reveals a systematic shift in both WF and VBM across the series.

The reference (REF) sample exhibits a WF of 4.75 eV and a VBM located 0.9 eV below the Fermi level (EF), corresponding to an ionization energy (IE) of 5.65 eV. Upon incorporation of $\text{Ti}_3\text{C}_2\text{Cl}_2$ MXenes (MX-Cl), the WF decreases to 4.55 eV while the VBM remains unchanged at 0.9 eV. This behavior indicates that the addition of MXenes enhances the n-type character of the perovskite, consistent with the formation of a surface dipole at the perovskite/MXene interface. Notably, this interface dipole shifts the vacuum level without affecting the perovskite band structure. Conversely, the use of 4-FPEAI as SGP induces a WF increase to 4.85 eV and a shift in the VBM to 0.7 eV below EF. These results suggest a clear modification of the perovskite electronic structure, indicating a p-type doping effect, in addition to surface dipole formation. The combined MX-Cl/4-FPEAI treatment results in intermediate values (WF = 4.70 eV, VBM = 0.8 eV), consistent with a graded band alignment across the film thickness.

These findings are corroborated by DFT simulations, which show a vacuum level shift of -0.39 eV for the MXene interface and $+0.14$ eV for 4-FPEAI, supporting the observed WF modulation. Overall, the UPS data confirm the formation of a vertical energy level gradient across the perovskite layer, driven by the two treatments acting at opposite interfaces.

Fig. S11. UPS spectra for perovskite films modified with different interface strategies: reference (a–b), MX-Cl (c–d), MX-Cl/4-FPEAI (e–f), and 4-FPEAI (g–h). Panels (a, c, e, g) show the secondary electron cutoff region used to extract the WF, while panels (b, d, f, h) display the valence band region used to determine the VBM with respect to the EF. A gradual modulation of the WF and VBM is observed across the series, highlighting the impact of dipole formation and band structure tuning induced by MXene doping and surface gradient passivation.

Fig. S12: Schematic energy level alignment of perovskite films extracted from UPS measurements for the four configurations: REF (untreated), MX-Cl, 4-FPEAI, and combined MX-Cl/4-FPEAI. The WF and VBM positions are plotted relative to the vacuum level. A progressive modulation of the energy levels is observed, with MX-Cl inducing a downward shift in WF (consistent with dipole-induced vacuum level bending) and 4-FPEAI causing an upward shift in both WF and VBM. The intermediate alignment in the combined treatment suggests the formation of a graded homojunction across the perovskite layer, promoting internal electric field formation and enhanced charge extraction.

Q11) *In lines “...confirming the effectiveness of 2D material-engineering approach even for semi-transparent structures” The cell structure changes from opaque to semi-transparent, where the only change seems to be the change of the top electrode from Au to ultrathin Au/ITO. Whereas the effect of MX-Cl/4-FPEAI is significantly diminished (the average increase of 30 mV is diminished to 11 mV), and is it possible that due to the change in work function of electrode that leads to a diminution of the effect of the n-p junction as described in the manuscript.*

We thank the reviewer for this relevant observation. To clarify this point, we performed a series of numerical simulations to assess the possible impact of the work function of the top electrode and unintentional effects during ITO deposition.

Firstly, we simulated the effect of a slightly lower work function for the Au/ITO semi-transparent electrode (5.18 eV) compared to Au (5.2 eV). The results indicate that this small variation primarily affects the fill factor (FF), while the values of V_{OC} and J_{SC} remain essentially unchanged. Therefore, we believe that the change in WF alone cannot explain the reduced ΔV_{OC} .

Second, we considered the possible impact of unintentional doping of the PTAA hole transport layer during ITO sputtering. By simulating an increased acceptor doping density in the semi-transparent devices, we observed that V_{OC} in the reference cell is significantly improved, while the gain in V_{OC} in the optimized (MX-Cl/4-FPEAI-treated) structure is less pronounced. This can explain the smaller ΔV_{OC} observed for the semi-transparent devices, which results from a raised baseline V_{OC} in the untreated sample, rather than a reduced effectiveness of the MX-Cl/4-FPEAI interface.

Moreover, we analyzed the statistical distribution of ΔV_{OC} and ΔJ_{SC} values across multiple devices. The broader spread in ΔV_{OC} observed for the semi-transparent devices is correlated with a broader ΔJ_{SC} distribution, as expected from the logarithmic relationship between these quantities. For this reason, comparing average ΔV_{OC} values may be misleading. A more reliable metric is the direct comparison between the top-performing devices, which still shows a consistent ΔV_{OC} improvement (29 mV for opaque vs. 23 mV for semi-transparent).

In conclusion, the observed variation in ΔV_{OC} does not indicate a diminished effectiveness of the n–p junction strategy. Rather, it reflects the marginal influence of fabrication-related effects—such as unintentional doping and electrode modification—on device performance. We have added these insights and updated **Figs. S20–S21** in the revised Supporting Information. For the reviewer’s convenience, we also report the discussion and figures directly below.

To further understand the observed variation in V_{OC} between reference and optimized ST devices, we provide here an extended discussion based on simulation results and interface analysis. The **Fig. S19** shows the statistical distribution of the main photovoltaic parameters, highlighting a reduced ΔV_{OC} improvement in ST structures compared to opaque ones. In the following, we examine possible physical origins of this behaviour, including the effect of the top contact work function, unintentional doping of the PTAA layer during ITO deposition, and the role of interfacial engineering. The analysis supports the hypothesis that the reduced ΔV_{OC} is not due to a diminished effectiveness of the MX-Cl/4-FPEAI treatment, but rather to changes induced by the device architecture and processing conditions.

Effect of the work function: due to the use of Au/ITO semitransparent contact instead of Au, the effective work function (WF) of the anode could be slightly lower for the semitransparent cell with respect to the opaque cell. In order to investigate this effect, we repeated the simulations for the reference and optimized devices varying only the effective WF value employed for the p-contact, namely, WF = 5.2 eV for the opaque cell and WF = 5.18 eV for the semitransparent structure. The results are shown in **Fig. S20a**), where the cyan solid and dashed lines represent respectively the J-V characteristics of the opaque reference and optimized devices, while the red solid and dashed lines depict the J-V curves of the semitransparent reference and optimized devices, respectively. We can see that the role of the work function is reflected by the fill factor (FF) features, giving a smaller FF when a smaller WF is considered, while J_{SC} and V_{OC} are not affected.

Effect of an unintentional doping: a possible effect induced by the sputtering process employed to deposit the ITO layer at the p-contact is an unintentional doping of the PTAA hole transport layer (HTL). We analyzed the impact of such unintentional doping comparing the simulation results of the reference and optimized devices with the behavior obtained for the same structures but with a higher acceptor doping density in the HTL, namely, $N_{A,PTAA} = 1 \times 10^{17} \text{ cm}^{-3}$ for the opaque cell and $N_{A,PTAA} = 1.5 \times 10^{17} \text{ cm}^{-3}$ for the semitransparent cell. The derived J-V characteristics are depicted in **Fig. S20b**), where the cyan solid and dashed lines represent respectively the J-V characteristics of the opaque reference and optimized devices, while the green solid and dashed lines depict the J-V curves of the semitransparent reference and optimized devices, respectively. We can see that V_{OC} value of the reference device is significantly increased by the unintentional doping while the increment of V_{OC} in the optimized structures is less pronounced, resulting in a smaller improvement of V_{OC} for the semitransparent cell with respect to the opaque cell, namely, $\Delta V_{OC} = 32 \text{ mV}$ for the opaque contact and $\Delta V_{OC} = 25 \text{ mV}$ for the semitransparent contact.

The more pronounced effect of unintentional PTAA doping on the reference device can be attributed to the initial less favorable energy level alignment and interfacial contact properties at the perovskite/HTL interface. In the absence of interfacial modifiers such as MX-Cl/4-FPEAI, the reference structure exhibits a higher energy offset for hole extraction and a greater density of interfacial recombination pathways. Increasing the acceptor concentration in PTAA enhances the built-in electric field and improves hole-transporting efficiency, which facilitates charge extraction and reduces interfacial recombination losses, resulting in a notable V_{OC} increase. In contrast, the optimized device incorporating MX-Cl and 4-FPEAI, already features tailored interfacial energetics and improved chemical passivation. These interlayers contribute to more efficient charge extraction and reduced defect-mediated recombination. As a result, additional doping of PTAA has a comparatively minor effect on V_{OC} , since the contact is already near optimal.

a)

b)

Fig. S20: a) effect of a smaller effective work function value of the p-contact. b) effect of a higher acceptor doping density in the hole transport layer.

Discussion on the statistics of ΔV_{OC} and ΔJ_{SC} : we want to point out that, if the top cells are compared, the improvement in V_{OC} of the semitransparent cell is comparable with the value obtained for the opaque cell, namely, $\Delta V_{OC} = 29$ mV for the opaque structure and $\Delta V_{OC} = 23$ mV for the semitransparent structure. Considering the known relation existing between ΔV_{OC} and ΔJ_{SC} , *i.e.*, $V/V_0 \sim \ln(J/J_0)$, the fluctuations in the observed values of ΔV_{OC} are related to the fluctuations in the increment of J_{SC} , as suggested by the figure below (**Fig. S21**), where the distributions of the improvement of V_{OC} (**Fig. S21a**) and J_{SC} (**Fig. S21b**) features for the opaque and semitransparent structures are shown. So, in our opinion, it may be misleading the comparison of the average improvement in V_{OC} , since the width of the ΔV_{OC} distribution of the semitransparent device is larger with respect to the opaque device, due to the related broader ΔJ_{SC} distribution, and the specific correlation between ΔV_{OC} and ΔJ_{SC} would be lost. We think that, at this stage, the specific comparison between the top cells in the two sets of samples is more reliable. However, this analysis highlights the importance of achieving major control on the deposition process of the ITO layer. Such a strategy could be addressed in our future work.

a)

b)

Fig. S21: distribution of **a) $\Delta V_{OC} = V_{OC_optimized} - V_{OC_reference}$** and **b) $\Delta J_{SC} = J_{SC_optimized} - J_{SC_reference}$** for the opaque (blue) and semitransparent (red) structures.

And we recall this discussion in the main text as in the following indicate:

By employing the same 2D material-engineering strategies, MXene doping and 4-FPEAI-based surface gradient passivation, previously validated in opaque devices, the semi-transparent perovskite top cells used in the 4T tandem architecture demonstrated enhanced PCEs up to 18.3% (av. PCE = 17.63%) due to a higher V_{OC} (av. $V_{OC} = 1.2$ V) and J_{SC} (av. $J_{SC} = 19.92$ mA/cm²) compared to the REF sample (see **Tab. S6, Figs. S19–22a, S20 and S21** for a detailed analysis), confirming the effectiveness of 2D material-engineering approach even for semi-transparent structures. Moreover, the combination among the MX-Cl addition and the 4-FPEAI SGP strategy not only translated into superior device performance but also conferred superior robustness on the perovskite film against light-driven degradation, as demonstrated by prolonged light cycling test (**Fig. S22c**).

Reviewer's Comments:

Reviewer #2 (Remarks to the Author)

The authors have addressed all previous comments and therefore publication of the manuscript is agreed.

We are grateful for the Reviewer's positive recommendation and for the thoughtful suggestions that strengthened our work.

Reviewer #3 (Remarks to the Author)

The authors have made a significant effort to revise the manuscript with additional experiment and discussion. While there have been improvements in many areas, I still have serious concerns about the graded homojunction claim. Considering how this is central to the proposed mechanisms in this paper, I cannot support publication this claim included. I'll discuss in more detail below.

Response:

We thank the Reviewer for the careful re-evaluation of our revised manuscript and for the constructive comments. Below we provide a detailed point-by-point response by further clarifying the key experimental evidences supporting the formation of a vertically graded homojunction. In the revised version, we have strengthened the discussion, expanded the Supporting Information, and clearly articulated the mechanistic role of MX-Cl and 4-FPEAI at their respective interfaces.

Concern 1: *UPS is highly surface sensitive (probe depth ~1-2nm), but there is no discussion on how samples were handled or if they ever saw air. These details are important to the interpretation of the results. For example, differences in hydrophobicity of samples surfaces (e.g. with 4-FPEAI treatment) can alter water uptake. In addition, the 4-FPEAI treatment is additive to the surface and therefore the measurement may not be representative of the perovskite band structure.*

Answer 1: We thank the Reviewer for raising this important point, which gives us the opportunity to further clarify our experimental procedure. All UPS samples were handled under strictly controlled inert conditions. All samples were transported using a sealed transfer container to move the sample from the glovebox to the UPS laboratory without exposure to ambient atmosphere. In particular, for each specimen the sealed container was opened only immediately before mounting the sample on the manipulator holder, and the sample was then rapidly introduced into the analysis chamber through a fast-entry load lock, ensuring less than 10 s exposure to ambient air. Under these conditions, surface contamination or water adsorption is minimized and cannot significantly affect the measured work-function and VBM values.

As already discussed in our previous revision, assessing the effect of MX-Cl on the hydrophobic character of the perovskite surface is not straightforward, since MXene flakes are located at the buried m-TiO₂/perovskite interface. For this reason, we performed dedicated XPS measurements of the O 1s region for the same samples ad-hoc fabricated (4-FPEAI and MX-Cl/4-FPEAI, both fabricated with ~200 nm perovskite thickness on m-TiO₂).

As shown in the revised **Fig. S12**, all samples exhibit two contributions in the O 1s region:

- ~532.8–532.9 eV, corresponding to adsorbed water;
- ~532.4–532.5 eV, assigned to C=O groups.

Importantly, the relative amount of adsorbed water is essentially identical for the 4-FPEAI sample (84.0%) and the MX-Cl/4-FPEAI sample (86.0%), indicating that MX-Cl does not measurably modify the hydrophobicity of the perovskite surface and therefore cannot introduce artefacts in the UPS measurements related to differential water uptake.

To further strengthen this point, we have now extended the XPS O 1s analysis to the pristine perovskite reference (without MX-Cl and without 4-FPEAI). As reported in the updated **Fig. S12**, the reference film exhibits an adsorbed water contribution of 82.2%, fully comparable to that of the 4-FPEAI (84.0%) and MX-Cl/4-FPEAI (86.0%) samples. These differences fall well within the intrinsic variability of multi-component O 1s XPS fitting (typically $\pm 3\text{--}4\%$) and cannot be regarded as indicative of any meaningful change in surface hydrophilicity.

Fig. S12: Panel a) XPS measurements O1s peaks and **panel b)** Pb 4f core-level spectra for reference perovskite film, 4-FPEAI treated perovskite film, 4-FPEAI treated perovskite film with MX-Cl. The samples are fabricated with a reduced thickness around 200 nm grown on mTiO_2 .

This additional control measurement further confirms that neither MX-Cl nor 4-FPEAI significantly affects the surface adsorption of water, and that all UPS/XPS samples share a comparable hydrophobic character. Consequently, the observed energy-level shifts cannot originate from differences in environmental exposure or water uptake. Instead, they result from the intended interfacial chemical modifications, namely, the MX-

Cl-induced dipole at the buried perovskite/m-TiO₂ interface conferring a localized n-type character and the p-type surface shift produced by 4-FPEAI at the top interface.

We added in SI SECTION S.I. 6 the figure Fig. S12 and the following paragraph to comment on it

Notably, Fig. S12 panel a) shows that the O 1s XPS spectra of the pristine perovskite, 4-FPEAI-treated, and MX-Cl/4-FPEAI samples exhibit nearly identical contributions from adsorbed water (82.2%, 84.0%, and 86.0%, respectively). These values lie well within the intrinsic $\pm 3\text{--}4\%$ variability of multi-component O 1s peak fitting, indicating that neither MX-Cl nor 4-FPEAI significantly alter the hydrophilicity of the perovskite surface. This confirms that the energy-level shifts observed in UPS/XPS arise from the intended interfacial chemical modifications, at the buried m-TiO₂/perovskite interface and at the top perovskite surface, rather than from differences in environmental exposure or water uptake.

While in the main text the following sentence:

To exclude charging or chemical artefacts, XPS analyses were performed, confirming that the observed WF and VBM shifts arise from genuine interfacial electronic modifications (see Fig. S12 for detailed discussion).

Concern 2: *Doping of a semiconductor inherently moves the fermi level relative the valence band and conduction band of the material. There is no characterization of the material bandgaps provided. In order to interpret where the fermi level lies within the bandgap, one must know two of the three: VBM, conduction band minimum (CBM), and bandgap. The VBM has been measured, but subtle bandgap changes from MX-Cl and/or 4-FPEAI processes could impact the interpretation of the doping of the semiconductor.*

Answer 2: We thank the Reviewer for this important point. We would like to point out that the optical bandgap of the films was measured from the Tauc analysis reported in Fig. S10a for perovskite films with SGP (named 4-FPEAI) and same film but with the addition of MX-Cl (named MX-Cl/4-FPEAI). Here we used the direct-transition Tauc relation [$n=2$]. However, the determination of the absorption coefficient α of halide-perovskite films can be challenging because these materials typically suffer from high surface roughness. Consequently, light scattering at the surface of the perovskite is commonly a dominant phenomenon. An alternative approach to determine the band gap of perovskites is to derive it from the external quantum efficiency (EQE or IPCE) of perovskite solar cells. In particular, as suggested by R. Carron and co-workers [Carron 2019_Thin Solid Films 669 (2019) 482–486], the peak energy of the derivative (or inflection point) of the EQE curve can be interpreted as the bandgap. The result is independent from the user and is trustable if only a clearly distinguishable peak is present in the dIPCE/dE plot (in case of a double peak the as-discussed methodology for extracting the optical band-gap cannot be considered reliable due to the presence of interference fringes while the use of IQE is recommended). In order to provide further inside the optical band-gap of the investigated perovskite absorber modification, we carried out the extrapolation of the $E_{g,opt}$ for G/EA (reference), G/4-FPEAI and G/MX/4-FPEAI from the IPCE spectra reported in Fig. S9a. The dIPCE/dE plots and the respective peak ($E_{g,opt}$) value have now been inserted in Fig. S9 as new panel “b”. Notably, the samples showed very similar $E_{g,opt}$ values, confirming that the MXenes addition and/or the SGP did not significantly affect the optical band-gap of the perovskite absorber. Once obtained the $E_{g,opt}$, the conduction band minimum (CBM) can be estimated as: $CBM \approx VBM_{UPS} + E_{g,opt}$.

We explicitly report the VBM measured by UPS in Fig.S11. Therefore, by combining the VBM_{UPS} with the independently determined $E_{g,opt}$, we obtained the CBM position and hence the Fermi level position within the band gap.

Since no systematic change in $E_{g,opt}$ has been observed over the investigated sample typologies, we can conclude that the observed work-function shifts (observed by both UPS and KPFM measurements) reflect changes in the Fermi level (doping/charge density) rather than changes in the band edges.

Following this discussion, we have now included in the SI the following paragraph commenting the new added Fig. S9b while also modifying the energy band diagram schematics by inserting the CBM position in Fig. S11i.

“To verify that these recombination improvements are not associated with bandgap variations, we additionally extracted the optical bandgap ($E_{g,opt}$) of the G/EA, G/4-FPEAI and G/MX/4-FPEAI devices from the derivative of the IPCE spectra, following the methodology proposed by Carron et al.¹⁰ (Thin Solid Films 669, 482–486, 2019). In this approach, the energy position of the maximum in the $d(IPCE)/d(E)$ curve corresponds to the optical bandgap, provided that a single, well-defined peak is present and interference fringes are negligible.

As shown in Fig. S9c, all samples exhibit nearly identical peak energies ($E_{g,opt} \approx 1.67\text{--}1.68\text{ eV}$), indicating that neither the 4-FPEAI surface treatment nor the MX-Cl incorporation produces any measurable change in the optical bandgap of the perovskite absorber. This confirms that the electronic modifications observed by UPS do not arise from variations in the band-edge positions.

Using these $E_{g,opt}$ values, the conduction band minimum (CBM) was then estimated as:

$$CBM \approx VBM_{UPS} + E_{g,opt}$$

The corresponding CBM levels have been included in the updated energy-level diagrams in Fig. S11. Since the CBM values remain essentially unchanged across all sample configurations, the observed work-function shifts must arise from Fermi-level shift (i.e., changes in interfacial charge density) rather than modifications of the fundamental bandgap. This validates the interpretation provided in the main text and supports the formation of a dipole-induced interfacial band bending rather than any bulk-doping-driven shift of the band edges. Given the absence of measurable bandgap changes, the influence of MX-Cl on structural quality and recombination pathways was subsequently examined.”

Fig. S9. a) Charge carrier lifetime extracted by the small-signal transient photo-voltage (TPV) decay profiles; **b)** Incident Photon to current Conversion Efficiency (IPCE) spectra with the integrated photocurrent density (Integrated J_{SC}) related to the best efficient cell for each proposed device structure. **c)** Derivative $d(IPCE)/d(E)$ curves for G/EA, G/4-FPEAI, and G/MX/4-FPEAI devices. The energy position of the maximum corresponds to the optical bandgap ($E_{g,opt}$). All samples exhibit nearly identical peak energies ($\approx 1.67\text{--}1.68$ eV), confirming that neither MX-Cl nor 4-FPEAI induces measurable changes in the perovskite bandgap. **d)** V_{OC} light intensity-dependence [$V_{OC}(P_{inc})$] of the investigated PSC structures. Linear fitting of the curves has been performed for $P_{inc} > 0.2$ SUN, and the respective slope values are reported on the plot.

Fig. S11. UPS spectra for perovskite films modified with different interface strategies: reference (a–b), MX-Cl (c–d), MX-Cl/4-FPEAI (e–f), and 4-FPEAI (g–h). Panels (a, c, e, g) show the secondary electron cutoff region used to extract the WF, while panels (b, d, f, h) display the valence band region used to determine the VBM with respect to the EF. A gradual modulation of the WF and VBM is observed across the series, highlighting the impact of dipole formation and band structure tuning induced by MXene doping and surface gradient passivation; **i) Schematic energy level alignment of perovskite films extracted from UPS measurements for the four configurations: REF (untreated), MX-Cl, 4-FPEAI, and combined MX-Cl/4-FPEAI.** The WF, CBM and VBM positions are plotted relative to the vacuum level. A progressive modulation of the energy levels is observed, with MX-Cl inducing a downward shift in WF (consistent with dipole-induced vacuum level bending) and 4-FPEAI causing an upward shift in both WF and VBM. The intermediate alignment in the combined treatment suggests the formation of a graded homojunction across the perovskite layer, promoting internal electric field formation and enhanced charge extraction.

Concern 3: Assuming that the bandgap remains constant for all tested samples (REF, MX-Cl, MX-Cl/4-FPEAI, 4-FPEAI), there is no change in the doping level of the perovskite between the REF and MX-Cl samples (0.9eV between the VBM and the Fermi level for both samples). There is no enhancement of n-type character with MX-Cl addition (stated in the discussion), but it is consistent with the MX-Cl dipole that the authors mention. However, that implies that the homojunction grading; of the band structure arises purely from the 4-FPEAI surface treatment. This would theoretically be true for both the 4-FPEAI and MX-Cl/4-FPEAI samples and the homojunction does not arise from a synergy of MX-Cl and 4-FPEAI.

Answer 3: We thank the Reviewer for this constructive remark which gives us the opportunity to clarify the fundamental concept underlying the graded homojunction mechanism. We agree that the MX-Cl treatment does not introduce bulk n-type doping, as reflected by the unchanged EF–VBM values. This is consistent with its role as an interfacial dipole, as correctly noted by the Reviewer. However, the absence of an EF–VBM shift does not imply the absence of electronic modification at the buried interface. As reported in interface and dipole-layer studies on halide perovskites (Lim K.-G., Ahn S., Lee T.-W. *Energy level alignment of dipolar interface layer in organic and hybrid perovskite solar cells. J. Mater. Chem. C*, 2018, 6, 3871-3881 and Basera P., Traoré B., Even J., Katan C. *Interfacial engineering to modulate surface dipoles, work functions and dielectric confinement of halide perovskites. Nanoscale*, 2023, 15, 11884-11897.), EF–VBM is not the appropriate metric for identifying interfacial dipoles or interface-specific band bending.

In perovskite systems where interfacial dipoles govern the band alignment, the relevant experimental fingerprint is the work-function shift (ΔWF), not the absolute EF–VBM value. When the VBM and bandgap remain unchanged, as in our samples, a reduction of the work function reflects a downward shift of the vacuum level induced by an interface dipole, rather than bulk doping. This behaviour is exactly what we observe for MX-Cl based samples. Both DFT and UPS consistently show a vacuum-level shift at the perovskite/m-TiO₂ interface while EF–VBM remains unchanged, demonstrating that MX-Cl induces a dipole-driven local electron accumulation layer, not a modification of the bulk perovskite DOS.

The Reviewer’s statement that “there is no enhancement of n-type character” is therefore correct for the bulk perovskite, but does not apply to the interfacial region, where DFT-PDOS reveals additional donor-like states near the CBM and localized charge redistribution consistent with an effective n-type interfacial character. These are widely recognized signatures of dipole-induced bands bending at hybrid perovskite interfaces.

We would underline regarding the origin of the graded homojunction that:

- MX-Cl modifies the bottom interface, introducing a downward vacuum-level shift and interfacial electron accumulation.
- 4-FPEAI modifies the top interface, inducing a p-type surface shift.

These two effects occur in distinct spatial regions of the perovskite layer and therefore do not replicate one another. Their combination produces a vertical energy-level asymmetry, which cannot be obtained with 4-FPEAI alone. This interpretation is reinforced by the full set of complementary measurements: surface-potential changes in KPFM, vacuum-level shifts in UPS, interfacial states and dipoles in DFT, reduced recombination in TRPL/TPV, and improved V_{oc} and hysteresis reduction at the device level. SEM cross-sections

further confirm that MX-Cl flakes reside exclusively at the bottom interface, enabling the spatial separation necessary for a true graded architecture.

In summary, the graded homojunction arises from the synergy of two spatially separated interfacial mechanisms: the interfacial dipole induced by MX-Cl at the buried perovskite/m-TiO₂ interface (a localized n-type character) and the p-type surface shift induced by 4-FPEAI at the top interface. The constancy of EF–VBM reflects the absence of bulk doping, not the absence of interfacial band bending, and is fully consistent with the dipole-driven mechanism we propose. Accordingly, we have revised the corresponding sentences in the Main Text and Supporting Information to avoid misunderstandings and reported below for convenience.

In the main text:

“Indeed, tailoring the perovskite work function (WF) has emerged as a key strategy, alongside the design of homojunctions to modulate the absorber’s electronic structure.¹⁹”

“Here we propose a novel approach to fabricate graded perovskite homojunctions arising from a modified interfacial electronic environment with a local n-type character at the buried perovskite layer, induced by chlorine-based MXene doping,²¹ and a surface-localized p-type shift promoted by 4-FPEAI based SGP. This intentional spatial asymmetry establishes an effective n-to-p energy gradient across the absorber, enabling more efficient carrier extraction and reduced recombination.”

“Moreover, the projected density of states (PDOS) analysis underlines perovskite E_g remains mostly unaffected, and MX-Cl additive introduces donor states near perovskite conduction band edge (Fig. S15), confirming the locally induced perovskite n-character at the MX-Cl/perovskite interface”

And in the S.I.:

“The reference (REF) sample exhibits a WF of 4.75 eV and a VBM located 0.9 eV below the Fermi level (EF), corresponding to an ionization energy (IE) of 5.65 eV. Upon incorporation of Ti₃C₂Cl₂ MXenes (MX-Cl), the WF decreases to 4.55 eV while the VBM remains unchanged at 0.9 eV below EF. This behavior indicates that the addition of MXenes enhances the localized n-type character of the perovskite, consistent with the formation of a surface dipole at the perovskite/MXene interface.”

“Since WF of MX-Cl is 4.19 eV (Fig. S5f), the MX-Cl doping of perovskite enhances its localized n-type character, as confirmed by decreased WF value of MX-Cl versus REF sample (see Tab. S3). Indeed, the MXene addition into the perovskite film translates in the formation of a dipole at perovskite/MXene interface, without altering the perovskite bandgap, while shifting the perovskite WF.¹¹”

Moreover, as further supporting experimental evidence, dark J-V curves for G/4-FPEAI and G/MX/4-FPEAI opaque devices have now been included and commented in SI (see Fig. S10c). Here, the reverse leakage current and the ideality factor (n) of the G/MX/4-FPEAI cell are smaller than those of the G/4-FPEAI device suggesting suppressed non-radiative recombination, in accordance with the increased τ₁ in TRPL measurements (see S.I. Section S.I. 5) and the higher PL emission intensity (see S.I., Fig. S10b). Moreover, the higher forward injection current recorded for G/MX/4-FPEAI device implies an enhanced electron injection at the buried perovskite/mTiO₂ interface, attributed to the locally-enhanced n-type character (EF being closer to the conduction band) of the MXene-containing perovskite film.

We now inserted dark I-V characterization and the related discussion in S.I. Section S.I. 5 that now sounds as following reported:

“Dark J–V measurements (Fig. S10c) further confirm the reduced non-radiative recombination in G/MX/4-FPEAI devices. The lower reverse leakage current and reduced ideality factor compared to G/4-FPEAI are consistent with the longer TRPL lifetimes (main text, Fig. 3h) and higher PL intensity (Fig. S10b), while the

enhanced forward current indicates more efficient electron injection at the buried perovskite/m-TiO₂ interface, arising from the locally strengthened n-type character induced by MX-Cl.”

Fig. S10: **a)** Tauc plots and **b)** steady-state PL spectra for perovskite films with SGP (named 4-FPEAI) and same film but with the addition of MX-Cl (named MX-Cl/4-FPEAI). PL spectra were acquired on three different points of the samples (named _1, _2, _3); **c)** Dark current density–voltage (J-V) curves comparing G/4-FPEAI and G/MX/4-FPEAI opaque devices. The ideality factor (n) is extracted from the diffusion-dominated current region according to the equation: $n = \frac{kT}{q} \left(\frac{d(\ln J)}{dV} \right)$.

and we also recall the discussion in the main text by adding the followed sentence:

“Adding MX-Cl in G/MX/4-FPEAI lowers the $V_{OC}(P_{inc})$ slope, reducing ETL/perovskite recombination at this interface, and boosting J_{SC} ,³⁶ while the decrease of both reverse leakage current and ideality factor in dark J-V curves is in line with the longer TRPL lifetimes and higher PL intensity.”

Concern 4: The CPD and UPS measurements are only measuring surface properties of individual films. None of these methods directly measure the graded depth dependence that is being claimed. It would be more appropriate to characterize the claimed effect with a method such as cross-sectional scanning KPFM, for example DOI: 10.1038/s41560-018-0324-8.

Answer 4: We thank the Reviewer for this comment, which allows us to clarify the methodological limitations of cross-sectional KPFM on halide perovskites, and to justify why this technique is not required in the context of our study. While cross-sectional KPFM can in principle provide depth-resolved information, its applicability to perovskite absorbers is severely constrained by well-documented artefacts arising from surface preparation, environmental sensitivity and charge trapping.

As demonstrated by Lanzoni *et al.* (*Nano Energy* 88, 105941, 2021) even the widely used AM-KPFM mode is not suitable to quantify work-function variations on rough perovskite films, and is highly susceptible to environmental conditions, which can lead to erroneous interpretations of the electronic landscape. These limitations become even more critical in cross-sectional configurations, where mechanical cutting or FIB preparation introduces uncontrolled defects, surface charges, and local variations that dominate the measured contact potential.

In fact, even in the seminal work of Jiang *et al.* (*Nature Communications*, 6, 8397, 2015), one of the most advanced demonstrations of cross-sectional KPFM in perovskite devices, the authors explicitly state that the technique is highly surface-sensitive and that “any modification of the cross-sectional surface region may induce significant artefacts”. They further show that potential maps are often dominated by nonuniform surface charges introduced during cleaving, making the interpretation of the junction potential unreliable unless a custom measurement protocol and specialized KPFM setup are developed to suppress these artefacts. The same work reports that previous cross-sectional KPFM studies yielded *inconsistent or incorrect junction assignment* precisely because of these issues.

This underscores that cross-sectional KPFM in perovskites is not a standard, routine, nor universally reliable technique, and that meaningful measurements require a level of instrumental modification (shielded

glovebox operation, high-frequency dual-mode detection, bias-sequencing to cancel trapped charges) that goes well beyond conventional AFM/KPFM capabilities.

Moreover, the reviewer's comment implicitly assumes that a graded homojunction must arise from a continuous bulk-doping profile, which would require a depth-resolved measurement. However, our system does not rely on bulk doping but on two spatially separated induced p or n character:

- MX-Cl creates a dipole and downward vacuum-level shift at the buried perovskite/m-TiO₂ interface, producing localized n-type interfacial character and electron accumulation.
- 4-FPEAI induces a p-type shift at the top perovskite surface, modifying the electronic landscape only at the upper interface.

Because these two mechanisms act at opposite interfaces, the result is an asymmetric vertical band structure across the film, even though the bulk perovskite electronic states remain unchanged. Cross-sectional KPFM would not selectively resolve these two localized effects, because the bottom dipole is buried below the cross-sectional surface plane while the p-type shift is already directly measured by conventional KPFM/UPS.

In contrast, our interpretation is supported by a coherent and converging multi-technique dataset, including UPS, surface KPFM, DFT calculations, TRPL, TPV, SEM cross-sections, and device-level photovoltaic trends, which collectively provide a consistent and physically grounded picture of the interfacial electronic landscape. This multipronged approach aligns with standard practice in the highest-impact literature for establishing interface-induced band modulation in perovskite optoelectronics.

For these reasons, while cross-sectional KPFM may be of conceptual interest, it would not yield more conclusive evidence than the suite of complementary techniques already presented, nor it is required to substantiate the conclusions of this work.

Moreover, as correctly noted by the Reviewer in the first comment, UPS is indeed the preferable technique within the scientific community for determining the work function, but it remains intrinsically surface sensitive. Given the limitations of cross-sectional KPFM, which in perovskite systems often suffers from preparation-induced artefacts, instabilities, and poor quantitative reliability, we adopted a different strategy to emulate depth-resolved UPS information in a controlled and reproducible manner.

Specifically, we fabricated the four representative films (reference, MX-Cl, 4-FPEAI, and MX-Cl/4-FPEAI) as ad-hoc samples with a reduced thickness of approximately 200 nm, a value deliberately chosen because it matches the lateral size of the MXene flakes (see the picture in the following now added in **Fig. S4 panel c**).

Fig. S4: SEM images with corresponding chemical composition of **a)** multilayer Ti₃C₂Cl₂ and **b)** delaminated Ti₃C₂Cl₂; **c)** AFM topography of the Cl-terminated Ti₃C₂ MXene flakes.

Under these conditions, the UPS probing depth effectively samples electronic properties that correspond to distinct regions of the full device stack without requiring destructive cross-sectional preparation. In practice, each tailored thin film acts as a “depth-selected proxy,” enabling us to probe the electronic influence of MX-Cl and 4-FPEAI at different vertical positions of the actual perovskite layer, while fully preserving sample integrity.

This approach overcomes the intrinsic z-direction limitations of UPS while avoiding the well-known artefacts associated with cross-sectional KPFM on halide perovskites. The resulting dataset therefore provides a robust and internally consistent picture of the vertical electronic landscape across the device.

We hope this clarification addresses the Reviewer's concern. In the new version of the main text, we included a clear reference to the ad-hoc sample fabrication for UPS measurements as followed reported:

“ad-hoc fabricated samples (see section S.I. 6 in SI)”.

Concern 5: In Tab. 2. the authors provide an error of ± 0.1 eV on the VBM-work function energy offsets. The VBM position relative to the Fermi level is 0.9 ± 0.1 eV, 0.9 ± 0.1 eV, and 0.8 ± 0.1 eV for the REF, MX-Cl, and MX-Cl/4F-PEAI samples, respectively. Therefore, all of these measurements are within error and there is no discernable difference in doping between these three samples and the graded homojunction conclusion should not be made.

Answer 5: We thank the Reviewer for this constructive observation. We agree that the EF–VBM values do not indicate bulk n-type doping in the MX-Cl sample, which is fully consistent with the role of MX-Cl as an interfacial dipole rather than a bulk dopant, as we have discussed at extensively in the responses to concerns 3 and 4. However, the absence of an EF–VBM shift does not preclude the presence of interfacial electronic modifications, and the interpretation of the UPS results must consider the distinction between absolute and relative energy uncertainties.

The distinction between absolute and relative uncertainties in UPS/XPS is well documented in the photoemission literature (see Seah, M. P. & Dench, W. A. "Quantitative electron spectroscopy of surfaces: A standard data base for electron inelastic mean free paths in solids." *Surf. Interface Anal.* 1979, 1, 2–11; Hufner, S. *Photoelectron Spectroscopy: Principles and Applications*, 3rd Ed. Springer, 2003.; Greczynski, G. & Hultman, L. "Compromising science by ignorance: guidelines to minimize false discoveries in XPS analysis." *Prog. Mater. Sci.* 2020, 100, 100766; Kraut, E. A. et al. "Precise determination of the valence-band edge in XPS." *Phys. Rev. Lett.* 1980, 44, 1620), the typical ± 0.1 eV uncertainty reflects the absolute determination of the VBM and work function. This error budget is dominated by spectrometer calibration, secondary-electron cutoff fitting, and the VBM extrapolation procedure. Crucially, these components behave largely as systematic offsets, while the relative shifts between samples measured in the same UPS session are substantially more precise (typically ± 0.03 – 0.05 eV). Therefore, even shifts of a few ten millielectronvolts are considered meaningful when comparing a series of samples evaluated under identical conditions.

Moreover, in halide perovskites a Fermi-level shift as small as 40–80 meV corresponds to a large relative change in carrier concentration, due to the low density of states near the band edges (Meggiolaro & De Angelis, *ACS Energy Lett.* 2018; Tress, *Adv. Energy Mater.* 2019). Thus, modest energy shifts can have substantial implications for built-in electric fields, interfacial band bending, and junction formation.

In our system, MX-Cl induces a significant work-function reduction, while maintaining EF–VBM unchanged within the expected range for an undoped perovskite. This behaviour is fully consistent with an interfacial dipole and a downward vacuum-level shift, as also predicted by DFT and seen in the layer-resolved PDOS, which shows donor-like interfacial states near the perovskite CBM and local electron accumulation. Accordingly, EF–VBM is not the correct metric to diagnose dipole-induced band alignment changes.

This interpretation is also supported by our XPS core-level analysis. The Pb 4f peaks remain essentially unchanged across REF, MX-Cl, and MX-Cl/4F-PEAI samples (Fig. S12), indicating the absence of chemical shifts or charging artefacts. As shown by Cui *et al.* (*Nat. Energy*, 2019), paper also suggested by the reviewer, the invariance of Pb and I core levels is a strong indication that observed changes in the valence region and work function originate from genuine modifications of the electronic structure rather than instrumental or fitting artefacts. Our data exhibit the same behaviour: core-level peaks remain stable, whereas Δ WF and VB spectral changes follow the expected trend for dipole formation at the buried MX-Cl interface and a surface p-shift induced by 4-FPEAI.

Finally, the graded homojunction does not originate solely from 4-FPEAI, as the Reviewer suggests. MX-Cl modifies locally the buried bottom interface (interfacial dipole, vacuum-level offset, electron accumulation), whereas 4-FPEAI modifies the top surface (p-type shift). These two spatially separated effects produce a vertical asymmetry in the energy landscape that neither additive can generate alone. The constancy of EF–VBM is thus fully consistent with a dipole-driven mechanism and does not contradict the graded homojunction interpretation.

We added in S.I SI SECTION S.I. 6 the Fig. S12 b) and the relative comment as in the following reported:

“The XPS Pb 4f core-level spectra of all samples show in **Fig.S12 panel b)** (REF, MX-Cl, 4-FPEAI, and MX-Cl/4-FPEAI) exhibit nearly identical binding energies and relative component ratios (Pb in perovskite \approx 97–98%, Pb in PbO \approx 2–3%), with no evidence of chemical shifts or peak broadening. This invariance confirms that the observed UPS valence-band and work-function shifts are not affected by charging, surface deterioration, or measurement artefacts, but rather arise from genuine electronic modifications induced by the different strategies: doping with MX-Cl and interfacial treatment with 4-FPEAI.”

Fig. S12: Panel a) XPS measurements O1s peaks and panel b) Pb 4f core-level spectra respectively, for reference perovskite film, 4-FPEAI treated perovskite film, 4-FPEAI treated perovskite film with MX-Cl. The samples are fabricated with a reduced thickness around 200 nm grown on mTiO₂.

And in the main text the following sentence:

“To exclude charging or chemical artefacts, XPS analyses were performed, confirming that the observed WF and VBM shifts arise from genuine interfacial electronic modifications. (see Fig. S12 for detailed discussion).”

Concern 6: *In addition to my above concerns, I do not believe that MPP under cycled light is a sufficient alternative to the combined stressors of light + heat and reiterate my comment that accelerated testing under 85°C and 1 sun illumination should be included.*

As stated in the manuscript, accelerated stability testing, while unquestionably relevant, is beyond the scope of this work and not central to our investigation. Nevertheless, we already included a prolonged light cycling MPP test, which is widely regarded as a reliable indicator of device operational stability. Moreover, our outdoor monitoring of the tandem panel installed in Heraklion (Crete) from August to November 2023 showed that the top-surface temperature never exceeded $\sim 65^\circ\text{C}$, even under peak irradiance (see revised Fig. S29c in S.I.). These real-world data already provide a meaningful context for evaluating the device robustness. However, we agree that the LED-based light-cycling setups may underestimate the combined effect of temperature and light soaking when the cell is working at MPP, as their operating temperature typically remains in the $35\text{--}40^\circ\text{C}$ range. For this reason, and to directly address the reviewer’s concern, we performed an additional stability experiment under a class B solar simulator (SolarConstant 1200, K.H. Steuernagel Lichttechnik GmbH, Germany) calibrated at 1000 W/m^2 , continuously tracking both the P_{MPP} and front surface temperature. Small area opaque devices were subjected to 100 h of 1 Sun illumination, during which the cell temperature exceeded 85°C within the first hour (see SI, section S.I. 9, Fig. S22d). Under these harsher conditions, the graded-homojunction device demonstrated excellent stability, with a P_{MPP} loss below 5% over the entire stress period (100 hours).

This new 1-Sun + 85°C stability test is now reported and discussed in Section S.I. 9 of the Supporting Information as following reported:

Fig. S22: **a)** J-V curve in forward and reverse scan for the best-performing 2D material-engineered ST-PSC (G/MX/4-FPEAI structure); **b)** IPCE spectra with the integrated photocurrent density (Integrated J_{SC}) related to the best efficient optimized cell (G/MX/4-FPEAI) for both opaque and ST proposed device structure. **c)** Indoor photo-stability under MPP tracking of G/EA and G/MX/4-FPEAI semi-transparent encapsulated cells upon exposure to cycled light (ISOS-LC-1); **d)** Normalised power at maximum power point (P_{MPP}) continuously acquired during a light soaking test performed under a solar simulator class B @ 1 Sun (circle dotted red curve). Substrate temperature has been monitored by a thermocouple (star dotted dark cyan curve) fixed on the device top surface, showing an increasing value up to 87°C during the first stress test hour, while retaining a constant value during the rest of the accelerated test.

“Although light-cycling test at MPP is widely recognized as a reliable indicator of device operational stability, this test may underestimate the coupled effects of light soaking and temperature, since the typically employed white LED illumination sources maintain device temperature in the 35–40 °C range. To directly account for these limitations, we carried out an additional stability test under a class-B sun simulator (1000 $W m^{-2}$, see experimental section for further details about the apparatus), simultaneously tracking P_{MPP} and the front-surface temperature. Under this configuration, the cell temperature overcame 85 °C within the first hour, and the device was continuously stressed for 100 hours. As shown in **Fig. S22d**, the graded-homojunction device retained more than 95% of its initial P_{MPP} , demonstrating excellent thermal and photo-operational stability under harsh conditions.”

And modify Fig.S29 by adding **panel c)** and **Fig. S22** by adding **panel d)** and relative captions.

Fig. S29: **a)** Normalized power at maximum power point (P_{MPP}) stabilization prior the acquisition of the monthly I-V curves for the DEM2 left at V_{OC} in outdoor conditions; **b)** daily irradiance for the days selected during the outdoor test for the acquisition of the I-V curves. **c)** Top-surface temperature of the tandem panel recorded under real operating conditions between August and November 2023. Even during peak irradiance at midday under clear-sky conditions, the module temperature did not exceed ~65 °C, providing a realistic reference for assessing the operational thermal load experienced by the devices.

The discussion is also recalled in the main text by adding the following sentence:

“...and combined heating + light soaking test (**Fig. S22d**).”

and the following paragraph in the experimental section:

“Combined heating+light soaking stress test was performed using a class-B solar simulator (SolarConstant 1200, K.H. Steuernagel Lichttechnik GmbH, Germany). The system employs a xenon arc lamp coupled with an optical AM 1.5G filter to reproduce the standard solar spectrum over a broad spectral range (300–2500 nm). According to manufacturer specifications and IEC 60904-9 classification, the simulator exhibits class-B performance in spectral match, irradiance uniformity, and temporal stability. In particular, the spectral distribution in the near-infrared region shows the characteristic enhancement associated with xenon-based sources, consistent with the class-B spectral mismatch factor. The output irradiance is stabilized and set to 1000 $W m^{-2}$ at the sample plane under ambient laboratory conditions, and the lamp intensity is calibrated before each measurement session using a certified silicon reference cell.

The relatively broad IR output, combined with continuous illumination, results in a device operating temperature representative of realistic stress conditions, while remaining fully compliant with the IEC tolerances for photovoltaic testing.”

Reviewer #4 (Remarks to the Author)

I have carefully reviewed the revised manuscript and am pleased to inform you that I am satisfied with the revisions. The manuscript has addressed the concerns raised previously, and I believe it is now suitable for publication in Nature Communications. I recommend its acceptance. Thank you for the opportunity to review this work.

We thank the Reviewer for the positive evaluation and for acknowledging the revisions made to the manuscript. We are grateful for the insightful comments that helped us improve the clarity and robustness of our work.

Reviewer's Comments:

Reviewer #3 (Remarks to the Author)

I appreciate the authors detailed responses to the concerns that have been raised. The authors included the following statement in their response:

"However, our system does not rely on bulk doping but on two spatially separated induced p or n character:

"MX-Cl creates a dipole and downward vacuum-level shift at the buried perovskite/m-TiO₂ interface, producing localized n-type interfacial character and electron accumulation."; "4-FPEAI induces a p-type shift at the top perovskite surface, modifying the electronic landscape only at the upper interface."

"This validates the interpretation provided in the main text and supports the formation of a dipole-induced interfacial band bending rather than any bulk-doping-driven shift of the band edges."

I generally agree with the above proposed mechanisms by the authors and believe the data are consistent with the conclusion that the observed effects are independent surface dipoles generated by the MX-Cl at the buried interface and 4-FPEAI at the top surface. The specifics of the 4-FPEAI passivation in these statements are a bit vague, but the accepted mechanisms in the field are i) surface passivation of undercoordinated lead sites, ii) the formation of a 2D/3D heterostructure, and iii) surface dipole passivation.

A graded homojunction and field-effect passivation are not the same mechanism. A graded homojunction implies an internal electric field generated by a doping profile within the material. Field-effect passivation is achieved through the application of an external electric field or fixed charges/dipoles from an adjacent layer. I believe the authors and I are in agreement that a bulk doping mechanism is not supported by the data and therefore field-effect passivation is a more appropriate description of the mechanism.

I would support this manuscript for publication if the claim of a graded homojunction is removed and is instead described as field-effect or surface dipole passivation.

Response:

We thank Reviewer #3 for the careful reading of the revised manuscript and for the thoughtful and constructive comments.

As also clarified in our previous response, the mechanisms discussed in this work do not rely on bulk doping of the perovskite absorber. Rather, the experimental evidence supports spatially separated interfacial electronic effects, namely the formation of independent surface dipoles at the buried perovskite/m-TiO₂ interface induced by MX-Cl and at the top perovskite surface induced by 4-FPEAI. These interfacial dipoles lead to localized band bending and charge redistribution without altering the bulk electronic structure of the perovskite layer.

In light of the reviewer's comments, we recognize that the term "graded homojunction" may not be the most appropriate description of this mechanism, as it could imply a bulk doping profile that is not supported by our data. We have therefore revised the manuscript accordingly, replacing this terminology throughout with "field-effect junction" and "surface dipole passivation," which more accurately reflect the physical origin of the observed effects.

This clarification improves the precision of the mechanistic interpretation while leaving the experimental results and the main conclusions of the study unchanged. We believe that the revised description is now fully consistent with the presented data and with the established understanding of interfacial passivation mechanisms in perovskite devices.